# Oncogenic *PIK3CA* induces centrosome amplification and tolerance to genome doubling

Inma M. Berenjeno[1], Roberto Piñeiro[1,2], Sandra D. Castillo[1], Wayne Pearce[1], Nicholas McGranahan[3], Sally M. Dewhurst[3], Valerie Meniel[4], Nicolai J. Birkbak[3], Evelyn Lau[1], Laurent Sansregret[1,3], Daniele Morelli[1], Nnennaya Kanu[3], Shankar Srinivas[5], Mariona Graupera[6], Victoria E.R. Parker[7], Karen G. Montgomery[8], Larissa S. Moniz[1], Cheryl L. Scudamore[9], Wayne A. Phillips [8], Robert K. Semple[7], Alan Clarke[4], Charles Swanton[1,3] & Bart Vanhaesebroeck[1]

Mutations in *PIK3CA* are very frequent in cancer and lead to sustained PI3K pathway activation. The impact of acute expression of mutant *PIK3CA* during early stages of malignancy is unknown. Using a mouse model to activate the *Pik3ca*[H1047R] hotspot mutation in the heterozygous state from its endogenous locus, we here report that mutant *Pik3ca* induces centrosome amplification in cultured cells (through a pathway involving AKT, ROCK and CDK2/Cyclin E-nucleophosmin) and in mouse tissues, and increased in vitro cellular tolerance to spontaneous genome doubling. We also present evidence that the majority of *PIK3CA*[H1047R] mutations in the TCGA breast cancer cohort precede genome doubling. These previously unappreciated roles of *PIK3CA* mutation show that PI3K signalling can contribute to the generation of irreversible genomic changes in cancer. While this can limit the impact of PI3K-targeted therapies, these findings also open the opportunity for therapeutic approaches aimed at limiting tumour heterogeneity and evolution.

[1] UCL Cancer Institute, Paul O'Gorman Building, University College London, 72 Huntley Street London, London WC1E 6DD, UK. [2] Roche-Chus Joint Unit, Complexo Hospitalario Universitario de Santiago de Compostela, Travesía da Choupana S/N, 15706 Santiago de Compostela, Spain. [3] The Francis Crick Institute, 1 Midland Road, London NW1 1AT, UCL Cancer Institute and Hospitals, 72 Huntley Street, London WC1E 6DD, UK. [4] European Cancer Stem Cell Research Institute, Cardiff University, Cardiff CF24 4HQ, UK. [5] Department of Physiology Anatomy and Genetics, University of Oxford, Oxford OX1 2JD, UK. [6] Vascular Signalling Laboratory, Institut d'Investigació Biomèdica de Bellvitge (IDIBELL), Barcelona 08908, Spain. [7] Institute of Metabolic Science, University of Cambridge, Addenbrooke's Hospital, Cambridge CB2 0QQ, UK. [8] Cancer Biology and Surgical Oncology Research Laboratory, Peter MacCallum Cancer Centre, Melbourne, 3000 VIC, Australia. [9] Mary Lyon Centre, MRC Harwell, Harwell OX11 0RD, UK. Inma M. Berenjeno and Roberto Piñeiro contributed equally to the work. Alan Clarke is deceased. Correspondence and requests for materials should be addressed to I.M.B. (email: i.berenjeno@ucl.ac.uk) or to C.S. (email: charles.swanton@crick.ac.uk) or to B.V. (email: bart.vanh@ucl.ac.uk)

Of the eight genes encoding catalytic PI3K subunits in mammals, only *PIK3CA*, which encodes the ubiquitously expressed p110α catalytic subunit, is frequently mutated in solid tumours[1]. *PIK3CA* mutations cluster in so-called hotspots, and give rise to a more active p110α protein that stimulates the PI3K pathway[2,3]. Thus far, the oncogenic potential of PI3K has largely been attributed to its role in stimulating processes such as cell survival and proliferation, spurring the development of inhibitors of the PI3K pathway as anti-cancer agents[3–7].

Several Cre recombinase-based mouse models have been created to explore the role of mutated p110α in cancer. Interestingly, whereas transgenic overexpression of mutant *Pik3ca* has been found to be an effective inducer of cancer[8], other models, in which mutated *Pik3ca* is expressed from its endogenous locus, demonstrate that mutant *Pik3ca*, on its own, is a weak oncogene (see for example ref. [9]), with cancer arising only after long latency periods (>1 year; reviewed in ref. [10]).

In the current study, we created an Flp recombinase-based knock-in mouse model of inducible expression of mutant *Pik3ca* from its endogenous locus. Using this model, we show that mutated *Pik3ca* is a weak oncogene on its own, but that it can cooperate with other oncogenic lesions, such as heterozygous loss of the *Apc* tumour suppressor. We also show that systemic induction of heterozygous mutant *Pik3ca* at embryonic or adult stages can have dramatic organismal consequences and leads to lethality.

We assessed signalling and cell biological changes induced early upon heterozygous expression of mutant *Pik3ca*, which allowed us to uncover two previously unappreciated roles of PI3K signalling, namely the induction of centrosome amplification and increased tolerance to tetraploidization, both of which have been implicated in tumourigenesis and tumour evolution[11–17].

## Results

### A mouse model of inducible endogenous p110α^H1047R expression.

As a model of physiological expression of mutant *Pik3ca* from its endogenous locus, we generated a mouse line in which one of the two wild-type (WT) *Pik3ca*^WT alleles is converted to *Pik3ca*^H1047R (Fig. 1a). Due to the presence of a neomycin (*Neo*) selection cassette in the targeted *Pik3ca* locus, the expression of the mutant p110α^H1047R protein was dampened, as shown in embryonic stem (ES) cells (Fig. 1b) and *Pik3ca*^H1047R+neo mouse embryonic fibroblasts (MEFs; Fig. 1c), resulting in minimal or no activation of the PI3K pathway, as assessed by Akt phosphorylation in these cells (Fig. 1b, c) and in *Pik3ca*^H1047R+neo mice (Supplementary Fig. 1a). A more detailed analysis of PI3K pathway activation in the inducible *Pik3ca*^H1047R+neo MEF system is reported in ref. [18].

In order to rescue the dampened expression of the p110α^H1047R protein, we removed the *Neo* cassette through recombination via its flanking frt sites. This was achieved by crossing *Pik3ca*^H1047R+neo mice with mice that express a germline Flp recombinase which is either constitutively active (*hACTB::Flpe* mice[19]) or inducible by tamoxifen (or its derivative 4-hydroxytamoxifen (4-OHT)) (*CAG::Flpe-ER*^T2 mice[20]). 4-OHT treatment of primary MEFs, isolated from *Pik3ca*^H1047R+neo mice crossed onto *CAG::FlpeER*^T2 mice, resulted in the removal of the *Neo* cassette (Supplementary Fig. 1b), restored p110α^H1047R expression levels similar to that of endogenous p110α^WT (Fig. 1c)

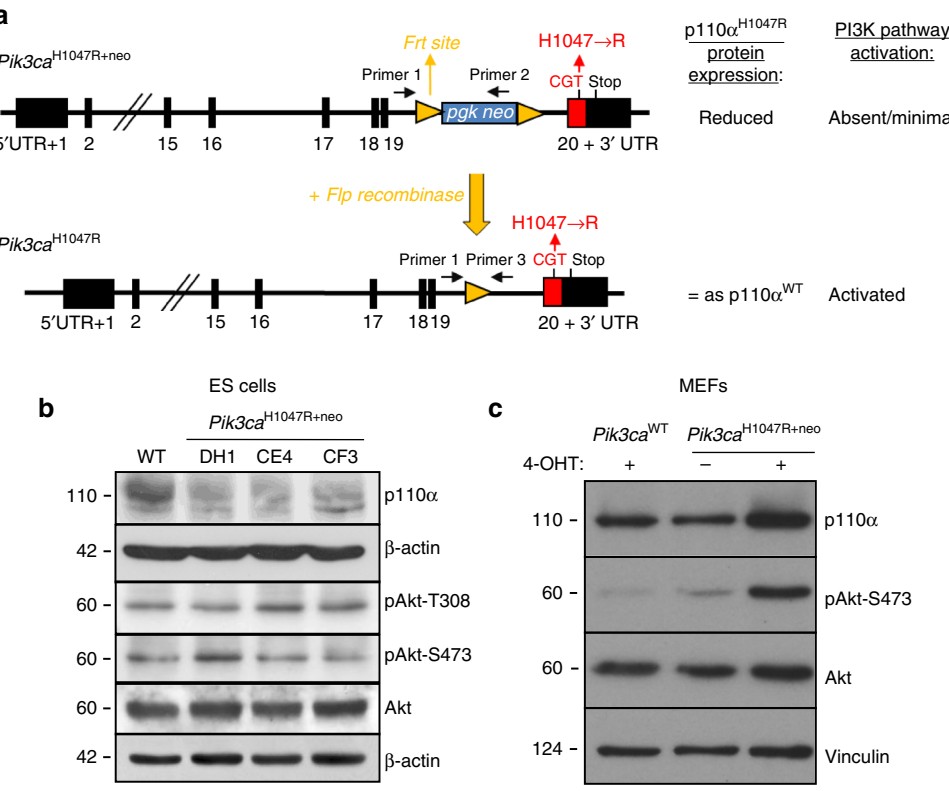

**Fig. 1** Gene targeting to create a hypomorph *Pik3ca*^H1047R+neo allele. **a** Targeted *Pik3ca* allele showing the *frt*-flanked *neo* selection cassette, before and after Flp-mediated recombination. Exon sequences are represented by filled black rectangles, intron sequences by a black line. The *frt* sites are represented as yellow triangles with the pointed end indicating orientation. The positions of the primers used for PCR screening are designated by arrows. **b** p110α expression levels and phosphorylation of Akt in *Pik3ca*^WT and *Pik3ca*^H1047R+neo ES cells. **c** p110α expression and Akt phosphorylation in *Pik3ca*^WT;*Flp-ER*^T2 and *Pik3ca*^H1047R+neo;*Flpe-ER*^T2 MEFs, 72 h after addition of 4-OHT or vehicle

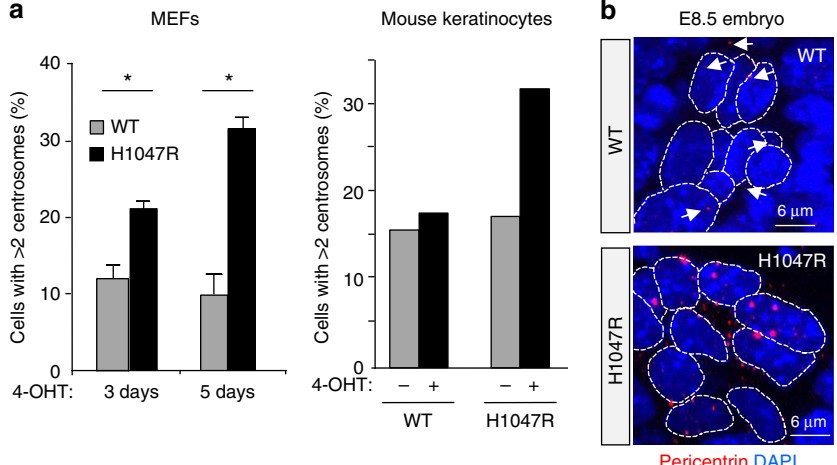

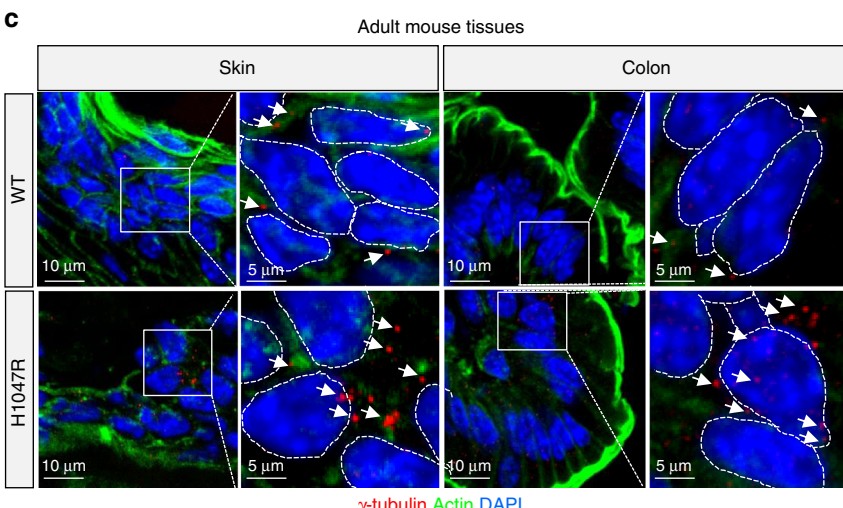

**Fig. 2** p110α^H1047R expression leads to centrosome amplification in cells and tissues. **a** Frequency of cells with centrosome amplification ($n > 2$) in WT and *Pik3ca*^H1047R primary E13.5 MEFs (72 h after addition of 4-OHT or vehicle; 200 cells were scored per genotype and per time point; values = mean ± SD) and mouse keratinocytes (isolated from 8-week-old *Pik3ca*^WT;*Flpe-ER*^T2 or *Pik3ca*^H1047R+neo;*Flpe-ER*^T2 mice, cultured for 7 days of which the last 48 h were in the presence of 4-OHT or vehicle; 400 cells were scored per genotype and per condition). Statistically significant differences are indicated by *($P <$ 0.05), as determined by non-parametric Mann–Whitney *t* test (one-tailed). **b** Whole-mount of E8.5 embryos stained for pericentrin. Dashed lines contour single-cell nuclei. White arrows point towards individual centrosomes in the WT cells. Mutant embryos show enlarged and amplified number of centrosomes per cell. **c** Cryosections of skin and colon of 8-week-old *Pik3ca*^WT;*Flpe-ER*^T2 and *Pik3ca*^H1047R+neo;*Flpe-ER*^T2 mice treated for 5 consecutive days with tamoxifen (80 mg kg^−1 by gavage), stained for γ-tubulin (red) and actin (green). White arrows point towards centrosomes. Dashed lines contour single-cell nuclei. White boxes surround areas that are magnified underneath

and led to PI3K pathway activation (Fig. 1c). Enhanced Akt phosphorylation was also observed in primary fibroblasts from human fibro-adipose overgrowth syndrome patients with mosaic, heterozygous expression of the *PIK3CA*^H1047L mutation[21] (Supplementary Fig. 1c).

**Organismal impact of heterozygous p110α^H1047R expression**. Heterozygous expression of p110α^H1047R induced by constitutive Flp expression from day 0 *post coitum*[19] resulted in mutant embryos which developed normally until embryonic day (E) 8.5 (Supplementary Fig. 2a). At E9.5, however, p110α^H1047R embryos had not grown in size compared to E8.5 and showed increased PI3K/Akt pathway activation, with a dramatic collapse in overall tissue integrity, a lack of vascular remodelling and absence of hierarchical organization of vessels, widespread apoptosis and increased levels of p53 (Supplementary Fig. 2a–e). No live embryos were recovered at E10.5. These observations are similar to those of a previous report on embryonic p110α^H1047R expression using a Cre-inducible *Pik3ca*^H1047R mouse line[22].

Tamoxifen-mediated Flp activation[20] in adult mice at 8 weeks of age, led to efficient recombination of the *Pik3ca*^H1047R locus (Supplementary Fig. 3a) and PI3K pathway activation in all tissues investigated (Supplementary Fig. 3b). These mice appeared to be in good health, but over time died suddenly, for reasons that are unclear, or had to be humanely terminated either because of displaying sudden ataxia and seizures or becoming acutely moribund without prior or accompanying disease symptoms. Most mice died within one year of age (median survival time 220 days post recombination of the mutated allele; Supplementary Fig. 3c). At the time of death, there was evidence for a weight increase in several organs, including the brain (Supplementary Fig. 3d). Importantly, *Pik3ca*^H1047R mice did not develop neoplastic lesions or cancer, as assessed by histopathology of multiple organs (Supplementary Tables 1 and 2).

Tamoxifen-induced expression of *Pik3ca*^H1047R together with intestine-specific heterozygous deletion of the *Apc* tumour suppressor gene (*Apc*^flox/+ mice) accelerated colon cancer progression (median survival time of 290 and 134 days in

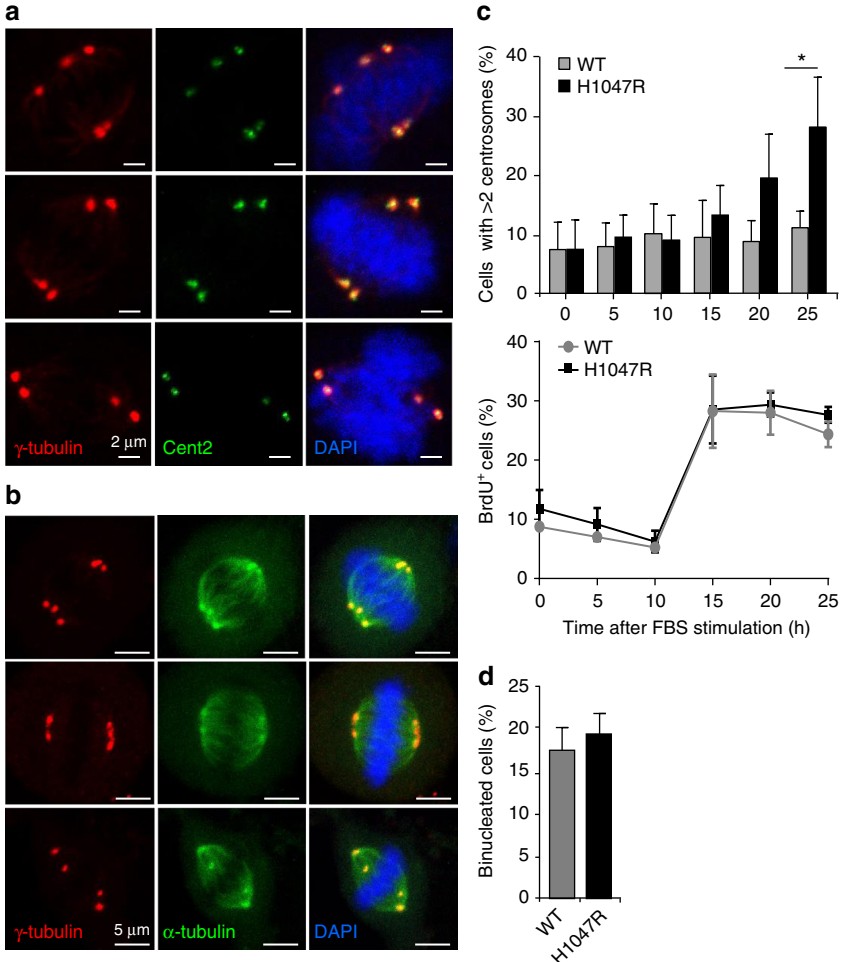

**Fig. 3** p110α^H1047R expression in MEFs leads to centrosome overduplication. **a** Representative immunofluorescence images showing extra centrosomes in p110α^H1047R MEF cells composed of two centrioles. **b** Representative immunofluorescence images showing extra centrosomes contributing to the mitotic spindle in p110α^H1047R MEF cells. **c** Centrosome number during the first cell cycle upon induction of p110α^H1047R expression (top panel; 200 cells per genotype and time point were scored for the centrosome analysis; values = mean ± SD) and parallel analysis of DNA synthesis (assessed by BrdU incorporation; bottom panel: 10,000 cells were acquired from two independent WT and two independent p110α^H1047R MEF lines. Statistically significant differences are indicated by *(P < 0.05), as determined by non-parametric Mann–Whitney t test (one-tailed). MEFs were plated and cultured overnight in serum-containing medium. The next morning, cells were serum-starved in the presence of 4-OHT for 48 h, followed by re-addition of 10% fetal bovine serum and sampling of cells at the indicated time points. **d** Percentage of binucleated MEFs in cultures treated 2 days with 4-OHT. Binucleation was assessed by IF using DAPI to stain DNA. A total of 200 cells were scored per genotype in 3 Pik3ca^WT;Flpe-ER^T2 and 3 Pik3ca^H1047R+neo;Flpe-ER^T2 independent MEF lines (values = mean ± SD)

Apc^flox/+ and Pik3ca^H1047R;Apc^flox/+ mice, respectively; Supplementary Fig. 4a–c).

In summary, these observations corroborate that heterozygously expressed mutant Pik3ca can have a major impact on the animal, both in adult life and during embryonic development. Our results also reinforce the concept that mutant Pik3ca is not efficient at initiating tumour formation on its own, but cooperates with other tumour-promoting genetic lesions[9,23–25].

**p110α^H1047R expression leads to centrosome amplification.** We next sought to understand the early cellular impact of endogenous p110α^H1047R expression, using primary MEFs as the main model. Pik3ca^H1047R induction in these cells led to Akt activation (Fig. 1c), an increase in cell number, loss of contact inhibition and a low level of colony formation, without obvious changes in cell death (Supplementary Fig. 5a–d).

Further characterization of p110α^H1047R MEFs revealed that these cells frequently displayed extra centrosomes (Fig. 2a).

Centrosomes are the main microtubule-organizing centre in animal cells and critical for the formation of the mitotic spindle. Immediately after cell division, each cell contains one centrosome which is then duplicated, in a cell cycle-dependent manner and in parallel with DNA replication[26], in preparation for the next round of cell division. Centrosomes are composed of two centrioles surrounded by pericentriolar matrix which can be detected by antibodies against pericentrin or γ-tubulin. p110α^H1047R expression in MEFs led to a significant increase in the number of cells with supernumerary centrosomes, with >30% of the cells showing more than two centrosomes 5 days after treatment with 4-OHT (Fig. 2a and Supplementary Fig. 6a and experimental controls shown in Supplementary Fig. 6b).

Supernumerary centrosomes were also more frequent in fibroblasts from a PIK3CA^H1047L-driven human overgrowth syndrome patient[21] (Supplementary Fig. 6c) and in MCF-10A human mammary epithelial cells transiently transfected with p110α^H1047R (Supplementary Fig. 6d). In MCF-10A cells, other cancer-associated modes of PI3K pathway activation, namely

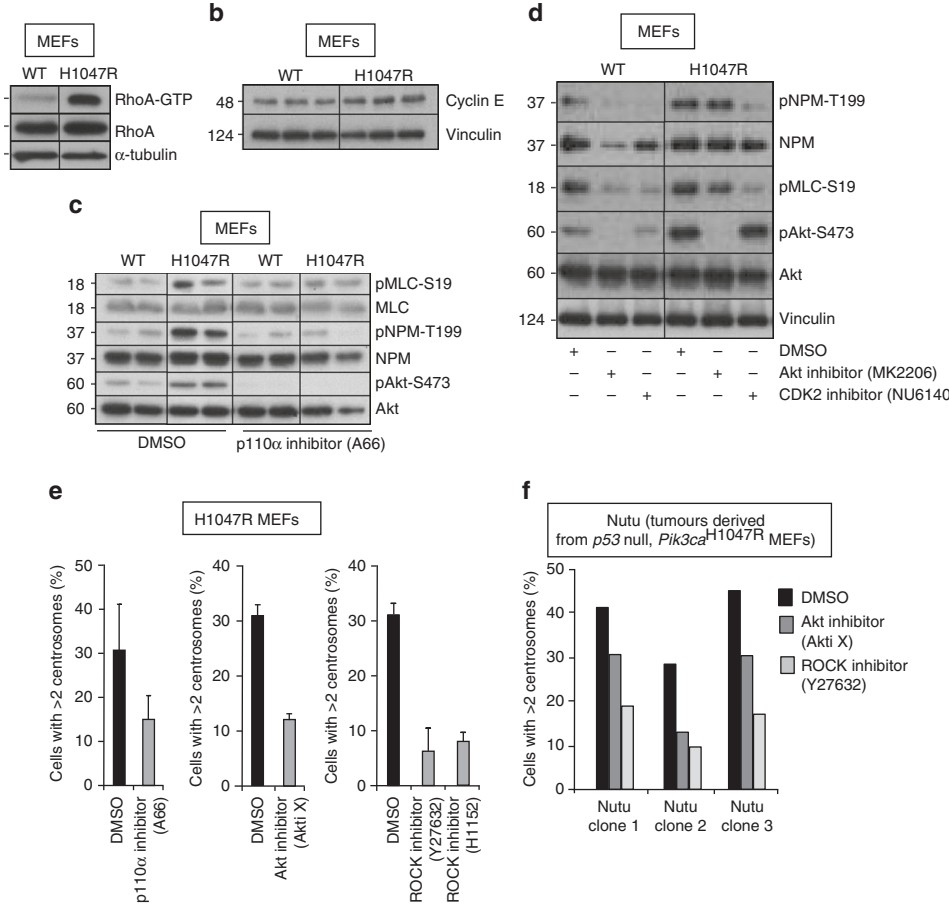

**Fig. 4** p110α^H1047R expression leads to centrosome overduplication through RhoA and ROCK pathway activation. **a** Assessment of RhoA activity in *Pik3ca*^WT;*Flpe-ER*^T2 and *Pik3ca*^H1047R+neo;*Flpe-ER*^T2 MEFs, 3 days after treatment with 4-OHT. The experiment was performed three times using three independent *Pik3ca*^WT;*Flpe-ER*^T2 and *Pik3ca*^H1047R+neo;*Flpe-ER*^T2 MEFs. A representative experiment is shown. **b** Cyclin E protein levels in *Pik3ca*^WT;*Flpe-ER*^T2 and *Pik3ca*^H1047R+neo;*Flpe-ER*^T2 MEFs, 3 days after treatment with 4-OHT. Cell extracts from three independent MEF lines per genotype are shown. **c** Activation of signalling pathways involved in centrosome duplication by p110α^H1047R expression, and effect of p110α inhibition (by 24 h of treatment with 3 μM A66). Cell extracts from two independent MEF lines per genotype are shown. **d** Effect of inhibition of CDK2 (by 2 h of treatment with 1 μM NU6140) or Akt (by 2 h of treatment with MK2206) on signalling pathways involved in centrosome duplication. **e** Impact of inhibition of p110α (by 3 μM A66), Akt (by 1 μM Akti X) or ROCK (by 10 μM Y27632 or 0.5 μM H1152) on p110α^H1047R-induced centrosome amplification in primary MEFs. A total of 100 cells were scored per condition using two independent p110α^H1047R MEF lines; values = mean ± SD. **f** Impact of inhibition of Akt (by 1 μM Akti X) or ROCK (by 10 μM Y27632) for 15 days on centrosome numbers in p110α^H1047R Nutu cells. Centrosomes were revealed by γ-tubulin staining. In total, 200 cells were scored per cell line and per condition

expression of the helical E545K mutant of *PIK3CA*, the D560Y mutant of *PIK3R1* (the p85 regulatory subunit of *PIK3CA*) as well as overexpression of *PIK3CA*^WT (reflecting *PIK3CA* amplification in cancer), all displayed more centrosomes than parental cells (Supplementary Fig. 6d).

Interestingly, evidence for in situ centrosome amplification was also observed in E8.5 p110α^H1047R embryos (Fig. 2b) and in adult skin and colon tissue, 2 weeks after the induction of p110α^H1047R (Fig. 2c). In line with this, keratinocytes explanted from adult mice, following a 2-week in vivo induction of p110α^H1047R, also showed extra centrosomes (Fig. 2a and Supplementary Fig. 6e).

**p110α^H1047R expression leads to centrosome overduplication**. Compared to WT cells, p110α^H1047R MEFs did not show any obvious increase in the number of senescent cells (Supplementary Fig. 7a), DNA damage (Supplementary Fig. 7b, c) or alterations in cell cycle profiles (i.e. prolonged G1/S or G2/M; Supplementary Fig. 7d), all of which are known causes of centrosome number deregulation[11,27].

Immunofluorescence analysis revealed that the extra centrosomes seen in p110α^H1047R MEFs were positive for exogenously expressed Cent2-GFP and composed of two centrioles, demonstrating that p110α^H1047R induction did not lead to pericentriolar matrix fragmentation or premature centriole splitting (representative images shown in Fig. 3a and Supplementary Fig. 6f). In addition, the extra centrosomes were found to co-localize with mitotic spindle microtubules (representative images shown in Fig. 3b), and to always contribute to the establishment of the mitotic spindle.

Alternative mechanisms for generating extra centrosomes include cytokinesis failure or multiple cycles of centrosome duplication prior to cell division[11]. After induction of p110α^H1047R in MEFs that were growth-arrested by serum-deprivation, centrosome amplification emerged in parallel with the initiation of DNA synthesis (as measured by BrdU incorporation) in the G1/S transition, and before the cells had gone through cytokinesis (Fig. 3c). These data suggest that early after induction of p110α^H1047R, centrosome amplification occurs during the first round of cell division, before the cells have

divided. Additionally, we observed that p110α[H1047R] expression did not increase cytokinesis failure (as measured by % of binucleated cells) in these cells 2 days after 4-OHT treatment (Fig. 3d). These observations suggest that centriole overduplication, but not cytokinesis failure, is the underlying mechanism leading to centrosome amplification during the early stages of induction of p110α[H1047R]. This is further supported by the observations from signalling studies, as described below.

**Centrosome overduplication by p110α[H1047R] via Akt and ROCK.** We found that p110α[H1047R] MEFs displayed higher basal levels of RhoA–GTP, a known activator of ROCK (Fig. 4a). Expression of p110α[H1047R] was also found to lead to increased levels of Cyclin E (Fig. 4b). CDK2-Cyclin E complexes are known to phosphorylate the Nucleophosmin (Npm)/B23 protein in S-phase, which then interacts with ROCKII allowing its activation and initiation of centrosome duplication[28,29]. In line with our conclusion that p110α[H1047R] expression leads to centrosome overduplication in S-phase, the activation of ROCK (as measured by the phosphorylation of MLC on Ser19) and the phosphorylation of Npm on T199 (a phosphorylation site of CDK2)[30] were enhanced in p110α mutant cells (Fig. 4c). Both these signalling events were inhibited either by a CDK2 inhibitor (Fig. 4d), and at longer time points, by a p110α-selective inhibitor (Fig. 4c) or an Akt inhibitor (Fig. 4d and Supplementary Fig. 8a). The precise interplay and timing of these signalling events remains to be determined. Our data are consistent with p110α[H1047R] signalling to the centrosome at the time of initiation of duplication at the G1/S transition, through a mechanism involving AKT, ROCKII and CDK2/Cyclin E-Nucleophosmin, most likely leading to enhanced activation of the centriole duplication machinery.

We next tested the impact of pharmacological intervention with PI3K/Akt or RhoA/ROCK signalling on p110α[H1047R]-induced centrosomal abnormalities, either before their genesis in primary cells or once they had been established in a tumour context.

Exposing primary MEFs to inhibitors of p110α, Akt or ROCK during the induction phase of p110α[H1047R] expression prevented centrosome amplification (Fig. 4e), with no significant impact on cell proliferation (Supplementary Fig. 8b). A similar experiment to the one shown in Fig. 3c but in the presence of a ROCK inhibitor, revealed that blocking ROCK activity prevented centrosome amplification before cells had gone through cytokinesis (Supplementary Fig. 9a), suggesting that centrosome amplification occurs during the first round of cell division upon p110α[H1047R] induction, in a ROCK-dependent manner.

We next created tumour-derived cell lines, by immortalizing primary p110α[H1047R] MEFs in vitro using p53 inactivation and inoculating them in nude mice. Once the injected cells had given rise to tumours, cell lines derived thereof were established in cell culture (further referred to as Nutu cells). A high fraction of Nutu cells had centrosome amplification (Fig. 4f). Similar to results in primary MEFs, treatment of Nutu cells with inhibitors of Akt and especially of ROCK reduced the fraction of cells with centrosome amplification (Fig. 4f), underscoring the importance of Akt and ROCK signalling in centrosome amplification. These data indicate that *Pik3ca*[H1047R]-induced centrosomal abnormalities, once established, can still be partially reverted pharmacologically. The overall importance of ROCK in the biology of oncogenic *PIK3CA* is illustrated by the observation that treatment with two different ROCK inhibitors (Y27632 and H1152) clearly decreases *Pik3ca*[H1047R]-induced cell transformation of primary MEFs (Supplementary Fig. 9b).

**Efficient centrosome clustering upon p110α[H1047R] expression.** Centrosome amplification can lead to the formation of multipolar spindles during mitosis[31–33] and therefore compromise cell fitness, as the progeny derived from cells that do not cluster the extra centrosomes can die or undergo cell cycle arrest after undergoing a multipolar division[33]. Since an enhanced level of cell death was not observed in p110α[H1047R] MEFs (Supplementary Fig. 5d), we speculated that these cells clustered their extra centrosomes efficiently. Indeed, live imaging by time-lapse microscopy of MEFs expressing centrin2 (a component of the centriole) tagged with GFP revealed that, of the cells with multiple centrosomes (30% in p110α[H1047R] MEFs vs. 10% in WT cells; Fig. 2a), most of the p110α[H1047R] cells were able to exit mitosis efficiently, with only 6% of mutant cells failing to do so, compared to a 35% failure rate in WT cells (n = 152 and 161 mitoses counted for WT and p110α[H1047R] MEFs, respectively; Fig. 5a and Supplementary Fig. 10 and Supplementary Movies 1, 2 and 3). These data show that, compared to WT cells, cells expressing p110α[H1047R] are more proficient at completing mitosis with extra centrosomes.

**Lack of detectable p110α[H1047R]-induced chromosome segregation errors.** It has previously been shown that cells with extra centrosomes acquire chromosome segregation errors after passing through a transient 'multipolar spindle intermediate' in which merotelic kinetochore microtubule attachment errors accumulate before centrosome clustering and anaphase[33]. The assessment of chromosome segregation errors during anaphase did not reveal differences in the frequency of segregation errors (~15% in each genotype; including acentric and centric chromosomes, and anaphase bridges) in MEFs, 72 h after treatment with 4-OHT (Fig. 5b). To better understand the observed lack of segregation errors upon p110α[H1047R] induction, we performed a more detailed analysis of the different spindle configurations observed in mitotic cells. We focused on cells with centrosome amplification going through mitosis (note that we observed ~10% centrosome amplification in WT and ~30% in p110α[H1047R] populations). Analysis of mitotic spindles in prometaphase and metaphase in these cells revealed a lower incidence of multipolar configurations in p110α[H1047R] than in WT cells (Fig. 5c). This suggests that even at earlier stages of mitosis, when erroneous kinetochore microtubule attachments are established (merotelic attachments), there is already a difference in the clustering efficiency of extra centrosomes between WT and p110α[H1047R]. The total number of the multipolar spindles in early mitosis were slightly higher in the mutant population (~8 WT cells with multipolar spindles vs. ~14 p110α[H1047R] cells with multipolar spindles in every 100 cells), yet we did not observe higher frequencies of single chromosome segregation errors in the mutant cells.

Altogether these findings point toward a higher clustering efficiency of extra centrosomes in p110α[H1047R]-expressing cells and a lower number of multipolar intermediates (see Discussion for further details).

**p110α[H1047R] expression allows tetraploid cell propagation.** Analysis of metaphase chromosome spreads revealed an increase in aneuploid cells in p110α[H1047R] MEFs populations (Fig. 6a). Indeed, a range of chromosome number alterations was observed 3 and 5 days after p110α[H1047R] induction, when cells had undergone multiple cell divisions (average doubling time (DT) 26.2 h and 20.9 h for WT and p110α[H1047R] MEFs, respectively) (Fig. 6b; Supplementary Fig. 11 shows the individual experiments performed 3 and 5 days after 4-OHT treatment; experimental controls are shown in Supplementary Fig. 12—an independent experiment in which MEFs were subjected to parallel FACS and metaphase spread analysis is shown in Supplementary Fig. 13).

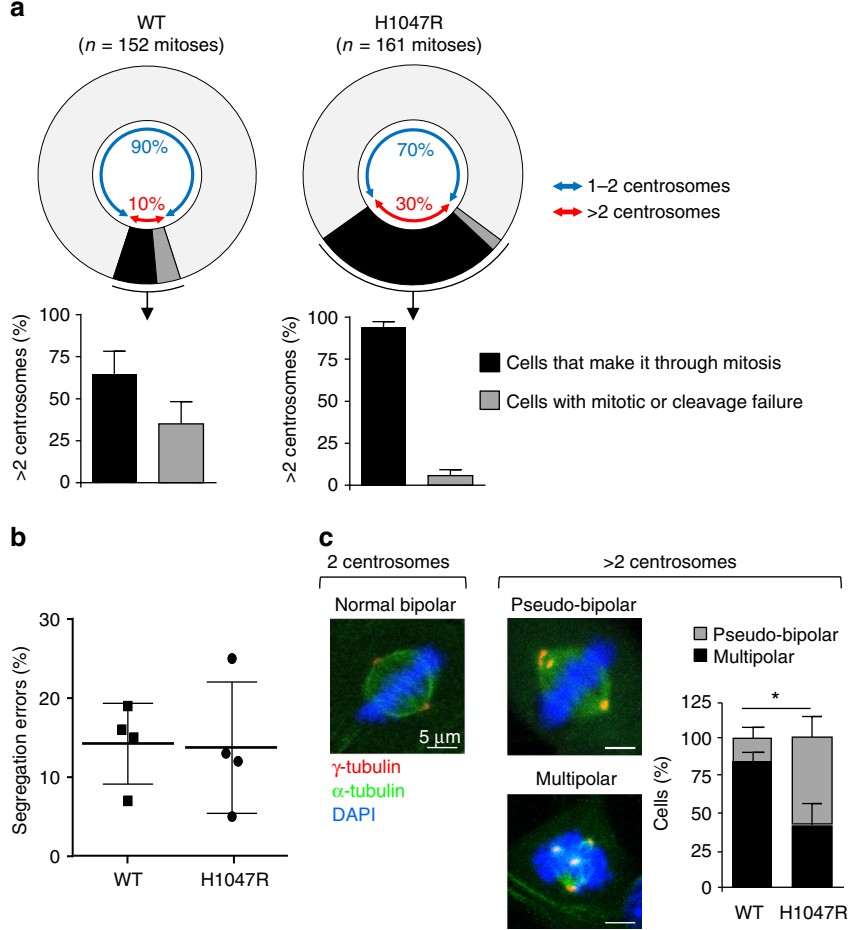

**Fig. 5** Efficient clustering of extra centrosomes and lack of chromosomes segregation errors early upon p110α[H1047R] expression. **a** Quantification of the number of cells with centrosome amplification that either make it through mitosis or experience mitotic slippage or cytokinesis failure. In total, 152 WT cells (50, 48 and 54 cells from 3 independent WT MEFs) and 161 Pik3ca[H1047R] cells (50, 61 and 50 cells from 3 independent Pik3ca[H1047R] MEFs) with centrosome amplification entering mitosis were analysed; values = mean ± SD. **b** Frequency of segregation errors occurring in anaphase of Pik3ca[WT];Flpe-ER[T2] and Pik3ca[H1047R+neo];Flpe-ER[T2] MEFs, 3 days after treatment with 4-OHT. A total of 103 WT cells from 3 independent WT MEFs (33, 32 and 38); and 101 Pik3ca[H1047R] cells (38, 39 and 24 from 3 independent Pik3ca[H1047R] MEFs) in anaphase were analyzed for the presence of chromosome segregation errors. Individual values are plotted. **c** Representative immunofluorescence images of mitotic spindle conformations observed in Pik3ca[WT];Flpe-ER[T2] and Pik3ca[H1047R+neo];Flpe-ER[T2] MEFs, 3 days after treatment with 4-OHT and quantification of the incidence of pseudo-bipolar and multipolar spindles in metaphases with > 2 centrosomes. 74 WT cells (29, 30 and 15 cells from three independent WT MEFs) and 83 Pik3ca[H1047R] cells (30, 27 and 26 cells from 3 independent Pik3ca[H1047R] MEFs) with centrosome amplification in prometaphase/metaphase were analyzed (values = mean ± SD). Statistically significant differences are indicated by *(P < 0.05), as determined by non-parametric Mann–Whitney t test (one-tailed)

Aneuploidy in p110α[H1047R] MEFs was characterized by a preponderance of cells with a near-tetraploid chromosome number (around 80 chromosomes) (Supplementary Figs. 11 and 13). The observation that a large fraction of aneuploidy cells have a near-tetraploid content with very few cells with 41–59 chromosomes suggests that these cells have become aneuploid following a genome doubling event (tetraploidization). Keratinocytes explanted from Pik3ca[H1047R] mice also showed this tendency (Fig. 6c). Given the lack of cytokinesis failure early upon induction of p110α[H1047R] in MEFs (Fig. 3d), these observations indicate that p110α[H1047R] expression facilitates the propagation of tetraploid cells that arise stochastically. This could be explained by an increased tolerance to tetraploidization[34], reflected by the fact that cells did not arrest as G1 tetraploids (4 N) but reached 8 N DNA content. Alternatively or in addition, the greater centrosome clustering efficiency observed earlier (Fig. 5a) would enable newly formed tetraploid cells to avoid catastrophic multipolar divisions.

In order to test whether p110α[H1047R] expression could provide tolerance to tetraploidization, primary WT and p110α[H1047R] MEFs were treated with dihydrocytochalasin B (DCB), an agent that blocks cytokinesis, to experimentally increase the fraction of tetraploid cells in the cultures. Cells were filmed for 30 h after DCB washout to assess cell division of binucleated cells resulting from cytokinesis failure. Strikingly, the number of p110α[H1047R] cells dividing after DCB washout was much higher than in WT cells (60% vs. 20%, respectively; Fig. 6d and Supplementary Movie 4), indicating that p110α[H1047R] cells have a less strict G1 tetraploidy checkpoint than WT cells, and display a higher tolerance to tetraploidy.

Inactivation of p53 is known to be an important mechanism leading to tolerance to genome doubling[35,36]. Both WT and p110α[H1047R] MEFs showed a proliferative arrest upon doxorubicin treatment (Supplementary Fig. 14a), with concomitant upregulation of p53 (Supplementary Fig. 14b), indicative of a functional p53 response to DNA damage. This suggests

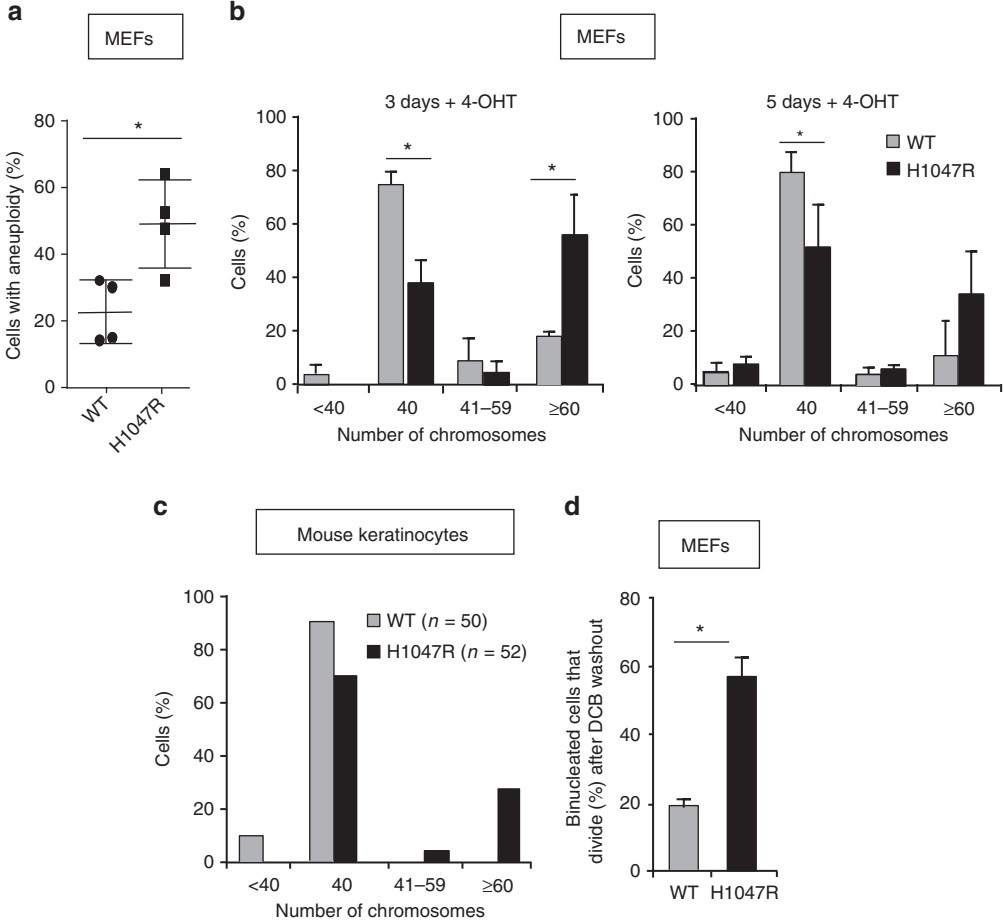

**Fig. 6** p110α^H1047R expression provides cellular tolerance to genome doubling. **a** Frequency of aneuploid cells (i.e. with a number of chromosomes that differs from 40), assessed by metaphase chromosome spreads, in primary MEFs 5 days after treatment with 4-OHT. Statistically significant differences are indicated by *(P < 0.05), as determined by non-parametric Mann–Whitney t test (two-tailed). Five independent Pik3ca^WT;Flpe-ER^T2 and 5 independent Pik3ca^H1047R+neo;Flpe-ER^T2 MEFs were analyzed, and ~50 metaphase spreads counted per MEF line. **b** Frequency of tetraploid cells in Pik3ca^WT;Flpe-ER^T2 and Pik3ca^H1047R+neo;Flpe-ER^T2 MEFs, 3 and 5 days after treatment with 4-OHT. Cells were considered to be tetraploid (or tetraploid-derived) if the number of chromosomes per cell was ≥ 70. Three independent Pik3ca^WT;Flpe-ER^T2 and 3 independent Pik3ca^H1047R+neo;Flpe-ER^T2 MEFs were analyzed, and ~50 metaphase spreads counted per MEF line and per time point (values = mean ± SD). Statistically significant differences are indicated by *(P < 0.05), as determined by the non-parametric Mann–Whitney t test. **c** Analysis of chromosome numbers in mouse keratinocytes 2 days after addition of 4-OHT. One Pik3ca^WT;Flpe-ER^T2 and one Pik3ca^H1047R+neo;Flpe-ER^T2 keratinocyte culture were analyzed. **d** Frequency of binucleated MEFs dividing within a period of 30 h after DCB-induced tetraploidization, assessed 3 days after treatment with 4-OHT. Three independent Pik3ca^WT;Flpe-ER^T2 and three independent Pik3ca^H1047R+neo;Flpe-ER^T2 MEFs were analyzed, and 60–80 binucleated cells were tracked per MEF line (values = mean ± SD). Statistically significant differences are indicated by *(P < 0.05), as determined by the non-parametric Mann–Whitney t test (one-tailed)

alternative tolerance routes to genome doubling in the presence of p110α^H1047R.

**PIK3CA mutations and genome doubling in human cancer.** Emerging evidence suggests that PIK3CA mutations can occur late in the evolution of some cancers, being present in only a subset of cancer cells within the tumour[37,38]. However, in breast cancers (see Methods for breast cancer subtypes analyzed), mutations in PIK3CA showed a significant tendency to be clonal, i.e. to be present in all tumour cells (Fig. 7a; P < 0.001), indicative of PIK3CA mutation being an early event in the evolution of this tumour type. In genome-doubled breast cancers, the majority of PIK3CA mutations were found to precede the genome duplication event (Fig. 7b), with a frequency significantly higher than would be expected by chance (P < 0.0001).

PIK3CA mutations, which are particularly enriched in estrogen receptor (ER)-positive and luminal subtype cancers (Supplementary Fig. 15a, b), showed a tendency to be mutually exclusive with

mutations in TP53 (Supplementary Fig. 15c; P < 0.0001, Fisher's exact test), which occur more frequently in basal-like tumours. In contrast to mutations in TP53, a known tolerance mechanism towards genome doubling[35–37], we did not observe an enrichment of PIK3CA mutations in genome-doubled breast tumours. All together, these data are consistent with a role of PIK3CA mutation as a potential tolerance mechanism for genome doubling in breast cancer, conceivably independent of the p53 pathway, and also suggest that additional genomic events may be required to facilitate such tolerance in TP53 WT tumours. It remains to be demonstrated that PIK3CA mutation can provide such activities in vivo.

The significant enrichment for clonal mutations and the significant tendency for mutations in PIK3CA to precede genome doubling remained significant when focusing on ER-positive breast cancers or luminal breast cancers (Supplementary Fig. 16a, b). However, in other subtypes, there were insufficient numbers of mutations in PIK3CA to reliably assess clonality and timing.

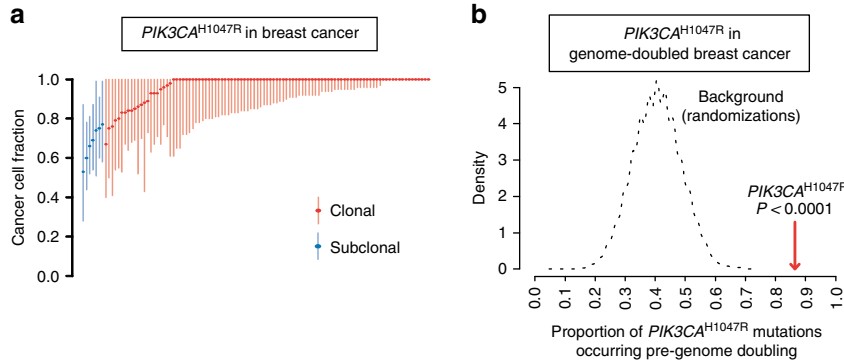

**Fig. 7** The cancer cell fraction and timing of PIK3CA^H1047R mutations in human breast cancer. **a** The cancer cell fraction of PIK3CA^H1047R mutations in the TCGA breast cancer cohort. Each symbol represents a non-silent somatic mutation in an individual tumour. On the basis of the probability distributions of the cancer cell fraction, mutations were determined to be either clonal (red circles, upper bound of confidence interval ≥ 1) or subclonal (blue circles, upper band of confidence interval < 1). Error bars represent the 95% confidence interval. PIK3CA^H1047R mutations are significantly more often clonal than subclonal ($P < 0.001$). **b** Timing of PIK3CA^H1047R mutations in the TCGA genome-doubled breast cancer cohort. The proportion of PIK3CA^H1047R mutations that likely occurred prior to genome doubling is depicted with a dotted line representing expected distribution based on background randomizations. The graph shows that the majority of PIK3CA^H1047R mutations likely precede genome doubling, more than expected by chance ($P < 0.0001$). The proportion of pre-genome doubling is the proportion of PIK3CA^H1047R mutations that occur pre-genome doubling. The distribution relates to the expected number of mutations occurring pre-genome doubling. This is based on taking the same number of mutations, and randomly sampling this 10,000 times. This gives an estimate of the likelihood that a random sample of mutations would show the same number of pre-doubled mutations

**Impact of PI3K pathway blockade on tetraploidization.** Treatment with inhibitors of p110α, Akt or ROCK during the induction phase of p110α^H1047R expression in primary MEFs prevented tetraploidization (shown for p110α and ROCK inhibitors in Fig. 8a).

We next tested the impact of these inhibitors on spontaneously in vitro transformed p110α^H1047R MEFs that differ in the percentage of diploid/tetraploid cells in the cell population (Fig. 8b; showing 2 clones—top and bottom graphs) and on Nutu cells, which are near-tetraploid (Fig. 8c (showing clone 1) and Supplementary Fig. 17 (showing clone 2)). In none of these chromosomal configurations, could these agents rescue the normal ploidy (Fig. 8b, c; Supplementary Fig. 17). This is in contrast to what was observed for centrosome amplification, which could be normalized pharmacologically (Fig. 4e, f).

Moreover, stably transformed diploid and tetraploid p110α^H1047R MEFs established after long-term culture, were found to be equally sensitive to inhibition of p110α with A66. (Supplementary Fig. 18 a, b).

Taken together, these data indicate that Pik3ca^H1047R-induced chromosomal abnormalities, once established, cannot be reverted pharmacologically, with p110α^H1047R expression providing tolerance, in a ROCK-dependent manner, to the early stages of tetraploidization but, once this genomic state is stably established, does not provide enhanced sensitivity to PI3K pathway inhibition.

## Discussion

In this study, we report on previously unappreciated biological consequences of expression of mutated p110α, namely centrosome amplification, tolerance to tetraploidization and induction of aneuploidy (Fig. 9). At present, the functional connection between these biological phenomena remains unclear but the observation that each of these responses can be blocked by inhibition of the ROCK kinase strongly suggests a functional interrelationship downstream of signalling by oncogenic p110α PI3K.

We show here that p110α^H1047R expression leads to supernumerary centrosomes in a variety of primary mouse and human cells, as well as in embryonic and adult mouse tissues. Indirect evidence for a role of PI3K in centrosome biology was previously found in human HeLa and HCT116 cancer cells, in which stable transfection of an oncogenic Met tyrosine kinase receptor was documented to induce centrosome amplification in a PI3K-dependent manner[39]. Our data in MEFs suggest that the underlying mechanism of p110α^H1047R-induced centrosome amplification early upon Pik3ca^H1047R expression is centrosome overduplication, at least partially mediated through the activation of RhoA/ROCK signalling, which has previously been linked to centrosome duplication[28]. Importantly, it had been previously unclear how signalling pathways affect these regulators of centrosome biology. Our manuscript shows, for the first time, that the RhoA/ROCK pathway is involved in centrosome amplification downstream of PI3K/AKT. Additionally, our phosphoproteomic analysis of MEFs also identified a large fraction of phosphopeptides, differentially regulated by p110α^H1047R, as proteins involved in the regulation of microtubule dynamics and centrosome biology[18].

Recent data indicate that centrosome amplification on its own is not a clear driver of cancer development but can cooperate with p53 loss to accelerate cancer development[15,16]. Similar observations were made with our mutant mice, where Pik3ca^H1047R expression did not lead to spontaneous cancer within the time frame analyzed, but accelerated the onset of cancer upon concomitant heterozygous loss of the Apc tumour suppressor gene; as shown previously[25], or in the presence of other oncogenic lesions, such as loss of p53 or PTEN[9,40,41]. However, the specific contribution of Pik3ca^H1047R-driven centrosome amplification in interaction of mutant PIK3CA with other cancer-promoting genetic lesions remains unknown.

Centrosome amplification can give rise to chromosome segregation errors, as a result of multipolar spindle formation during the clustering process, resulting in aneuploidy[33,42]. Despite the presence of centrosome amplification and increased centrosome clustering efficiency, the frequency of chromosome segregation errors appeared to be unaltered early after p110α^H1047R expression in MEFs and less multipolar spindles were observed. Given the observation that all extra centrosomes contribute to the formation of the mitotic spindle (Fig. 3b), we speculate that p110α^H1047R expression allows extra centrosome clustering without the formation of multipolar spindles and/or enables cells

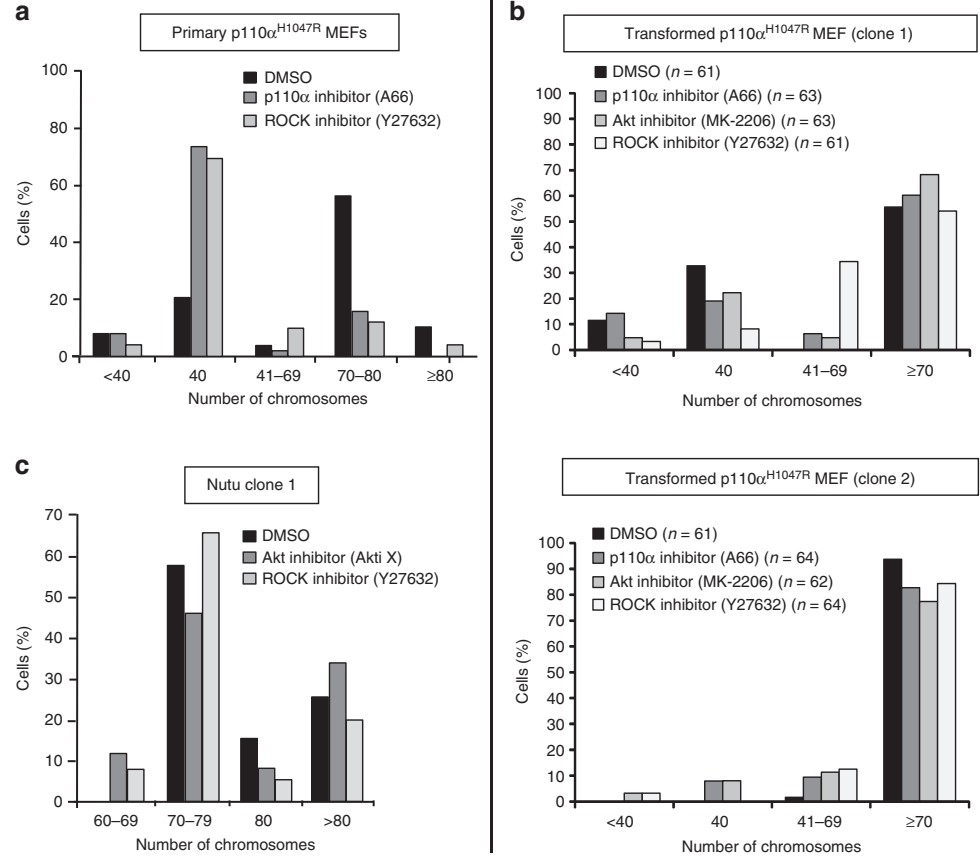

**Fig. 8** Impact of pharmacological blockade of PI3K pathway components on tetraploidization in primary and tumour-derived MEFs. **a** Impact of inhibition of p110α (by 3 μM A66) or ROCK (by 10 μM Y27632) on p110α[H1047R]-induced chromosomal abnormalities in primary MEFs. One Pik3ca[H1047R+neo];Flpe-ER[T2] MEF line was analyzed and at least 50 metaphase spreads were counted per condition. Inhibitors were added together with the 4-OHT to induce p110α[H1047R] expression. **b** Impact of 14 day inhibition of p110α (by 3 μM A66), Akt (by 1 μM MK-2206) or ROCK (by 10 μM Y27632) on chromosome numbers of two independent stably transformed p110α[H1047R] MEF clones with a different degree of tetraploidy. **c** Impact of inhibition of Akt (by 1 μM Akti X) or ROCK (by 10 μM Y27632) for 21 days on chromosome numbers of p110α[H1047R] Nutu clone 1 (shRNA p53-immortalized p110α[H1047R] MEF clone passed through nude mice as tumour). Chromosome counting was performed on DAPI-stained metaphase spreads. Fifty spreads were counted per line and condition, unless otherwise stated

to correct merotelic attachments more efficiently. It is also possible that the number of cells with centrosome amplification/multipolar spindles that potentially missegregate chromosomes in our MEF cell model is too low to observe differences in chromosome missegregation between WT and Pik3ca[H1047R] cells. In this regard, it is informative to consider the study by Ganem et al.[33]. Indeed, these authors found that in RPE-1 cells treated with DCB, in which 70% of the cells in the population have extra centrosomes and multipolar spindles, only 12% of those (i.e. the equivalent of ~8.4 cells out of 70) missegregate a chromosome. In our MEF study, we find 30% of centrosome amplification in p110α[H1047R] MEFs. Assuming that all the MEF cells with centrosome amplification go through mitosis, and assuming a similar situation to the Ganem et al. study[33], we would expect that only 3.6 cells out of 100 (i.e. 12% of the equivalent of 30 cells with centrosome amplification out of 100) would missegregate a chromosome.

It is important to stress that our study focused on assessing the cellular impact of Pik3ca[H1047R] activation in a non-transformed context. It is possible that deregulation of centrosome biology by oncogenic Pik3ca in a complex genetic background, as in cancer, might increase the frequency of chromosome segregation errors in cells. It is also possible that other cellular processes that are known to be regulated by centrosomes, such as cell polarity

invasion and metastasis (reviewed in refs. [12,43]), will be affected by Pik3ca[H1047R]-induced alterations in centrosome numbers, as these processes are known to be controlled by PI3K[44,45].

In addition to arising as a result of chromosome missegregation in diploid cells during mitosis, aneuploidy can also result when tetraploid cells, generated by whole-genome doubling from diploid cells, continue to proliferate. Tetraploid cells are genomically unstable and accumulate both numerical and structural chromosomal abnormalities, which can contribute to tumour heterogeneity and evolution[46,47]. Metaphase chromosome spread analysis of the mutant Pik3ca MEF population revealed a significant increase in the fraction of near-tetraploid cells, suggesting that the levels of aneuploidy observed in mutant cells most likely arise from unstable tetraploids, instead of diploid cells gaining or losing chromosomes. Tetraploidy can arise through various errors in cell division, such as cytokinesis failure or defects in mitosis, none of which were found to be affected by p110α[H1047R] early upon induction. MEFs are known to spontaneously become increasingly tetraploid at each passage in culture, through unknown mechanisms[48]. Our results therefore suggest that, rather than acting as an inducer of tetraploidization per se, p110α[H1047R] helps the cell to tolerate the presence of a doubled genome. Our data are consistent with the finding that hyperactivation of growth factor signalling, in our case by sustained

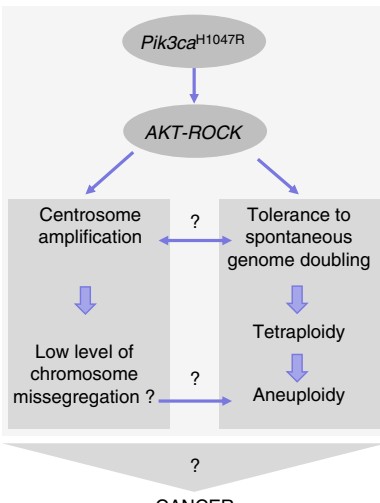

**Fig. 9** Summary of key observations on the impact of *PIK3CA* mutation on centrosome amplification and possible tolerance to genome doubling

signalling by the downstream effector p110α, can overcome the proliferative blockade of tetraploid cells[34]. More specifically, it is most likely that the pro-survival signalling induced by constitutive activation of Akt in mutant *Pik3ca* cells leads to increased resistance to the stress induced by tetraploidization. Our evidence suggests that mutant *Pik3ca* provides this protection through signalling pathways involving Akt and ROCK.

*Pik3ca*$^{H1047R}$-induced tolerance to tetraploidization likely impacts on both the observed supernumerary centrosomes and increased aneuploidy in *Pik3ca*$^{H1047R}$ MEFs ≥ 5 days after *Pik3ca*$^{H1047R}$ induction. Indeed, at this stage, the observed centrosome amplification might be the result of a combined phenotype: (1) p110α$^{H1047R}$ signalling leading to centrosome duplication, as observed early (48–72 h) upon *Pik3ca*$^{H1047R}$ induction (Figs. 2a and 3c), and (2) tolerance to tetraploidization, resulting in cells with multiple centrosomes. Likewise, unstable tetraploids might be prone to accumulate single chromosome segregation errors[14]. Further investigation will be required to clarify the connection between these phenomena.

Inactivation of p53 is considered to be an important mechanism to tolerate the presence of a doubled genome[35], with mutations in *TP53* being enriched in genome-doubled tumours[37]. Whereas *PIK3CA* mutations appear to occur late in the development of many cancers, we found *PIK3CA* mutation to be an early, clonal event in breast cancer. Analysis of genome-doubled breast cancer revealed an enrichment of mutations in *TP53* (ref. [37]), but not in *PIK3CA*. However, *PIK3CA* mutations in these cancers were found to generally precede the genome doubling event and to exhibit a significant tendency for mutual exclusivity with *TP53*. These data are consistent with a *PIK3CA* tolerance mechanism to genome doubling, acting independently of *TP53* inactivation. The lack of enrichment of *PIK3CA* mutations in genome-doubled tumours also suggests, not entirely surprisingly, a requirement for additional genomic events to facilitate such tolerance in *TP53* WT tumours. Other studies have also provided evidence for *PIK3CA* deregulation as an early event in tumour evolution occurring before whole-genome doubling in colorectal adenocarcinoma (*PIK3CA* mutation)[49] and lung squamous cell carcinoma (*PIK3CA* amplification)[50].

The evidence for a role of tetraploidization in human tumour evolution is becoming stronger. Indeed, a significant proportion of human tumours have undergone a whole-genome duplication event during their development[49,50]. Pharmacological intervention with PI3K in cancer is currently aimed at targeting cancer cell proliferation and survival. Our data show that pharmacological inhibition of the PI3K/ROCK pathway can reduce the fraction of cells that has centrosome amplification within a population, but cannot revert the tetraploidy-derived aneuploidy facilitated by the presence of *Pik3ca*$^{H1047R}$, once it has been established. This could help to explain the currently limited therapeutic impact of PI3K-targeted therapies in cancer and the observation that *PIK3CA* mutation is not, on its own, always a clear predictor of sensitivity to PI3K pathway inhibitors[51]. On the other hand, our findings also suggest that pharmacological intervention with PI3K signalling in cancer might reduce tumour heterogeneity and evolution by limiting the perpetuation of genomically unstable tetraploid cells. Genomic analyses of *PIK3CA*-mutant positive tumour samples from patients treated with PI3K pathway inhibitors might uncover evidence for such a new role for PI3K in a clinical context. The timing of intervention with PI3K pathway inhibitors in cancer might therefore be of critical importance for maximizing their therapeutic impact.

## Methods

**Reagents**. In case the concentration of antibody stocks was known, we used 1 μg of antibody per immunoprecipitate, 1–2.5 μg ml$^{-1}$ for western blotting and 1–5 μg ml$^{-1}$ for immunohistochemistry. When the concentration of commercial antibody stocks was unknown, we used 10 μl per immunoprecipitate, 1/1000–5000 for western blotting and 1/50–500 for immunohistochemistry. Antibodies used for Western blotting were from the following commercial providers (catalogue numbers are in brackets): Cell Signaling Technologies: Akt/PKB (9272), pAkt-T308 (9275), pAkt-S473 (9271), pNPM-T199 (3541), NPM (3542), pMLC-S19 (3675), MLC (3672), Cyclin E (4129), vinculin (4650), p53 (9282); Sigma: β-actin (A1978), α-tubulin (T6074) and α-actinin (A5044); BD Biosciences: p110α (P94520); Santa Cruz Biotechnology: RhoA (sc-418). Antibodies used for immunofluorescence were from the following providers: Abcam: γ-tubulin (ab11316), pericentrin (ab4448), β-tubulin (ab6046) and GFP (ab1218); Sigma: α-tubulin (T6074); Cell Signaling Technologies: cleaved-caspase-3 (9664); Merck Millipore: pHistone H2A.X-S139 (05–636); Santa Cruz Biotechnology: Endomucin (sc-65495); 53BP1 (sc-22760); Fitzgerald: human Crest (90C-CS1058). Rhodamine Phalloidin (R415, Thermo-Fisher Scientific) was used to detect F-actin. Secondary antibodies conjugated to fluorescein isothiocyanate (FITC) or Cy3 were from Jackson ImmunoResearch Labs, and secondary antibodies conjugated to AlexaFluor 594 (A11012) and AlexaFluor 647 (A21445) were from ThermoFisher Scientific. Other reagents were sourced as follows: from Selleck Chemicals: Y27632 (S1049) and MK-2206 (S1078); Merck: Akti-X (124020); Symansis: A66 (sy-a66); Tocris: H1152 (2414) and NU6140 (3301); Sigma: Tamoxifen (T5648), (Z)-4-hydroxytamoxifen (H7904) and dihydrocytochalasin B (D1641).

**Mice**. Mice were kept in individually ventilated cages and cared for according to UK Home Office regulations, with procedures approved by the Ethics Committees of Queen Mary University London, UK and University College London, UK. Unless otherwise mentioned, the mice used were backcrossed to the C57BL/6 background for 10 generations, and littermates were used as controls.

**Generation of *Pik3ca*$^{H1047R+neo}$ mice**. Gene targeting was carried out by Taconic Artemis (Cologne, Germany) and shown schematically in Fig. 1a. Genomic DNA covering the last 6 exons of the coding region of *Pik3ca* was isolated and a point mutation, resulting in conversion of CAT (H) to CGT (R) in the C-terminal p110α kinase domain, was introduced by site-directed mutagenesis. In addition, a *neo* selection cassette flanked by *frt* sites was inserted into exons 19 and 20 in the targeting vector using standard restriction enzyme- and PCR-based cloning techniques. The linearized targeting construct was electroporated into C57BL/6 J ES cells. G418-resistant clones with targeted integration of the vector were identified by Southern blot analysis of *Hin*dIII-digested, *Xmn*I-digested or *Bsr*GRI-digested genomic DNA using different probes homologous to genomic sequences flanking the selection cassette. The presence of the introduced CGT sequence was confirmed by DNA sequencing. Chimeras were generated from clones with the CGT mutation and founders identified by PCR analysis of tail DNA. Male chimeras carrying the CGT mutation (germline transmitters) were bred with C57BL/6 females to generate the *Pik3ca*$^{H1047R+neo}$ hypomorph mice. The presence of the *Pik3ca*$^{H1047R+neo}$ mutant allele was monitored by PCR using the following primers (shown schematically in Fig. 1a): primer 1 = onco-p110α-F: 5′-GGTTCCAGCCTGAATAAA GCC-3′; primer 2 = onco-p110α + cass: 5′-AGATCAGCAGCCTCTGTTCC-3′, giving rise to an expected PCR fragment of 307 bp (Supplementary Fig. 1b). In each amplification cycle, denaturation was performed for 30 s at 95 °C, annealing for 30 s at 60 °C and extension for 1 min at 72 °C; the number of cycles was 35 followed by a final extension of 10 min at 72 °C.

$Pik3ca^{H1047R+neo}$ hypomorph mice were bred with mice that express a germline FLP recombinase, which was either constitutively active ($hACTB::Flpe$ mice[19]) or inducible by 4-hydroxytamoxifen (4-OHT) treatment ($CAG::Flpe-ER^{T2}$ mice[20]), in order to induce Flpe-mediated deletion of the *neo* selection cassette. Flpe-mediated or Flpe-ER$^{T2}$-mediated recombination of the *neo* selection cassette was monitored by PCR using the following primers (Supplementary Fig. 1b): primer 1 = onco-p110α-F: 5′-GGTTCCAGCCTGAATAAAGCC-3′; primer 3 = onco-p110α-R: 5′-CACAGCTGTCCTGGGTAAGG-3′. The same PCR conditions as described above where used. The size of the expected PCR fragments was 425 bp (recombined mutant allele) or 340 bp (non-recombined locus).

**Isolation of primary MEFs**. Embryos were obtained from timed pregnant females. The day of the presence of a copulation plug was considered as day E0.5. E13.5 embryos were minced, dissociated using trypsin and cells allowed to adhere on tissue culture dishes in DMEM, 10% FBS, penicillin and streptomycin, as described in ref. [52] Unless otherwise indicated, early passage (P2 to P3) MEFs were used in experiments.

**Cell culture**. MEFs and human dermal fibroblasts were cultured in DMEM with 10% FBS, penicillin and streptomycin. Mouse primary keratinocytes were cultured in Keratinocyte-SFM 1× (Invitrogen, 37010–022), 20 μM CaCl$_2$ and 20 mM HEPES. MCF-10A cells (ATCC CRL 10317) were cultured in DMEM/F12 with 5% horse serum, 20 ng ml$^{-1}$ EGF, 0.5 μg ml$^{-1}$ hydrocortisone, 100 ng ml$^{-1}$ cholera toxin, 10 μg ml$^{-1}$ insulin, penicillin and streptomycin. MCF-10A cells were used as an epithelial cell model to confirm the findings made with primary cell lines derived from the inducible $Pik3ca^{H1047R+neo}$ mouse model.

**In vivo induction of $Pik3ca^{H1047R}$ in mice**. For postnatal removal of the *neo* cassette by tamoxifen-inducible Flp-mediated recombination, $Pik3ca^{WT};Flpe-ER^{T2}$ and $Pik3ca^{H1047R+neo};Flpe-ER^{T2}$ mice at the age of 8 weeks were dosed with tamoxifen in corn oil (80 mg kg$^{-1}$) by gavage for 5 consecutive days. Animals were followed over time and culled when they became apparently ill. A healthy control animal was culled in parallel.

**In vitro induction of $Pik3ca^{H1047R}$ in MEFs**. $Pik3ca^{WT};Flpe-ER^{T2}$ and $Pik3ca^{H1047R+neo};Flpe-ER^{T2}$ MEFs cultured in standard growth medium were treated for 72 h (unless stated otherwise) with 1 μM 4-OHT solubilized in ethanol (vehicle). Culture medium was replenished every 48 h with fresh 4-OHT. Flpe-ER$^{T2}$-mediated recombination of the *Neo* selection cassette was monitored by PCR.

**Isolation and induction of $Pik3ca^{H1047R}$ in mouse skin keratinocytes**. Adult mouse dorsal epidermal keratinocytes were isolated as described in ref. [53] For use in experiments to determine aneuploidy, 8-week-old $Pik3ca^{WT};Flpe-ER^{T2}$ or $Pik3ca^{H1047R+neo};Flpe-ER^{T2}$ mice were treated with tamoxifen (80 mg kg$^{-1}$) per gavage for 5 consecutive days. Two weeks after the last day of treatment, the dorsal skin was shaved and isolated, cut into small pieces, washed once with PBS and twice with 70% ethanol, submerged in trypsin (T4424, Sigma) and incubated overnight at 4 °C. The epidermis was detached and minced and cells were plated on 10 cm dishes. After 24 h, cells were washed once with Hank's solution, incubated with culture medium which was replenished every other day until the cultures reached confluence (6–7 days after seeding). Coating of culture dishes for keratinocyte isolation was as follows: dishes were coated with collagen and fibronectin prepared by incubation of 10 μl of collagen (354236, BD Biosciences) and 10 μl fibronectin (354008, BD Biosciences) in 980 μl of EMEM (06–174 g, Lonza) +5 mM HEPES (15630–056, Gibco) to obtain a final concentration of 30 μg ml$^{-1}$ collagen and 10 μg ml$^{-1}$ fibronectin, incubated with coating solution at 4 °C overnight on a rocker platform followed by three washes with Hank's solution (14175–129, Gibco). For centrosome experiments, keratinocytes were explanted from non-treated 8-week-old mice, cultured for 5 days, and then seeded on coated coverslips and, once adhered, treated with 4-OHT (1 μM) or vehicle for 2 consecutive days.

**Human subjects**. This study was approved by the UK National Research Ethics Committee in accordance with the Declaration of Helsinki. Written informed consent was obtained from all participants. Human dermal fibroblasts were obtained from skin biopsies taken from the right arm and right leg of a human subject (C1)[21]. Sanger sequencing and target-enriched next generation sequencing (mean coverage ×1000) were used to confirm the presence of a 50% heterozygous $PIK3CA^{H1047L}$ mutation burden in leg cells. Arm cells were confirmed to be WT for $PIK3CA$.

**Mouse colon cancer model**. All mice used were on an outbred background and were from the same breeding colony. Mice bearing heterozygous *Apc*-targeted alleles (Apc$^{flox/+}$)[54] were crossed with mice bearing the $VillinCreER^{T}$ transgene[55] (expressed in the small and large intestine) to generate $VillinCreER^{T};Apc^{flox/+}$ mice. $VillinCreER^{T};Apc^{flox/+}$ mice were then crossed with $Pik3ca^{H1047R+neo};Flpe-ER^{T2}$ mice, and kept on an outbred background to generate $Pik3ca^{H1047R+neo};Flpe-ER^{T2}:VillinCreER^{T};Apc^{flox/+}$ mice. Mice were genotyped for the Apc$^{flox/+}$-targeted allele and the $VillinCreER^{T}$ recombinase transgene as described previously[54,55].

$VillinCreER^{T}$ and $Flpe-ER^{T2}$ recombinase activity from all experimental groups was induced at 6–12 weeks of age via gavage of an 80 mg kg$^{-1}$ dose of tamoxifen in corn oil for 5 consecutive days. Mice were killed by cervical dislocation when they became symptomatic of disease of an intestinal tumour burden, indicated by pale feet caused by intestinal tumour burden-induced anaemia, a hunched appearance and/or a distended abdomen. Intestines were fixed in Methacarn (methanol: chloroform:glacial acetic acid (4:2:1)), and the number of lesions was scored macroscopically.

**Immortalization of MEFs by p53 shRNA**. Phoenix ecotropic HEK293T cells were transfected using FuGENE with a shRNAmir retroviral vector targeting p53 (ref. [56]) and supernatants harvested 48 h later. Early passage primary MEFs were incubated with retroviral supernatants, selected with 1 mg ml$^{-1}$ puromycin and p53-null immortalized MEFs expanded and frozen for further experiments.

**Generation of Nutu cell lines**. A total of 10$^6$ $Pik3ca^{WT};Flpe-ER^{T2}$ or $Pik3ca^{+neo};Flpe-ER^{T2}$ primary and p53 shRNA-immortalized MEFs were treated with 4-OHT in vitro and injected subcutaneously into the upper and lower parts of each flank of female HsdHli:CD1-$Foxn1^{nu}$ nude mice ($n = 8$, Harlan UK). Only p53 shRNA-immortalized $Pik3ca^{H1047R}$ MEFs gave rise to tumours and with low penetrance (three out of eight tumours injected) after a latency period of 2–3 months. Tumours were excised, minced, trypsinized and propagated as cell lines in culture, further referred as Nutu1, Nutu2 and Nutu3.

**Generation of in vitro Pik3ca$^{H1047R}$-transformed MEFs**. $Pik3ca^{WT};Flpe-ER^{T2}$ or $Pik3ca^{H1047R+neo};Flpe-ER^{T2}$ primary MEFs were seeded in 15 cm dishes in standard culture medium and treated with 4-OHT. Cells were allowed to reach confluency and kept in culture for 4 weeks until foci were established. Only $Pik3ca^{H1047R+neo};Flpe-ER^{T2}$ MEFs yielded colonies. Individual well-isolated colonies were harvested by cloning cylinders (Millipore, TR-1004) and transferred to 12-well plates. Cells were then expanded and frozen for further experiments.

**Centrosome analysis**. For centrosome analysis in cultured cells, cells grown on coverslips treated with poly-L-lysine (P8920, Sigma) or on collagen I coated glass culture slides in the case of MCF-10A cells (Corning, 354630), were fixed with ice-cold methanol for 10 min, blocked with PBS containing 3% BSA for 40 min and incubated overnight at 4 °C with primary antibodies. All primary antibodies were detected using species-specific fluorescently labelled secondary antibodies (Molecular Probes) and DNA was detected with 0.2 μg ml$^{-1}$ DAPI (D9542, Sigma). Specimens were mounted in mowiol mounting medium (475904, Calbiochem). Confocal laser scanning microscopy was performed with an LSM 710 (Zeiss) confocal microscope.

For centrosome analysis in adult tissues, 8-week-old $Pik3ca^{WT};Flpe-ER^{T2}$ or $Pik3ca^{H1047R+neo};Flpe-ER^{T2}$ mice were treated with tamoxifen (80 mg kg$^{-1}$) per gavage for 5 consecutive days. Two weeks after this treatment, tissues were snap-frozen in OCT freezing medium (TFM5, Electron Microscopy Sciences) and 10 μm cryosections affixed to poly-L-lysine coated slides. Sections were thawed at room temperature for 30 min before staining, fixed for 15 min in 4% PFA in PBS at room temperature, washed 3 times for 5 min with PBS, blocked with mouse IgG blocking reagent (MKB-2213, Vector labs) for 1 h, washed times for 5 min with PBS and blocked for 1 h in 5% donkey serum in PBS-0.3% Tween (PBS-T). Slides were then incubated overnight at 4 °C with antibodies to pericentrin (1:500 in PBS-T), washed 3 times in PBS, incubated with anti-rabbit-Cy3 antibody (1:500 in PBS-T) and AlexaFluor 488-phalloidin (1:500 in PBS-T) for 1–2 h in the dark at room temperature, washed three times with PBS and mounted with Vectashield mounting media containing DAPI (H-1200, Vector Labs).

For centrosome analysis in embryos, whole-mount embryo immunofluorescence was performed. E8.5 embryos were isolated, transferred to 1.5 ml Eppendorf tubes and fixed in 4% PFA in PBS at 4 °C overnight on a rocker platform, followed by three washes with 0.1% Triton X-100 in PBS (PBT). Embryos were incubated in 0.25% Triton X-100 in PBS for 15 min and washed three times with PBT before blocking with 2.5% normal donkey serum (D9663, Sigma), 2.5% normal goat serum (G9023, Sigma) and 3% bovine serum albumin (A7906, Sigma) in PBT for 3 h. Following removal of the blocking solution, anti-pericentrin antibody in 200 μl PBT was added to the Eppendorf tube containing the embryos and incubated overnight at 4 °C. The following day, 1 ml PBT was added to the solution containing the embryos, embryos washed 5 times in PBT and incubated in PBT for an extra 20 min. Then, 200 μl PBT solution containing the secondary Cy3-anti-rabbit antibody was added and incubated overnight at room temperature. Embryos were then washed 5 times for 5 min in PBT and once for 15 min. Finally, embryos were mounted on depression slides (S175201, Fisher scientific) in Vectashield mounting media containing DAPI (H-1200, Vector Labs) for 2 days.

**Analysis of chromosome segregation errors**. Cells cultured on poly-L-lysine-treated coverslips were fixed in 10% Triton X-100, 1 M PIPES, 0.5 M EGTA, 1 M MgCl$_2$ and 4% formaldehyde, blocked with PBS containing 3% BSA for 40 min and incubated for 1.5 h at room temperature with primary antibodies: human Crest (1:300) and β-tubulin (1:600). Secondary antibodies (1:500) goat anti-rabbit conjugated to AlexaFluor 594 and goat anti-human AlexaFluor 647 were added for 1.5

h at room temperature. DNA was stained with 1 µg ml⁻¹ DAPI (10236276001, Roche) and coverslips mounted in Vectashield (Vector H-1000). Images were acquired using an Olympus DeltaVision RT microscope (Applied Precision, LLC) equipped with a Coolsnap HQ camera.

**Transient transfection.** MCF-10A cells were transfected using Lipofectamine 3000 Transfection Reagent (ThermoFisher Scientific; L3000008) according to the manufacturer's instructions. The plasmids used were pBabe-HA-*PIK3CA*$^{WT}$, pBabe-HA-*PIK3CA*$^{H1047R}$, pBabe-HA-*PIK3CA*$^{E545K}$, pBabe-HA-*PIKR1*$^{ED560Y}$ or empty pBabe vector, all obtained from Addgene (deposited by Jean Zhao, Dana-Farber Cancer Institute, Boston, USA). Cells were fixed 72 h after transfection for analysis of centrosome numbers.

Chromosome spread analysis. MEFs and Nutu cells were treated with 0.1 µg ml⁻¹ colcemid (KayoMax, GIBCO) and mouse keratinocytes with 50 ng ml⁻¹ of nocodazole (M1404, Sigma) for 4 h, trypsinized, and resuspended in 0.56% KCl for 30 min at 37 °C. Cells were then fixed with 3:1 ice-cold methanol:acetic acid, pelleted and washed three times with methanol:acetic acid before being dropped on a pre-cleaned glass slide from a height of ~6 inches. Slides were allowed to dry and stained with Vectashield mounting medium containing DAPI (Vector labs). Images of metaphase spreads were taken with a 63x objective on a Zeiss 510 microscope and chromosomes were counted using Image J. The cell population with <40 chromosomes is defined as those with 36–39 chromosomes.

**Time-lapse microscopy.** To study cell fate during and after mitosis of cells with more than two centrosomes, *Pik3ca*$^{H1047R+neo}$;*Flpe-ER*$^{T2}$ mice were crossed to *GFP-Cent2* transgenic mice[57] to generate *Pik3ca*$^{H1047R+neo}$;*Flpe-ER*$^{T2}$;*GFP-Cent2* MEFs. Cells were seeded in µ-Plate 96 wells (Ibidi; IB-8962) in DMEM medium 48 h after treatment with 4-OHT, followed by filming of the cells for 30 h at 37 °C. To study the efficiency of the G1 checkpoint, an assay of DCB-induced tetraploidy was used. Briefly, MEFs cells were seeded on ibidi µ-Plate 96 wells in DMEM medium with 4-OHT for 24 h, followed by overnight incubation in the presence of 10 µM DCB followed by washing out of this agent in fresh culture medium and by filming of the cells for 30 h at 37 °C. Images were collected with a high-content analysis microscope (ImageXpress Micro XLS Widefield High-Content Analysis System/Molecular Devices) every 15 min using MetaXpress High-Content Image Acquisition and Analysis Software (Molecular Devices). Time-lapse images from the whole duration of the experiment were then converted into a movie using Image J, and cells with more than two centrosomes or binucleated cells entering mitosis were tracked.

**Measurement of RhoA activity.** The capacity of RhoA–GTP to bind to GST–rhotekin–RBD was used to determine the cellular levels of active GTPase. Cells were lysed in extraction buffer (50 mM Tris.HCl pH 7.4, 100 mM NaCl, 5 mM MgCl₂, 1% NP-40, 10% glycerol, 1 mM DTT, protease and phosphatase inhibitor cocktail) and the RhoA pull-down assay was performed by incubation of cleared lysates with GST–rhotekin–RBD (Cytoskeleton Inc) for 45 min at 4 °C, followed by four washes of the beads in wash buffer (50 mM Tris.HCl, pH 7.6, 150 mM NaCl, 10 mM MgCl₂, 1 mM DTT, protease and phosphatase inhibitor cocktail). Bound proteins were solubilized by the addition of 25 µl of SDS–polyacrylamide gel electrophoresis (SDS–PAGE) Laemmli loading buffer, followed by separation on 12.5% SDS–PAGE gels and western blotting for RhoA.

MEF focus formation assay. Primary Pik3ca$^{H1047R+neo}$;Flpe-ER$^{T2}$ MEFs (passage 2) were seeded at 10⁶ cells per 6 cm dish in the presence of 4-OHT with or without Y27632 (10 µM) or H1152 (0.5 µM) until they reached a confluent monolayer. Medium was replaced every 2 days, containing fresh inhibitors, and confluent monolayer cultures stained with crystal violet after 10 days of total culture to reveal the presence of transformed foci.

**MEF proliferation assay.** Early passage (P2–P4) MEFs were incubated at 2 × 10³ cells per well in 96-well plates in 100 µl medium (DMEM, 10% FBS, penicillin and streptomycin) in the presence of the indicated drugs for 72 h. MTS reagent (G5421, Promega) was added and absorbance at 492 nm was read 3 h later and analyzed using Magellan software. For cell counting experiments, 2.25 × 10⁵ cells were seeded in six-well plates in the presence of 4-OHT. Cells were collected every 24 h and counted using Casy Cell Counter (Roche). The population doubling time was calculated using the following formula: DT = $T$ ln2/ln(Xe/Xb). $T$ is the incubation time in hours, Xb is the cell number at the beginning of the incubation time and Xe is the cell number at the end of the incubation time (ATCC animal cell culture guide).

**BrdU incorporation assay.** MEFs were serum-starved for 48 h in the presence of 4-OHT followed by incubation in FBS-containing medium for the indicated times. One hour before collection, cells were incubated in labelling medium containing 10 µM 5-bromo-2'-deoxyuridine (B5002, Sigma) followed by trypsinization, washing with PBS and fixation in 70% ethanol. Following two further washes in PBS, cells were incubated with 2 M HCl for 20 min and probed with anti-BrdU monoclonal antibody (5292, Cell Signaling Technologies) and detected with fluorescein isothiocyanate-conjugated goat anti-mouse IgG2a (Jackson ImmunoResearch Labs). Cells were resuspended in 500 µl propidium Iodide solution (50 mg ml⁻¹;

P4864, Sigma) and RNase A (100 µg ml⁻¹; R5503, Sigma) and analyzed using a BD Accuri C6 flow cytometer (BD Bioscience) and FlowJo software.

**Cell cycle analysis.** Experiments were carried out in synchronized or non-synchronized MEF cell populations. For experiments with synchronized cells, MEFs were induced with 4-OHT for 72 h. For the last 24 h of this culture, cells were incubated in the presence of 12 µM Aphidicolin (178273, Merck) followed by washing out of this agent ('release') in culture medium for the times indicated. After washing with PBS, cells were detached by trypsinization, collected and fixed in ice-cold 70% ethanol. For experiments with asynchronous cells, MEFs incubated in culture medium were induced with 4-OHT on day 0, and further incubated for 3 and 5 days. After washing with PBS, cells were detached, collected and fixed in ice-cold 70% ethanol. To determine the mitotic index, cells were probed with pH3-S10 polyclonal antibody (Ab5176, Abcam) and detected with FITC-labelled goat anti-rabbit IgG2a (Jackson ImmunoResearch Labs). Cells were resuspended in 500 µl propidium iodide solution (50 mg ml⁻¹; P4864, Sigma) and RNase A (100 µg ml⁻¹; R5503, Sigma) and analyzed using a FACSCalibur (BD Bioscience) and FlowJo software.

**Apoptosis assay.** The assay was performed using a FITC-Annexin V Apoptosis Detection Kit II (BD Pharmingen) according to manufacturer's instructions. Briefly, 5 days after incubation in the presence of 4-OHT, MEFs were detached by trypsinization, washed with cold PBS and resuspended in 1× binding buffer at a final concentration of 1 × 10⁶ cells ml⁻¹. A total of 100 µl of this suspension was incubated with FITC-Annexin V and PI solution for 15 min at room temperature in the dark. After incubation, 400 µl of binding buffer was added and the cells were analyzed using a FACSCalibur (BD Bioscience) and FlowJo software.

**Senescence assay.** The assay was performed using a Senescence Cells Histo-chemical Staining Kit (CS0030, Sigma) according to manufacturer's instructions. Briefly, 72 h after incubation in the presence of 4-OHT, MEFs were fixed for 7 min in fixation buffer, washed in PBS and stained with staining mixture overnight at 37 °C without CO₂. The cells were observed under a bright field microscope (DM IL LED, Leyca). The percentage of β-galactosidase-positive cells was calculated by counting blue-stained cells and the total number of cells.

**DNA damage checkpoint experiments.** For cell counting, MEFs were seeded in 12-well plates in the presence of 4-OHT. The day after, cells were treated with 0.2 µg ml⁻¹ doxorubicin (D1515, Sigma) for 20 h. After the incubation time, the drug was washed out and cells allowed to proliferate. Cells were collected and manually counted for the indicated times. For p53 Western blotting, MEFs were seeded in 6 cm plates in the presence of 4-OHT. 48 h after induction, cells were treated with 0.2 µg ml⁻¹ doxorubicin for 18 h, after which cells were lysed and total protein isolated.

**Endomucin staining.** Freshly isolated E9.5 mouse embryos were fixed in 4% paraformaldehyde overnight at 4 °C. After washing in PBT, embryos were dehydrated in increasing concentrations of methanol (50%, 80% methanol/PBT, 100% methanol), rehydrated in decreasing concentrations of methanol, followed by a 30 min wash in Pblec buffer (0.2 mM CaCl₂, 0.2 mM MgCl₂, 0.2 mM MnCl₂ and 2% Triton X-100 in PBS) and overnight incubation at 4 °C with antibody to endo-mucin, diluted 1:20 in Pblec. After five washes in PBT, embryos were incubated with Cy5-labelled secondary antibody diluted 1:100 in PBS, 0.5% BSA, 0.25% Tween-20. Finally, embryos were washed three times in PBS, post-fixed for 1 min in 4% paraformaldehyde and mounted.

**Immunoblotting.** Cells were lysed in lysis buffer (10 mM Tris.HCl pH 7.5, 1% Triton X-100, 0.5% NP-40, 150 mM NaCl, 2 mM CaCl₂, 0.1 mM Na₃VO₄, protease inhibitors (Calbiochem), 1 mM PMSF). The lysates were incubated for 20 min and cleared by centrifugation at 4 °C. The samples were denatured in sample buffer (10% SDS, 20% glycerol, 200 mM Tris.HCl pH 6.8, 10 mM DTT, 0.05% bromo-phenol blue), resolved by SDS-PAGE, and transferred onto an Immobilon-P PVDF membrane (Millipore). The blots were incubated in blocking buffer (5% (w/v) nonfat dry milk in Tris-buffered saline plus Tween 20) for 1 h and incubated with primary antibodies for 16 h at 4 °C. The blots were incubated with anti-mouse or anti-rabbit IgG horseradish peroxidase-conjugated secondary antibodies (GE Healthcare) for 1 h at room temperature. The antibody–antigen complex was visualized by Pierce ECL Western Blotting Substrate (Thermo Scientific). Uncropped immunoblots and larger blot areas are presented in Supplementary Fig. 19.

**Mutation and copy number analysis.** TCGA breast cancer mutation and copy number analysis was performed as described in ref. [37], with genome doubling assessed as outlined in ref. [14]. In brief, mutations were classified as clonal or subclonal based on the 95% confidence interval of the posterior distribution of the cancer cell fraction. Mutations where the 95% confidence interval overlapped with 1 were grouped as clonal, with all other mutations classified as subclonal. To assess whether H1047R mutations in *PIK3CA* were significantly more often clonal than

expected by chance in breast cancer, a permutation test was used, as outlined in ref. [37]. In brief, the same number of mutations as observed at H1047R sites in *PIK3CA* were randomly sampled from the entire cohort of mutations in TCGA breast cancer 10,000 times, obtaining a background distribution, representing the expected proportion of clonal/subclonal mutations. A *P*-value was then obtained by comparing the observed proportion of clonal/subclonal mutations to this background distribution. The same technique was used to assess whether *PIK3CA* mutations occurred more often before or after genome doubling. In this case only genome-doubled breast tumours were used and mutations were timed relative to copy number events. Timed mutations were restricted to those occurring in regions with at least two copies of the major allele. Given that a genome doubling event will double mutations preceding the event, all mutations present at multiple copies were classified as preceding doubling, while those only present at one copy were classified as occurring after the duplication.

Breast cancer subtype classification was performed using the genefu package in R, grouping breast tumours into five subtypes: luminal B-like; Her2-like; basal-like; luminal A-like; and normal. ER status was assessed by immunohistochemistry and obtained from clinical data from TCGA portal.

Tendency towards mutual exclusivity between mutations in *PIK3CA* and *TP53* was assessed using a Fisher's exact test, using all breast cancer tumours and using either all non-silent mutations or the ten most frequent hot spots mutations. This mutual exclusivity remained significant when focusing on only genome-doubled tumours ($P = 0.04$). A two-tailed Fisher's exact test was used to assess significance.

**Data availability**. All relevant data are available from the authors.

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

## Acknowledgements

We thank Susana Godinho for feedback and experimental advice, Maria Whitehead for assistance in the writing of the manuscript, Romain Galmes and Elena Lopez-Guadamillas for experimental help, members of the Cell Signalling Group for critically reading the manuscript and Susan M. Dymecki for the *Flpe-ER*[T2] mice. We are grateful to the human subjects who provided primary cells for this study. Personal fellowships were from EU Marie Curie (PIEF-GA-2008-219945; to IMB), EMBO (ALTF 165-2013; to S.C.) and the Ong Hin Tiang & Ong Sek Pek Foundation, Malaysia (to E.L.). M.G. was supported by the Ministry of Economy and Innovation (Spain) (SAF2014-59950-P), the Catalan Government (2014-SGR-725), the People Programme (Marie Curie Actions) from the European Union's Seventh Framework Programme FP7/2007-2013 (REA grant agreement 317250), the Institute of Health Carlos III and the European Regional Development Fund under the integrated Project of Excellence no. PIE13/00022 (ONCOPROFILE). W.A.P. was supported by the National Health and Medical Research Council (NHMRC) of Australia (project grant No. 1080491). V.E.R.P. was supported by the Wellcome Trust (097721/Z/11/Z). R.K.S. is supported by the Wellcome Trust (WT098498), the Medical Research Council (MRC_MC_UU_12012/5). V.M. and A.C. were supported by Cancer Research UK (C1295/ A15937). S.S is supported by BBSRC (BB/J00989X/1) and Wellcome Trust Senior Investigator Award (103788/Z/14/Z). C.S. is a senior Cancer Research UK clinical research fellow and is funded by Cancer Research UK (TRACERx), the CRUK Lung Cancer Centre of Excellence, Stand Up 2 Cancer (SU2C), the Rosetrees Trust, NovoNordisk Foundation (ID 16584), the Prostate Cancer Foundation, the Breast Cancer Research Foundation, the European Research Council (THESEUS) and National Institute for Health Research University College London Hospitals Biomedical Research Centre. B.V. is supported by Cancer Research UK (C23338/A15965) and the UK NIHR University College London Hospitals Biomedical Research Centre.

## Author contributions

I.M.B., R.P. and B.V. designed research; I.M.B., R.P., S.D.C., S.M.D., V.M., N.K., W.P., K. G.M., E.L., L.S., D.M., M.G. and L.S.M. performed research; V.M. and R.K.S. contributed new reagents, C.L.S, N.J.B. and N.M. performed data acquisition and/or analysis; I.M.B., R.P. and B.V. analyzed data; I.M.B., R.P. and B.V. wrote the paper with input from L.S. and C.S. All authors discussed the results and commented on the manuscript.
