## [Peer Review File · Nature Communications]

Reviewers' comments:

Reviewer #1 (Remarks to the Author):

In this manuscript, Berenjano et al. discovered novel roles of oncogenic PIK3CA in centrosome amplification and tetraploidy tolerance. Using a combination of in vivo and in vitro models, the authors showed that expression of p110 α H1074R at endogenous levels promotes centrosome amplification most likely through activation of the Akt/ROCK pathways. In addition, the authors provide evidence for p110 α H1074R playing a role on tolerating genome doubling (or tetraploidization). The data presented in this manuscript is novel and impressive and the statistical analysis is well done revealing significant effects on cellular physiology driven by the p110 α H1074R allele. Importantly, these findings point toward previously unappreciated roles for this oncogenic mutation in cancer progression.

However, as presented the data are confusing and need either better explanation or additional experiments to clarify the significance of the results.

The authors show that p110 α H1074R induces centrosome amplification. In addition, they show that this phenotype does not have any effects on apoptosis (figure S5d), senescence (supp. figure 7a), cell cycle progression (supp. figure 5e, supp. figure 7d), DNA synthesis (figure 2c), DNA damage (supp. figure 7b), generation of binucleated cells (figure 2d), or chromosome segregation errors (figure 4b). Instead, the authors show that p110 α H1074R cells are more proficient at completing mitosis when they have more than 2 centrosomes compared to wild type cells. This raises several questions that need to be addressed in the manuscript. Do the authors hypothesize that centrosome duplication is the reason why this mutation causes embryonic lethality? Is centrosome amplification the mechanism by which p110 α H1074R increases tumor burden in the Apc mice (figure s4c)? It is not clear what the functional consequence of centrosome amplification is, resulting from p110 α H1074R expression.

There is a major discrepancy between the amplification of centrosomes and the results presented in figure 5a, which show that p110 α H1074R promotes aneuploidy in MEFs. These MEFs were analyzed 3 days after p110 α H1074R induction indicating that aneuploidy arises quickly. Does this mean that the MEFs analyzed in figures 1a and 1b, the mouse keratinocytes in figure 1b, the E8.5 embryonic cells in figure 1c, and the adult mouse tissues in figure 1d are all aneuploid and karyotypically heterogeneous?

At least the MEFs don't seem to be heterogeneous or aneuploid because DNA profiles presented in supp. figure 5e and supp. figure 7d look normal. This is really confusing. On the one hand p110 α H1074R causes centrosome amplification without promoting chromosome segregation errors, yet it causes genome doubling and aneuploidy. But the MEFs DNA content is not affected?

The authors hypothesize that p110 α H1074R causes aneuploidy by promoting genome duplication. If cell cycle progression and DNA synthesis are not affected, when is the genome doubling taking place? The finding that p110 α H1074R may promote genome doubling is very interesting and this reviewer thinks that this is probably the most important discovery by the authors. It is reasonable to think that centrosome amplification causes genome doubling. However, the results do not support this.

Although the number of chromosomes point to tetraploidization events, a more detailed karyotypic analysis (e.g. SKY spectral analysis) could be done to show that the genomes are close to 4N rather than 2N plus multiple copies of specific chromosomes that may give fitness advantage in the context of p110 α H1074R. This is specially a concern because tetraploidization induces p53 while p110 α H1074R does not.

Minor points:

In page 2, line 72, "Fig. 1b" refers to the wrong figure.

In supplemental figure 1b, what do DH1, CE4, CF3 mean?

In supplemental figure 1c and several other figures the levels of p110 α and total Akt are not shown.

In page 2, line 73, Fig. 1b refers to the wrong figure.

Page 4 line 179, supplemental fig. 3d refers to the wrong figure.

Page 4 line 203, supplemental fig. 4a refers to the wrong figure.

Reviewer #2 (Remarks to the Author):

The article entitled: "Oncogenic PIK3CA induces centrosome amplification and tolerance to genome doubling" by I.M. Berenjeno describes the consequences of mutations in the p110 α isoform of PI3K, using both in vivo and in vitro model systems. The authors show that expression of the oncogenic form of p110 α (p110 α h1047R) results in centrosome amplification (more than two centrosomes) and tolerance to tetraploidization. Although the questions developed here are important for cancer biology, and understanding whether and how centrosome amplification and tetraploidization lead to tumour formation are timely questions, this study appears extremely preliminary and descriptive not adding much to the field. Identification and characterization of the underlying mechanisms involved in these processes should be performed. Thus I cannot recommend its consideration for publication by Nature communications.

Major points:

1- The findings that MEFs or mouse tissues expressing p110 α H1047R display centrosome application deserves better characterization. The authors conclude that p110 α H1047R expression leads to centrosome overduplication through RhoA/Rock, but how this is achieved remains to be determined. Also, these proteins have not been involved in increasing centrosome numbers, previously, so how to explain these findings? Through defects in cell division? Through activation or repression of a member of the centriole duplication machinery? It should be investigated how de-regulation of signalling pathways lead to centrosome amplification.

2- The authors have a mouse model, that would allow them to investigate these questions in different organs and tissues. The choice of using MEFs, is not easy to understand as these cells very easily accumulate defects in the processes being analysed here.

3- Which clustering mechanisms are being used in these cells, to sustain bipolar (or pseudo-bipolar spindle formation)?

2- The authors determine the morphology of the mitotic spindle in what appears prometaphase like figures. These are very difficult to evaluate (Fig. 4C), but the correct way of doing this analysis is to analyse spindle morphology in anaphase figures. Multipolar configurations in prometaphase might be resolved in bipolar anaphases... The figure representing a "pseudo-bipolar" is not very different from the one showing a multipolar configuration...

3- It is difficult to understand the analysis of segregation errors in WT and p110 α H1047R cells. If control cells have increased multipolarity, how to explain that they have the same level of segregation errors found in p110 α H1047R cells?

4- The p110 α H1047R appears to facilitate the tolerance to tetraploidization even in the presence

of WT p53. This is a really interesting observation, but how this is achieved should be determined.

5- It would be useful for the reader to know how many cells from how many different animals were used in each experimental condition.

PIK3CA has been extensively studied and is known to be frequently mutated in solid cancers. Alterations in the *PIK3CA* including gene amplifications and somatic missense mutations have been reported in many human cancer types including cancers of the colon, breast, brain, liver, stomach and lung. These somatic missense mutations were proposed to increase the kinase activity of *pik3ca* contributing to cellular transformation. However, the impact of mutant *pik3ca* in early stages of malignancy is still unknown. This is addressed in the manuscript entitled "**Oncogenic PIK3CA induces centrosome amplification and tolerance to genome doubling**", by Inma M. Berenjeno and colleagues.

In the paper, the authors generated a Flp recombinase-based knock in mouse model of mutant *pik3ca* from its endogenous locus and in the heterozygous state. They were able to show that heterozygous expressed mutant *pik3ca* alone did not develop cancer. However, tamaxifen-induced expression of *pik3ca* mutant with intestine-specific heterozygous deletion of the *Apc* tumor suppressor gene accelerated the progression of cancer; suggesting that the mutant *pik3ca* is a weak oncogene on its own. To better understand the biological sequences and early cellular impact of the expression of mutant *pik3ca*, the authors used both primary MEFs and tumor-derived cell. They showed that mutant *pik3ca* induction leads to centrosome amplification through centrosome over duplication which is mediated by the activation of the ROCK Ser/Thr kinase and the phosphorylation of nucleophosmin on T99. However, by using live imaging by time-lapse microscopy of MEFs expressing centrin2 tagged with GFP, they showed that centrosome amplification did not lead to more segregation errors. To further understand the underlying mechanism, they performed an analysis of the different spindle configurations in mitotic cells with more than two centromeres and were able to show that the mutant *pik3ca* cells are more proficient at completing mitosis and resolve multipolar spindle intermediates more efficiently than the wild-type. In spite of the unaltered frequency of segregation errors in the mutant *pik3ca*, analysis of metaphase chromosome spreads showed an increase in aneuploid cells pointing towards the possibility of genome doubling, suggesting an alternative tolerance routes to genome doubling than the p53. This was further proved by genetics analysis showing that mutations in *pik3ca* and *p53* have a strong tendency to be mutual exclusive suggesting that the role of *pik3ca* mutation as a potential tolerance mechanism for genome doubling in breast cancer is independent of the p53 pathway. Importantly, this manuscript shows that the *pik3ca* induced abnormalities cannot be reverted pharmacologically while centrosome amplification can, which explains the currently limited therapeutic impact of PI3K-targeted therapies and further supports the fact that *pik3ca* mutation is not on its own a clear predictor of sensitivity to PI3K pathway inhibitors. On the other hand, the study suggests that pharmacological intervention with PI3K signaling in cancer may reduce tumor heterogeneity and evolution by limiting the perpetuation of genomically unstable tetraploid cells.

Although the findings in this manuscript are significant, several concerns remain to be addressed:

- 1- Three hotspots mutations are within the helical and kinase domains of *pi3kca*: H1047L/R, E542K/Q and E545K. By examining publically available data

through cBioportal (Gao et al. *Sci. Signal.* 2013 & Cerami et al. *Cancer Discov.* 2012), E545K seems to be the most recurrent in all cancer types in general as well as in colon cancer alone. However, in breast cancer, 1047 is the most recurrent. It is unclear why the authors chose to work only on one of these hotspots especially that they focused on colon cancer when inducing *pik3ca* mutant with intestine-specific heterozygous deletion of the Apc.

PIK3CA mutations in All cancers:

PIK3CA mutations in Breast Cancer:

PIK3CA mutations in Colon Cancer:

- 2- Figure 1: Could the authors provide a figure that shows the bands (pAKT-S473) more clearly?
- 3- Please add abbreviations to figures: (Error bars, S.D.; WT, wild-type)
- 4- The authors conclude that tamoxifen-induced expression of *pik3ca* mutant with intestine-specific heterozygous deletion of the Apc tumor suppressor gene accelerated the progression of colon cancer; suggesting that the mutant *pik3ca* is a weak oncogene on its own. To make the assumption that the mutant *pik3ca* is a weak oncogene on its own, other cancer types and other cancer promoting genetic lesions should be considered. Also the most recurrent mutation in colon cancer is E545K which was not included in the experiment.
- 5- The authors mention that mutations in PIK3CA and TP53 are mutually exclusive in breast cancer; but did not indicate if the analysis showed significance in other/all cancer types. Also they did not indicate if the analysis showed significance when only looking at clonal mutations which is an indicative of *pik3ca* mutation being an early event in the evolution of cancer.
- 6- Given the importance of combined analysis of PIK3CA mutation and PTEN expression, is there a correlation between the presence of *pik3ca* mutations and the expression of PTEN?
- 7- The authors hypothesized a mutation in PIK3CA is an early event in the evolution of breast cancer. No rationale is given to why they picked breast

cancer. Is it because the *pik3ca* mutations is the highest reported? Is the same seen in other solid cancers; for example, colon, brain, liver, stomach and lung? Which also have high *pik3ca* mutation frequency?

- 8- Missing references in the introduction. For example: ‘Thus far, the oncogenic potential of PI3K has largely been attributed to its role in stimulating processes such as cell survival and proliferation, spurring the development of inhibitors of the PI3K pathway as anti-cancer agents.’

Oncogenic *PIK3CA* induces centrosome amplification and tolerance to genome doubling (NCOMMS-16-09268-T)

Point-by-point response to the Reviewers comments

We found the Reviewers' feedback very helpful and constructive, and believe that addressing their comments has greatly improved our manuscript. Below we provide answers/clarifications to the questions/points raised. We have updated the manuscript where required.

Reviewer #1

The Reviewer mentioned that 'The data presented in this manuscript is novel and impressive and the statistical analysis is well done, revealing significant effects on cellular physiology driven by the p110 α ^{H1074R} allele. Importantly, these findings point toward previously unappreciated roles for this oncogenic mutation in cancer progression'.

Major comments:

1. Do the authors hypothesize that centrosome duplication is the reason why this mutation causes embryonic lethality? Is centrosome amplification the mechanisms by which p110 α ^{H1074R} increases tumor burden in the *Apc* mice (Fig. s4c)? It is not clear what the functional consequence of centrosome amplification is, resulting from p110 α ^{H1074R} expression.

These are two excellent questions. A key problem in addressing the functional consequence of the observed centrosome amplification is the lack of tools or approaches available to *selectively* interfere with the process of centrosome amplification.

With regard to the first question: Centrosomes control, amongst others, cell polarity, migration and division, all of which are critical during embryonic development. In experiments that are not included in the manuscript, we assessed the possible role of centrosome amplification in p110 α ^{H1074R}-induced embryonic lethality by exploiting our finding that inhibition of the ROCK Ser/Thr kinase blocks p110 α ^{H1074R}-induced centrosome amplification in cells (Fig. 3d). In an attempt to rescue the p110 α ^{H1074R}-induced embryonic lethality, we crossed our p110 α ^{H1074R} mice with ROCKII knockout mice, as ROCKII is the isoform involved in centrosome duplication. However, heterozygous or homozygous loss of ROCKII in the p110 α ^{H1074R} background did not affect the timing of embryonic lethality, and the embryos still died around E9.5. This is perhaps not as unexpected given the number and diversity of cell biological processes that p110 α controls, in addition to centrosome biology. We speculate that the reason for the p110 α ^{H1074R}-induced embryonic lethality is multifactorial, which is also suggested by recent findings of vascular defects in embryos in a similar p110 α ^{H1074R} mouse model [2].

With regard to the increase in tumour burden of the *Apc*^{+/-}*Pik3ca*^{H1047R} mice, we had commented in the discussion of our manuscript that the specific contribution of centrosome amplification to the cooperation between these two genetic lesions is unknown (see text highlighted below):

"Similar observations were made with our mutant mice, where *Pik3ca*^{H1047R} expression did not lead to spontaneous cancer within the time frame analysed, but accelerated the onset of cancer upon concomitant heterozygous loss of the *Apc* tumour suppressor gene; as shown previously [3], or in the presence of other oncogenic lesions, such as loss of p53 or PTEN [4-6]. However, the specific contribution of *Pik3ca*^{H1047R}-driven centrosome amplification in interaction of mutant *PIK3CA* with other cancer-promoting genetic lesions remains unknown"

To be absolutely clear and upfront with the Reader, we have now inserted a new sentence in the discussion, to more explicitly state that it is not possible to experimentally address this important question at this point in time: 'Unfortunately, there are no tools available to selectively and/or directly interfere with centrosome amplification to formally test its importance in general, and in p110 α^{H1074R} biology in particular.'

2. There is a major discrepancy between the amplification of centrosomes and the results presented in Fig. 5a, which show that p110 α^{H1074R} promotes aneuploidy in MEFs. These MEFs were analyzed 3 days after p110 α^{H1074R} induction indicating that aneuploidy arises quickly. Does this mean that the MEFs (analyzed in Fig. 1a and 1b), the mouse keratinocytes in Fig. 1b, the E8.5 embryonic cells in Fig. 1c, and the adult mouse tissues in Fig. 1d are all aneuploid and karyotypically heterogeneous?

This comment on the possible link between centrosome amplification and aneuploidy is related to Comment 4 from this Reviewer on the link between centrosome amplification and genome doubling: our data indicate that there is no clear link between these phenomena. Please see Comment 4 below for how we have addressed this issue in the revised manuscript.

3. At least the MEFs don't seem to be heterogeneous or aneuploid because DNA profiles presented in supp. Fig. 5e and supp. Fig. 7d look normal. This is really confusing. On the one hand p110 α^{H1074R} causes centrosome amplification without promoting chromosome segregation errors, yet it causes genome doubling and aneuploidy. But the MEFs DNA content is not affected?

The DNA profiles shown in Fig. 5e and Supp. Fig. 7d were generated with the aim of analysing the different phases of the cell cycle. For this reason, the cells with a higher DNA content (>2N) were filtered out on purpose (by FACS gating), given that they were not relevant for cell cycle analysis.

In experiments where the >2N cell population was not gated out, we did not detect a clear difference in DNA content in p110 α mutant compared to WT cells. We have also been struggling to understand this, and speculate that this might be due to insufficient sensitivity/resolution of PI analysis by FACS to be able to reveal the range and level of karyotypic differences observed by metaphase spread analysis (Suppl. Fig. 10).

4. The authors hypothesize that p110 α^{H1074R} causes aneuploidy by promoting genome duplication. If cell cycle progression and DNA synthesis are not affected, when is the genome doubling taking place? The finding that p110 α^{H1074R} may promote genome doubling is very interesting and this reviewer thinks that this is probably the most important discovery by the authors.

It is important to stress that we did not hypothesize that p110 α^{H1074R} leads to aneuploidy by promoting genome duplication but, instead, that p110 α^{H1074R} expression helps cells to tolerate so-called spontaneous tetraploidization, i.e. the presence of extra genome copies acquired through basal errors in cell division, which are independent of p110 α^{H1074R} . MEFs are known to be prone to such spontaneous tetraploidization *in vitro*, as we had explained and referenced in the discussion (see highlighted text):

"Tetraploidy can arise through various errors in cell division, such as cytokinesis failure or defects in mitosis, none of which were found to be affected by p110 α^{H1074R} expression. MEFs are known to spontaneously become increasingly tetraploid at each passage in culture, through unknown mechanisms [7]. Our results therefore suggest that, rather than acting as an inducer of tetraploidisation *per se*, p110 α^{H1074R} helps the cell to tolerate the presence of a doubled genome."

It is reasonable to think that centrosome amplification causes genome doubling. However, the results do not support this.

This interpretation by the Reviewer is absolutely correct: our initial hypothesis was indeed a direct connection between the observed centrosome amplification and aneuploidy induced by p110 α ^{H1047R} expression. However, to our surprise (and disappointment), our further observations did not support this hypothesis. Indeed, contrary to our expectations, *Pik3ca*^{H1047R} mutation does not lead to an increase in single chromosome segregation errors (Fig. 4b). Moreover, cytokinesis failure, a possible mechanism leading to genome doubling, was also not observed in p110 α ^{H1074R} MEFs. The observation that p110 α ^{H1074R} MEFs are very efficient at clustering their centrosomes might be the explanation for the lack of an increase in single chromosome segregation errors, in spite of the presence of extra centrosomes. The lack of higher cytokinesis failure in p110 α ^{H1074R} MEFs, together with the results from the experiment that DCB-induced p110 α ^{H1074R} binucleated MEFs divide more often than DCB-induced WT binucleated MEFs (Fig. 5D), led us to propose the idea that p110 α ^{H1047R} provides tolerance to spontaneously occurring genome doubling.

It is clear from Comments 2 and 4 from Reviewer 1 that the somewhat counter-intuitive observations in our data might be confusing to the Reader, and we therefore propose to clarify these points very explicitly at the beginning of the discussion, by including the scheme below (Fig. 7), and by inserting the following accompanying text: 'At present, the functional connection between these two phenomena in the context of *Pik3ca* mutation remains unclear, as well as their importance in *Pik3ca*-driven cancer biology'.

5. Although the number of chromosomes point to tetraploidization events, a more detailed karyotypic analysis (e.g. SKY spectral analysis) could be done to show that the genomes are close to 4N rather than 2N plus multiple copies of specific chromosomes that may give fitness advantage in the context of p110 α ^{H1074R}. This is specially a concern because tetraploidization induces p53 while p110 α ^{H1074R} does not.

With regard to the last point, it has been shown that tetraploidization does not necessarily correlate with induction of p53 [8].

With regard to the first point: Given that it has been convincingly shown (references [9, 10]) that a triploid genome more likely derives from a tetraploid genome that has lost 1N than through sequential gain of 1N from the 2N state, we believe that the genomic alterations close to 4N observed in p110 α ^{H1074R} MEFs highly likely derive from tetraploid cells that lose chromosomes. We had in fact stated this in the text (text highlighted in grey below) but agree that this may have been a bit 'hidden' and have therefore further explained this point in the text (inserted text highlighted in yellow).

“The observed chromosomal abnormalities in p110 α ^{H1047R} MEFs showed a preponderance of cells with a chromosome number around 4N (i.e. tetraploid) (Suppl. Fig. 10a). This became more apparent upon grouping the observed chromosomal abnormalities in two sets, namely cells with 41-59 or ≥ 60 chromosomes [with the latter group including tetraploids (N=80), triploids (N=60) and all other derivatives]. The cut-off of N=60 was based on the notion that triploid cells are most likely derived from tetraploid precursors than from diploid cells that gain additional chromosomes [9, 10]. Indeed, a triploid genome has been reported to more likely originate from a tetraploid genome that has lost 1N, than through sequential gains of individual chromosomes from the 2N state [9, 10]. This grouping of the observed chromosomal abnormalities in cells with 41-59 or ≥ 60 chromosomes revealed an increase in the fraction of MEFs with ≥ 60 chromosomes, 3 and 5 days after p110 α ^{H1047R} induction (Fig. 5b), pointing towards an increase in tetraploidisation”.

We agree with the Reviewer that detailed karyotypic analysis for example by SKY spectral analysis could be of interest to investigate the possible gain of multiple copies of specific chromosomes that may give fitness advantage in the context of p110 α ^{H1074R}. However, preliminary analysis by CGH (data not shown) does not reveal evidence for a positive selection of specific chromosomes in primary MEFs, 5 days after induction of p110 α ^{H1047R} expression.

Minor points:

- In page 2, line 72, "Fig. 1b" refers to the wrong figure.

This has been changed to Suppl. Fig. 1b

- In Suppl. Fig. 1b, what do DH1, CE4, CF3 mean?

These are the names of the ES cell clones. We have now specified this in the corresponding figure legend.

- In Suppl. Fig. 1c and several other figures the levels of p110 α and total Akt are not shown.

The aim of the blot shown in Suppl. Fig. 1C is to show no major alterations in the PI3K/AKT signalling pathway, which we believe this figure clearly illustrates, using the expression of vinculin as a protein loading control.

- In page 2, line 73, Fig. 1b refers to the wrong figure.

We believe this comment refers to minor comment 1 which we have addressed above.

- Page 4 line 179, Suppl. Fig. 3d refers to the wrong figure.

We do not think there is a mistake in the text, apologies if we have misunderstood this comment.

- Page 4 line 203, Suppl. Fig. 4a refers to the wrong figure.

We do not think there is a mistake in the text, apologies if we have misunderstood this comment.

Reviewer #2

The article entitled: "Oncogenic *PIK3CA* induces centrosome amplification and tolerance to genome doubling" by I.M. Berenjeno describes the consequences of mutations in the p110 α isoform of PI3K, using both *in vivo* and *in vitro* model systems. The authors show that expression of the oncogenic form of p110 α (p110 α ^{H1047R}) results in centrosome amplification (more than two centrosomes) and tolerance to tetraploidization. Although the questions developed here are important for cancer biology, and understanding whether and how centrosome amplification and tetraploidization lead to tumour formation are timely questions, this study appears extremely preliminary and descriptive not adding much to the field. Identification and characterization of the underlying mechanisms involved in these processes should be performed.

We appreciate the Reviewer's comment that 'the questions developed here are important for cancer biology'.

We have to kindly disagree with his/her view that our current study does 'not add much to the field'. Having worked in the PI3K field for over 20 years, the discovery of two previously unknown biological activities of oncogenic *Pik3ca* is a major conceptual advance in this field. Indeed, the early cell biological impact of oncogenic *Pik3ca* in cancer remains largely unexplored, in particular under conditions of mutational activation as observed in cancer, i.e. from the endogenous promotor and in the heterozygote state. We also aimed to assess the impact of *Pik3ca* activation at short time points after induction, in the absence of any other oncogenic mutations as well as in the absence of acute stimulation. This contrasts with other studies that examine the effect of PI3K activation in transformed cell lines, at long time points after mutation and/or in the presence of acute stimuli. In order to achieve these aims, we took advantage of an inducible genetic model that allows heterozygous expression of the *Pik3ca*^{H1047R} hot-spot mutation from the endogenous locus. As far as we are aware, such analysis of low level *Pik3ca*-driven pathway activation as observed in cancer has not been carried out to date. Using this new model and approach, we now show, for the first time, that oncogenic *Pik3ca* leads to centrosome amplification and tolerance to genome doubling, two major phenomena implicated in cancer. These findings are highly relevant since they suggest that oncogenic *Pik3ca* might contribute to tumour heterogeneity and evolution through irreversible genomic effects, which could explain the lack of success of PI3K inhibitors in the clinic.

We agree that the underlying molecular mechanism(s) of centrosome amplification and tolerance to genome duplication provided remain somewhat unexplored, but are in fact nowhere near as superficial as this Referee comment implies. Moreover, it is also important to put this query in the right context: **the signalling mechanisms underlying these phenomena are amongst the main mysteries/unanswered questions in these fields.** Indeed, signalling in these biological phenomena are largely unexplored areas of research in general, with little or no pathways/mechanisms uncovered to date. We feel that uncovering the exact molecular mechanism by which p110 α ^{H1074R} activation leads to centrosome amplification and tolerance to genome duplication, as a sub-part of the current report, which focuses on reporting the discovery of new biological activities of mutated *Pik3ca*, is beyond the scope of the current manuscript.

It is also critical to mention that it is currently not possible to experimentally address the importance of centrosome amplification or tetraploidization in tumour formation *in vivo*: indeed, as we had mentioned in the manuscript, **there are no experimental tools available to selectively and/or directly interfere with these phenonema to formally test their importance in general, and in p110 α ^{H1074R} biology in particular.** This cause and consequence question is a major limitation that this field has been struggling with for many years, and something we unfortunately are not in a position to overcome in the current manuscript.

Having said all this, we have now included new experimental data in the revised manuscript detailing the signalling mechanism of p110 α PI3K towards centrosome amplification, as detailed below. With regards to the

tolerance to tetraploidization, our data are consistent with the finding that hyperactivation of growth factor signalling, in our case by sustained signalling by the downstream effector p110 α , can overcome the proliferative blockade of tetraploid cells [1]. We have now made it more clear in the text that it is most likely that the well-established pro-survival signalling induced by constitutive AKT activation in mutant *Pik3ca* cells leads to increased resistance to the stress induced by tetraploidization.

Major points:

1. The findings that MEFs or mouse tissues expressing p110 α ^{H1047R} display centrosome application deserves better characterization. The authors conclude that p110 α ^{H1047R} expression leads to centrosome overduplication through RhoA/ROCK, but how this is achieved remains to be determined. Also, these proteins have not been involved in increasing centrosome numbers, previously, so how to explain these findings?

It is incorrect to state that RhoA and ROCK have been not been previously involved in centrosome amplification:

- Overactivation of RhoA has been shown to promote centrosome amplification [11]
- ROCKII controls centrosome numbers through its interaction with phosphorylated nucleophosmin [12]
- ROCK and centrosome amplification have been involved in tumourigenesis [13]

Importantly, it had been previously unclear how signalling pathways affect these regulators of centrosome biology. Our manuscript shows, *for the first time*, that the RhoA/ROCK pathway is involved in centrosome amplification downstream of PI3K/AKT. This novelty has now been mentioned in the revised text in the discussion.

Through defects in cell division?

Our data show that cytokinesis failure is not the cause for centrosome amplification.

The data in Fig. 2c show that in the first round of cell division in the presence of p110 α ^{H1047R}, cells acquire centrosome amplification in parallel with the initiation of DNA duplication (as measured by BrdU incorporation) and thus before they go through cytokinesis (note that the doubling population time is ~24h in the mutant MEFs). Also, in Fig. 2d we show that there are no differences in the number of binucleated cells in WT and p110 α ^{H1047R} cultures.

All together, these data support the notion that p110 α ^{H1047R} expression leads to centrosome overduplication in the G1/S transition at the time of centriole duplication and rule out cytokinesis failure as the cause for centrosome amplification.

Through activation or repression of a member of the centriole duplication machinery?

Our data are compatible with an enhanced activation of the centriole duplication machinery by PI3K. Indeed, we find that p110 α ^{H1047R} expression leads to centrosome overduplication in the G1/S transition, by impinging on Npm-T199 phosphorylation and ROCKII activation, both of which have previously been shown to be involved in the initiation of centriole duplication.

We have now changed the text as shown below to make this point more clear:

Alternative mechanisms for generating extra centrosomes include cytokinesis failure or multiple cycles of centrosome duplication prior to cell division[14]. After induction of p110 α ^{H1047R} in MEFs that were growth-arrested by serum-deprivation, centrosome amplification emerged in parallel with the initiation of DNA

synthesis (as measured by BrdU incorporation) **in the G1/S transition**, and before the cells had gone through cytokinesis (Fig. 2c). These data show that centrosome amplification occurs during the first round of cell division upon p110 α^{H1047R} induction and **before the cells have divided**, ruling out cytokinesis failure as the primary cause of centrosome amplification early after p110 α^{H1047R} induction in MEFs. In agreement with this, we observed that p110 α^{H1047R} expression did not increase cytokinesis failure in these cells (Fig. 2d).

These observations are compatible **with centriole overduplication** being the underlying mechanism of p110 α^{H1047R} -induced centrosome amplification. **This is further supported by the observations from signalling studies, as described below.**

It should be investigated how de-regulation of signalling pathways lead to centrosome amplification.

We agree that this is an important question. We have therefore included new experimental data on the signalling between p110 α^{H1047R} and the centrosomes, by showing that overactivation of the PI3K pathway by p110 α^{H1047R} expression leads to overduplication of the centrosome in the G1/S transition through a pathway involving AKT, ROCKII and CDK2/Cyclin E-Nucleophosmin, as detailed below.

Cyclin E is known to bind to and activate the G1 phase CDK2 kinase and this CDK2-Cyclin E complex is known to phosphorylate the Nucleophosmin (Npm)/B23 protein on T199 in S-phase [15]. Npm/B23, in turn, is a protein previously implicated in centrosome amplification and has been shown to interact with the ROCKII Ser/Thr kinase, allowing its activation and initiation of centrosome duplication [12, 16].

We found that p110 α^{H1047R} MEFs displayed higher protein levels of Cyclin E (Fig. 3c), enhanced phosphorylation of Npm on T199 (a phosphorylation site of CDK2) as well as increased activation of ROCK II, as measured by the phosphorylation of its downstream target MLC on Ser19 (Fig. 3d). Both Npm-T199 and MLC-Ser19 phosphorylation levels were decreased by treatment either with a p110 α -selective inhibitor (Fig. 3d), a CDK2 inhibitor or an AKT specific inhibitor (Fig. 3e; Supplementary Fig. 8a). While the precise interplay and timing of these signalling events requires further investigation, our data are compatible with sustained activation of several key players involved in centrosome duplication, as shown in the scheme below.

New Fig. 3c

New Fig. 3e

New Supplementary Fig. 8a

We have now updated the abstract and result section accordingly, as shown below.

Abstract:

We report here that mutant *Pik3ca* induces two cancer-related phenomena, namely induction of centrosome amplification, through a pathway involving AKT, ROCKII and CDK2/Cyclin E-Nucleophosmin and, in parallel, an increase in tolerance to spontaneous genome doubling.

Results:

We found that p110 α ^{H1047R} MEFs displayed higher basal levels of RhoA-GTP, a known activator of ROCK (Fig. 3a). Increased basal RhoA-GTP levels and phosphorylated MLC (downstream effector of ROCK) were also observed in human *PIK3CA*^{H1047L} fibroblasts (Fig. 3b). Expression of p110 α ^{H1047R} was also found to lead to increased levels of Cyclin E (Fig. 3c). CDK2-Cyclin E complexes are known to phosphorylate the Nucleophosmin (Npm)/B23 protein in S-phase, which then interacts with ROCKII allowing its activation and initiation of centrosome duplication[12, 16]. In line with our conclusion that p110 α ^{H1047R} expression leads to centrosome overduplication in S-phase, the activation of ROCK (as measured by the phosphorylation of MLC on Ser19) and the phosphorylation of Npm on T199 (a phosphorylation site of CDK2)[15] were enhanced in p110 α mutant cells (Fig. 3d). Both these signalling events were inhibited either by a CDK2 inhibitor (Fig. 3e), and at longer time points, by a p110 α -selective inhibitor (Fig. 3d) or an AKT inhibitor (Fig. 3e and Supplementary Fig. 8a). The

precise interplay and timing of these signalling events remains to be determined. Our data are consistent with p110 α ^{H1047R} signalling to the centrosome at the time of initiation of duplication at the G1/S transition, through a mechanism involving AKT, ROCKII and CDK2/Cyclin E-Nucleophosmin, most likely leading to enhanced activation of the centriole duplication machinery.

2. The authors have a mouse model, that would allow them to investigate these questions in different organs and tissues. The choice of using MEFs, is not easy to understand as these cells very easily accumulate defects in the processes being analysed here.

As molecular/mechanistic investigation of processes at the cellular level in mice *in vivo* is highly challenging, we made use of primary cells in order to investigate the impact of oncogenic *Pik3ca*, early upon its expression. Mouse embryonic fibroblasts (MEFs) have been used for this type of studies in the past. We agree with this Reviewer that MEFs are prone to acquiring aberrations *in vitro* if no extra care is taken while culturing these cells, however, we have always used very early passage cells to avoid accumulation of stress and damage resulting from *in vitro* culture. Moreover, we have reproduced and verified our findings in many independent biological replicates, always including WT cells as control. Most importantly, we confirmed our centrosome amplification findings from MEFs in **primary keratinocytes (Fig. 1b, Fig. 5c and Supplementary Fig. 6c)** and in **tissues both of adult mice and embryos (Fig. 1c, Fig. 1d)**. For follow-up signalling and cell biology studies, we continued to use MEFs since we found keratinocyte primary cultures to be highly sensitive to *in vitro* stress, to show limited proliferative potential (low number of passages) and limited cell yield after isolation, which all severely limited the utility of these cells as a model to study cell biological processes in detail.

3. Which clustering mechanisms are being used in these cells, to sustain bipolar (or pseudo-bipolar spindle formation)?

This is an interesting question, but we have not explored the mechanism underlying the more efficient extra centrosome clustering in p110 α ^{H1047R} cells. Instead, we focused on understanding the biological consequences of the abnormal numbers and efficient clustering of centrosomes. In order to be able to perform such studies, it would not be possible to use the primary cells from our mice, but require the generation of purpose-built cell models with labelled reporters.

We would also like to re-iterate the reason for using primary cells: the early cell biological and signalling events set in motion by switching on mutant *PIK3CA* from its endogenous promoter and in the heterozygous state, as is always observed in malignant cells, remains unknown. Indeed, all studies to date have used non-inducible models of *PIK3CA* expression, with analysis performed long after these mutations have been established. Unfortunately, the benefits of using primary cells in which no other oncogenic lesions are present at the start of *Pik3ca* activation, technically limit our analysis of molecular mechanisms.

2. The authors determine the morphology of the mitotic spindle in what appears prometaphase like figures. These are very difficult to evaluate (Fig. 4C), but the correct way of doing this analysis is to analyse spindle morphology in anaphase figures. Multipolar configurations in prometaphase might be resolved in bipolar anaphases... The figure representing a "pseudo-bipolar" is not very different from the one showing a multipolar configuration...

The Reviewer raises an important point since the cell division outcome in cells with extra centrosomes can only be accurately assessed after anaphase, given that cells can still cluster extra centrosomes during metaphase to anaphase transition. However, the aim of our analysis was to show that even at the earlier stages of mitosis, when erroneous kinetochore-microtubule attachments are established (merotelic attachments), there is already a difference in the clustering efficiency between WT and p110 α ^{H1047R} cells. This suggests that the efficient clustering in early mitosis might prevent chromosome missegregation. In fact, in a study using the human RPE-1 cells [17] (which are unique in that they have a much higher basal incidence of centrosome

amplification compared to other cells), induction of extra centrosomes leads to almost 70% of multipolar prometa/metaphases and due to an efficient clustering, only 12% of those missegregate chromosomes in anaphase. Thus, it is likely that the high clustering efficiency we observed in early mitosis might prevent chromosome missegregation in anaphase. We realize we had not made this point sufficiently clear and have now updated the text as follows:

It has previously been shown that a high clustering efficiency in early mitosis might prevent chromosome missegregation in anaphase [17]. In order to check whether a similar mechanism could explain the lack of segregation errors upon p110 α ^{H1047R} induction, we performed a more detailed analysis of the different spindle configurations observed in mitotic cells with more than two centrosomes, both in prometaphase and metaphase. Whereas the normal mitotic spindle is bipolar, the spindles in cells with extra centrosomes are either pseudo-bipolar (with the extra centrosomes clustered at two opposite spindle poles) or multipolar (both schematically shown in Supplementary Fig. 9a). Bipolar and pseudo-bipolar configurations are expected to allow cells to complete mitosis, whereas multipolar spindles need to resolve into pseudo-bipolar spindles for cells to progress through mitosis [17]. We focused on cells with centrosome amplification going through mitosis (note that we observed ~10% centrosome amplification in WT and ~30% in p110 α ^{H1047R} populations). Analysis of mitotic spindles in prometaphase and metaphase in these cells revealed a higher incidence of pseudo-bipolar configurations in p110 α ^{H1047R} than in WT cells (Fig. 4c). This suggests that even at earlier stages of mitosis, when erroneous kinetochore-microtubule attachments are established (merotelic attachments), there is already a difference in the clustering efficiency of extra-centrosomes between WT and p110 α ^{H1047R} MEFs, a finding that could explain the lack of differences in the frequencies of segregation errors observed between WT and *Pik3ca* mutant populations, in spite of the higher frequency of cells with centrosome amplification observed in the latter.

Based on this Referee comment, we have now reanalysed our original data and removed the anaphase figures (of which there were very few) from our analysis, and focused instead only on clustering efficiency during prometa/metaphase. The results of this updated analysis are shown in the figure below. In fact, removing the anaphases from the analysis reveals an even higher difference between WT and p110 α ^{H1047R} cells. We have updated the graph in Fig. 4c, and now also show a pseudo-bipolar metaphase in which the clustering of extra-centrosomes is more obvious.

3. It is difficult to understand the analysis of segregation errors in WT and p110 α ^{H1047R} cells. If control cells have increased multipolarity, how to explain that they have the same level of segregation errors found in p110 α ^{H1047R} cells?

We understand the point raised by the Reviewer and realise that the data, as represented in Fig. 4c, might have been somewhat confusing, by giving the impression that there are more multipolar cells in the WT population, which is not the case.

We found that the absolute number of multipolar mitosis is not higher in WT cells than in p110 α ^{H1047R} cells. However, when specifically focusing on cells with extra centrosomes that are going through mitosis, the percentage of multipolar spindles is higher in the WT cells (as was represented in Fig. 4c). Taking into account that at any time point in an exponentially growing culture, the number of cells with centrosome amplification (including all phases of the cell cycle) is ~10% in WT and ~30% in p110 α ^{H1047R} cells (Fig. 1b), and assuming that all those cells will divide, then the 79% of multipolarity shown in Fig. 4c refers to the 10% of cells with extra centrosomes in WT cells and the 46% multipolarity observed in the p110 α ^{H1047R} cells refers to the 30% of cells with extra centrosomes. This would translate in to ~8 WT cells with multipolar spindles *versus* ~14 p110 α ^{H1047R} cells with multipolar spindles in every 100 cells. So, if anything, the total number of multipolar mitosis is slightly higher in the mutant cells. Despite this, we did not observe higher chromosome segregation errors in mutant cells, most likely due to a more efficient ability of the mutant cells to cluster extra centrosomes.

In the revised manuscript, we have updated the text as follows:

Indeed, when considering the absolute numbers, the total number of the multipolar spindles in early mitosis were slightly higher in the mutant population (~8 WT cells with multipolar spindles *versus* ~14 p110 α ^{H1047R} cells with multipolar spindles in every 100 cells), yet we did not observe higher frequencies of single chromosome segregation errors in the mutant cells. Altogether these findings points toward a higher clustering efficiency of extra centrosomes in p110 α ^{H1047R}-expressing cells.

4. The p110 α ^{H1047R} appears to facilitate the tolerance to tetraploidization even in the presence of WT p53. This is a really interesting observation, but how this is achieved should be determined.

We agree that this is a very interesting question: understanding tolerance mechanisms to tetraploidization are amongst the biggest questions/mysteries in this new field, and we believe that addressing these questions as a sub-part of the current dataset is beyond the scope of this manuscript.

Our data are consistent with the finding that hyperactivation of growth factor signalling, in our case by sustained signalling by the downstream effector p110 α , can overcome the proliferative blockade of tetraploid cells [1]. We have made it now more clear in the text (by inserting the text below) that it is most likely that the well-established pro-survival signalling induced by constitutive activation of AKT in mutant *Pik3ca* cells leads to increased resistance to the stress induced by tetraploidization.

'More specifically, it is most likely that the well-established pro-survival signalling induced by constitutive activation of AKT in mutant *Pik3ca* cells leads to increased resistance to the stress induced by tetraploidization.'

5- It would be useful for the reader to know how many cells from how many different animals were used in each experimental condition.

We have now included additional information in figure legends to address this point (text highlighted in yellow in revised manuscript).

Reviewer #3

This Reviewer carefully summarizes our findings, and comments that ‘Although the findings in this manuscript are significant, several concerns remain to be addressed’.

1. Three hotspots mutations are within the helical and kinase domains of *pi3kca*: H1047L/R, E542K/Q and E545K. By examining publically available data through cBioportal (Gao et al. *Sci. Signal.* 2013 & Cerami et al. *Cancer Discov.* 2012), E545K seems to be the most recurrent in all cancer types in general as well as in colon cancer alone. However, in breast cancer, 1047 is the most recurrent. It is unclear why the authors chose to work only on one of these hotspots, especially that they focused on colon cancer when inducing *pik3ca* mutant with intestine-specific heterozygous deletion of the Apc.

We have to admit that the basis for these conclusions by the Reviewer is unclear to us, as this is not what we find ourselves in the **cBioPortal analysis** of *PIK3CA* mutations in the helical domain (E542 and E545) and kinase domain (H1047) of *PIK3CA*, as shown in the Table below (graphical representations of these BioPortal analysis are shown in Appendices I-III):

	Total number of PIK3CA missense mutations	Number of E545A and other E545 missense mutations (% of total mutations)	Number of H1047R and other H1047 missense mutations (% of total mutations)
All cancer types	3365	488 (15)	701 (21)
Breast cancer	1848	244 (13)	566 (31)
Colorectal cancer	26	6 (23)	5 (19)

Table 1. Number of cancers with *PIK3CA* missense mutations and frequencies of helical domain (E545A/X) and kinase domain (H1047R/X) mutations in *PIK3CA* (source cBioportal <http://www.cbioportal.org/index.do>).

Analysis of all cancer types (a total of 3365 missense mutations) reveals that mutations in the *PIK3CA* kinase domain are the most frequent, including in breast cancer, and more prevalent than the helical domain *PIK3CA* mutations. The number of *PIK3CA* mutations reported in cBioportal for colorectal cancer are very low (26 cases of missense mutations), with no major differences in the frequency of E545K *versus* H1047R mutations.

Similar conclusions have been reached in a recent analysis of the **COSMIC database** of all *PIK3CA* mutations by other authors, shown in Fig. 1a of Dogruluk *et al.* [18]:

Since mutations in the kinase domain of *PIK3CA* are the most frequent in cancer, we decided to model this mutation in mice and cells, focusing on one of the hot-spot mutations in this locus, namely H1047R.

It would be indeed of interest to explore whether our findings also hold true for helical domain mutations, such as E545K. However, it is not straightforward, from a mouse gene-targeting perspective, to create 'clean' conditional alleles to model point mutations in the middle of a gene (as opposed to the 3' end of the gene, for example to conditionally generate the C-terminal H1047R mutation as in our study).

2. Fig. 1: Could the authors provide a figure that shows the bands (pAKTS473) more clearly?

We believe the bands shown in Fig. 1b are clear and very well document the increase in Ser473 phosphorylation of AKT upon *PIK3CA*^{H1047R} expression.

3. Please add abbreviations to figures: (Error bars, S.D.; WT, wild-type).

We have followed the 'in house' *Nature Communications* style in preparing our manuscript and figure legends.

4. The authors conclude that tamoxifen-induced expression of *pik3ca* mutant with intestine-specific heterozygous deletion of the *Apc* tumor suppressor gene accelerated the progression of colon cancer; suggesting that the mutant *pik3ca* is a weak oncogene on its own. To make the assumption that the mutant *pik3ca* is a weak oncogene on its own, other cancer types and other cancer-promoting genetic lesions should be considered. Also the most recurrent mutation in colon cancer is E545K which was not included in the experiment.

We believe that there is some misunderstanding at play here: our conclusion that *Pik3ca*^{H1047R} is a weak oncogene on its own is based on long-term observation of mice carrying just this mutation, and not on the observation that *Pik3ca*^{H1047R} cooperates with *Apc* loss in colon carcinogenesis. We therefore do not see the rationale to perform crosses with other mouse lines with cancer-predisposing genes (including the E545K mutation) to reach this conclusion.

5. The authors mention that mutations in *PIK3CA* and *TP53* are mutually exclusive in breast cancer; but did not indicate if the analysis showed significance in other/all cancer types.

When we considered all cancer types, we did not observe a significant tendency for mutual exclusivity.

Also they did not indicate if the analysis showed significance when only looking at clonal mutations which is an indicative of *pik3ca* mutation being an early event in the evolution of cancer.

We thank the Reviewer for this important suggestion. When restricting the analysis to clonal mutations in breast cancer, the significant tendency for mutual exclusivity between *TP53* and *PIK3CA* remained, consistent with the Reviewer's suggestion that *PIK3CA* mutation is an early event in the evolution of breast cancer. Given that we had already drawn this conclusion in the manuscript (focusing on all ER-positive breast cancers), we have not further elaborated on this in the revised manuscript.

6. Given the importance of combined *PIK3CA* mutation and PTEN expression, is there a correlation between the presence of *pik3ca* mutations and the expression of PTEN?

We have not found a correlation between *PIK3CA*^{H1047R} mutation and PTEN expression [19] as shown below.

Western blot analysis of MEF lysates 48 h after 4-OHT treatment (4 independent MEFs are analysed per genotype, each lane shows an independent MEF preparation).

This is an interesting question but, in our opinion, not entirely relevant to the current manuscript. We have therefore not included or mentioned these data in the revised manuscript.

7. The authors hypothesized a mutation in *PIK3CA* is an early event in the evolution of breast cancer. No rationale is given to why they picked breast cancer. Is it because the *pik3ca* mutations is the highest reported? Is the same seen in other solid cancers; for example, colon, brain, liver, stomach and lung? Which also have high *pik3ca* mutation frequency?

Our analysis was focused on breast cancer for a number of reasons. Firstly, of the nine cancer types where we were able to perform clonal analysis, breast cancer represented the largest cohort, with over 900 tumour samples. Secondly, *PIK3CA* kinase domain mutations occur at the highest prevalence in breast cancer compared to the remaining cancer types. Finally, in breast cancer, *PIK3CA* mutations showed a significant tendency to be early events. We believe we had argued the case to study breast cancer sufficiently well in the manuscript and have not updated the revised version.

8. Missing references in the introduction. For example: 'Thus far, the oncogenic potential of PI3K has largely been attributed to its role in stimulating processes such as cell survival and proliferation, spurring the development of inhibitors of the PI3K pathway as anti-cancer agents.'

We have now included the following additional references:

- PI3K signalling: the path to discovery and understanding [20]
- The emerging mechanisms of isoform-specific PI3K signalling [21]
- The phosphoinositide 3-kinase pathway [22]

REFERENCES

1. Ganem, N.J., et al., *Cytokinesis failure triggers hippo tumor suppressor pathway activation*. Cell, 2014. **158**(4): p. 833-48.
2. Hare, L.M., et al., *Heterozygous expression of the oncogenic *Pik3ca*(H1047R) mutation during murine development results in fatal embryonic and extraembryonic defects*. Dev Biol, 2015. **404**(1): p. 14-26.
3. Hare, L.M., et al., *Physiological expression of the PI3K-activating mutation *Pik3ca*(H1047R) combines with *Apc* loss to promote development of invasive intestinal adenocarcinomas in mice*. Biochem J, 2014. **458**(2): p. 251-8.
4. Adams, J.R., et al., *Cooperation between *Pik3ca* and *p53* mutations in mouse mammary tumor formation*. Cancer Res, 2011. **71**(7): p. 2706-17.
5. Kinross, K.M., et al., *An activating *Pik3ca* mutation coupled with *Pten* loss is sufficient to initiate ovarian tumorigenesis in mice*. J Clin Invest, 2012. **122**(2): p. 553-7.
6. Van Keymeulen, A., et al., *Reactivation of multipotency by oncogenic PIK3CA induces breast tumour heterogeneity*. Nature, 2015. **525**(7567): p. 119-23.
7. Borel, F., et al., *Multiple centrosomes arise from tetraploidy checkpoint failure and mitotic centrosome clusters in *p53* and *RB* pocket protein-compromised cells*. Proc Natl Acad Sci U S A, 2002. **99**(15): p. 9819-24.
8. Storchova, Z. and C. Kuffer, *The consequences of tetraploidy and aneuploidy*. J Cell Sci, 2008. **121**(Pt 23): p. 3859-66.
9. Laughney, A.M., et al., *Dynamics of tumor heterogeneity derived from clonal karyotypic evolution*. Cell Rep, 2015. **12**(5): p. 809-20.
10. Zack, T.I., et al., *Pan-cancer patterns of somatic copy number alteration*. Nat Genet, 2013. **45**(10): p. 1134-40.
11. Kanai, M., et al., **RhoA* and *RhoC* are both required for the ROCK II-dependent promotion of centrosome duplication*. Oncogene, 2010. **29**(45): p. 6040-50.
12. Ma, Z., et al., *Interaction between ROCK II and nucleophosmin/B23 in the regulation of centrosome duplication*. Mol Cell Biol, 2006. **26**(23): p. 9016-34.
13. Ferretti, R., et al., **Morgana/chp-1*, a ROCK inhibitor involved in centrosome duplication and tumorigenesis*. Dev Cell, 2010. **18**(3): p. 486-95.
14. Fukasawa, K., *Oncogenes and tumour suppressors take on centrosomes*. Nat Rev Cancer, 2007. **7**(12): p. 911-24.
15. Tokuyama, Y., et al., *Specific phosphorylation of nucleophosmin on Thr(199) by cyclin-dependent kinase 2-cyclin E and its role in centrosome duplication*. J Biol Chem, 2001. **276**(24): p. 21529-37.
16. Hanashiro, K., M. Brancaccio, and K. Fukasawa, *Activated ROCK II by-passes the requirement of the CDK2 activity for centrosome duplication and amplification*. Oncogene, 2011. **30**(19): p. 2188-97.
17. Ganem, N.J., S.A. Godinho, and D. Pellman, *A mechanism linking extra centrosomes to chromosomal instability*. Nature, 2009. **460**(7252): p. 278-82.
18. Dogruluk, T., et al., *Identification of Variant-Specific Functions of PIK3CA by Rapid Phenotyping of Rare Mutations*. Cancer Res, 2015. **75**(24): p. 5341-54.
19. Moniz, L.S., et al., *Phosphoproteomic analysis identifies the cytosolic 5' 3'-Nucleotidase NT5C as an effector of PI3K-mediated actin dynamics*. Scientific Reports (under review).
20. Vanhaesebroeck, B., L. Stephens, and P. Hawkins, *PI3K signalling: the path to discovery and understanding*. Nat Rev Mol Cell Biol, 2012. **13**(3): p. 195-203.
21. Vanhaesebroeck, B., et al., *The emerging mechanisms of isoform-specific PI3K signalling*. Nat Rev Mol Cell Biol, 2010. **11**(5): p. 329-41.
22. Cantley, L.C., *The phosphoinositide 3-kinase pathway*. Science, 2002. **296**(5573): p. 1655-7.

APPENDIX I

ALL CANCERS:

a. 488 cases out of 3365 (15%) with mutations in the helical domain (E545A/D/G/K/Q/R/V)

b. 701 cases out of 3365 (21%) with mutations in the kinase domain (H1047L/Q/R/Y)

APPENDIX II

BREAST CANCER

a. 244 cases out of 1848 (13%) with mutations in the helical domain (E545A/D/G/K/R)

b. 566 cases out of 1848 (31%) with mutations in the kinase domain (H1047R/Q/R/Y)

APPENDIX III

COLON CANCER

a. 6 cases out of 26 (6%) with mutations in the helical domain (E545A/G/K)

b. 5 cases out of 26 (5%) with mutations in the kinase domain (H1047L/R)

Reviewers' comments:

Reviewer #1 (Remarks to the Author):

In this manuscript, Berenjano et al. discovered novel roles of oncogenic PIK3CA in centrosome amplification and tetraploidy tolerance. Using a combination of in vivo and in vitro models, the authors showed that expression of p110 α H1074R at endogenous levels promotes centrosome amplification through activation of the Akt/ROCK pathways. In addition, the authors provide evidence for p110 α H1074R playing a role on tolerating tetraploidization. Importantly, these findings point toward previously unappreciated roles for this oncogenic mutation in cancer progression

Berenjano et al. successfully addressed most of my and other reviewers' comments. As a result, the revised version of their manuscript is significantly improved. However, before fully endorsing this revised manuscript for publication in Nature Communications, a few issues remain confusing:

1- In Figure 1b, Berenjano et al. show that 30% of MEFs expressing the H1047R allele have more than 2 centrosomes. Figure 5a shows that as much as 50% of these MEFs are aneuploid. Do all aneuploid/tetraploid cells have increased number of centrosomes?

If so, these two phenotypes must be functionally coupled. How do the authors conclude that the results obtained in the phenotypic analysis of the MEFs (better growth, colony formation, etc) are due to increased number of centrosomes and not due to aneuploidy?

2. Several results presented in the manuscript seem to favor the hypothesis that centrosome amplification and tetraploidy tolerance are independent of each other. However, Berenjano et al. now show that inhibition of ROCK signaling which prevents centrosome overduplication also impedes tetraploidization (Figures 3f and 6a). It makes more sense that centrosome duplication leads to tetraploidization.

Although acknowledged by the authors in their response, it is very confusing that no differences in DNA content in the mutant cells are observed by FACS given that as much as 50% of the cells are tetraploid (Figure 5). In addition, despite the fact that Figure 4 shows that chromosome missegregation is not increased, chromosome counts shown in Figures 5, 6 and supplemental Figure 10 show significant numbers of aneuploids (not fully tetraploids). If these cells come from unstable tetraploids, chromosome missegregation events must take place.

3. Another point that could be better explained is the response of tetraploid cells to drugs that inhibit PI3K signaling. The data suggest that PI3K signaling plays a key role on tolerating tetraploidy yet this pathway is dispensable once tetraploids form. Are these cells more sensitive to the inhibition of PI3K signaling? Shouldn't they be?

Reviewer #3 (Remarks to the Author):

I have read the revised manuscript by Inma M. Berenjano and colleagues. The authors have done a good job addressing the reviewers comments and updating the manuscript where required. In this paper, the authors used Flp recombinase-based knock in mouse models to activate the Pik3caH1047R hotspot mutation from its endogenous locus and in the heterozygous state. They were able to show that heterozygously expressed mutant pik3ca alone did not develop cancer. However, tamaxifen-induced expression of pik3ca mutant with intestine-specific heterozygous deletion of the Apc tumor suppressor gene accelerated the progression of cancer and further proved that the mutant pik3ca is a weak oncogene on its own. To better understand the biological effects of mutant pik3ca, the authors used both primary MEFs and tumor-derived cells. Their data show an induction of centrosome amplification through the AKT, ROCKII and CDK2/Cyclin E-

Nucleophosmin pathways and an increase in tolerance to spontaneous genome doubling. However, centrosome amplification did not lead to more segregation errors. In spite of the unaltered frequency of segregation errors in the mutant *pik3ca*, analysis of metaphase chromosome spreads showed an increase in aneuploidy cells pointing towards the possibility of genome doubling, suggesting an alternative tolerance routes to genome doubling than the p53 pathway. This was further proved by genetic analysis showing that mutations in *pik3ca* and p53 have a strong tendency to be mutually exclusive suggesting that the role of *pik3ca* mutation as a potential tolerance mechanism for genome doubling in breast cancer is independent of the p53 pathway.

This revised manuscript addressed several key deficiencies. The authors performed signaling studies which showed that mutant *Pik3ca* induces centrosome amplification through a pathway involving AKT, ROCKII and CDK2/Cyclin E-Nucleophosmin. They also included an improved analysis of the difference in the clustering efficiency of extra-centrosomes between WT and p110 α H1047R MEFs. This was improved by removing the anaphases from the analysis which revealed a higher difference between the WT and p110 α cells.

A few concerns remain to be addressed:

1- Although the authors rationalize why they focus their attention on the characterization of the H1047 mutation and explain the difficulty to model other mutations namely helical domain mutations in the rebuttal, I still believe that the paper would have a much greater impact if a larger number of variants were characterized. In addition, please explicitly state the high frequency of these mutations in cancers and please provide p values for the likelihood occurrence of the mutations based on calculations that considers: 1) gene size 2) coverage of the particular gene 3) background mutation rate.

2- The authors indicated that there are no tools available to selectively interfere with centrosome amplification to test its interaction with other cancer-promoting genetic lesions. However, there was no attempt made to at least explain what the authors think the functional consequence of the observed centrosome amplification may be. Please elaborate on this point.

Minor Points:

3- There are still missing references in the introduction: 'Several Cre recombinase-based mouse models have been created to explore the role of mutated p110 α in cancer'

4- Figure 3 e needs fixing- panels are not aligned.

5- In the text the authors referred to supplementary Fig 5 f as Fig 5f: Enhanced Akt phosphorylation was also observed in primary fibroblasts from human fibro-adipose overgrowth syndrome patients with mosaic, heterozygous expression of the PIK3CAH1047L mutation²¹ (Fig. 5f). Also Fig 6 d should be supplementary fig 6d in the following: 'and in MCF-10A human mammary epithelial cells transfected with p110 α H1047R (Fig. 6d).

6- Figure 6 is missing (D).

Reviewers' comments

Reviewer #1 (Remarks to the Author):

In this manuscript, Berenjeno *et al.* discovered novel roles of oncogenic *PIK3CA* in centrosome amplification and tetraploidy tolerance. Using a combination of *in vivo* and *in vitro* models, the authors showed that expression of p110 α ^{H1074R} at endogenous levels promotes centrosome amplification through activation of the Akt/ROCK pathways. In addition, the authors provide evidence for p110 α ^{H1074R} playing a role on tolerating tetraploidization. Importantly, these findings point toward previously unappreciated roles for this oncogenic mutation in cancer progression.

Berenjeno *et al.* successfully addressed most of my and other reviewers' comments. As a result, the revised version of their manuscript is significantly improved. However, before fully endorsing this revised manuscript for publication in *Nature Communications*, a few issues remain confusing:

We appreciate this Referee's positive comments on our revised manuscript, and agree that some issues in the manuscript may have come across as somewhat 'confusing'. We believe this has now been rectified in the revised manuscript, as explained in more detail below.

1. In Fig. 1b, Berenjeno *et al.* show that 30% of MEFs expressing the H1047R allele have more than 2 centromeres (authors' comment: this should be 'centrosomes' rather than 'centromeres'). Fig. 5a shows that as much as 50% of these MEFs are aneuploid. Do all aneuploid/tetraploid cells have increased number of centrosomes?

We do not know whether all the aneuploid cells in Fig. 5 also have extra-numbers of centrosomes since these phenotypes were assessed using two different techniques (immunofluorescence and metaphase spreads) that do not allow monitoring of these two parameters concomitantly.

In general, it is known that tetraploid cells derived from cells that spontaneously acquire a double genome, might initially have 4 centrosomes. However later on, if these cells manage to divide, the daughter cells, which could become aneuploid, might end up with a normal number of centrosomes.

If so, these two phenotypes must be functionally coupled.

For an answer to this comment, we refer to our reply to Comment 2 of this Referee below.

How do the authors conclude that the results obtained in the phenotypic analysis of the MEFs (better growth, colony formation, etc) are due to increased number of centrosomes and not due to aneuploidy?

We have to correct the Referee as we had not drawn this conclusion anywhere in the manuscript.

*What we can conclude, however, is the important role of ROCK in all new biological roles of *PIK3CA* mutation reported in our manuscript, suggesting that all these biological activities are interconnected to drive *PIK3CA*^{H1047R}-induced cell-transformation, as shown in our interpretation in Fig 7.*

We had previously shown that ROCK inhibition blocks PIK3CA^{H1047R}-induced centrosome and chromosomal abnormalities, pointing towards a potential role of these two phenomena in cellular transformation induced by PIK3CA^{H1047R}. What we had not tested at the time was a possible role of ROCK in cell transformation. We have now carried out this experiment and find that treatment with two different ROCK inhibitors (Y27632 and H1152) clearly decreases PIK3CA^{H1047R}-induced cell transformation on primary MEFs (new Supplementary Fig. 9b), further strengthening the interconnection between centrosome amplification, tolerance to tetraploidization, aneuploidy and cell transformation.

We have now referred to these ROCK data at the beginning of the revised discussion.

New Supplementary Fig. 9b

2. Several results presented in the manuscript seem to favor the hypothesis that centrosome amplification and tetraploidy tolerance are independent of each other. However, Berenjano *et al.* now show that inhibition of ROCK signaling which prevents centrosome overduplication also impedes tetraploidization (Fig. 3f and 6a). It makes more sense that centrosome duplication leads to tetraploidization.

We reply to this Reviewer's comments 1 and 2 in several steps, commenting on each new biological activity induced by p110 α reported in our MS:

A. Does centrosome amplification lead to tetraploidization?

In order to check whether centrosome amplification led to tetraploidization through cytokinesis failure, we performed live imaging by time-lapse microscopy of MEFs expressing GFP-tagged centrin2. This analysis revealed that, of the cells with multiple centrosomes (Fig. 1b), most of the p110 α ^{H1047R} cells were able to exit mitosis efficiently, with only 6% of mutant cells failing to do so, compared to a 35% failure rate in WT cells (Suppl. Fig. 4a and Suppl. Fig. 9b and Suppl. Videos 1, 2 and 3). These data show that compared to WT cells, cells expressing p110 α ^{H1047R} are more proficient at completing mitosis with extra centrosomes and that there is no increase in cytokinesis failure in mutant cells with an extra number of centrosomes. We therefore conclude that centrosome amplification in our model does not lead to the observed tetraploidization. It remains possible though that centrosome amplification could be relevant to tolerance to tetraploidization which, as previously mentioned in the text, happens spontaneously in cell culture, and in MEFs in particular.

B. Does centrosome amplification lead to an increase in aneuploidy?

With regards to centrosome amplification contributing to aneuploidy due to chromosome segregation errors, it is informative to consider the study by Ganem et al. (Nature 2009 Jul 9;460(7252):278-82). Indeed, using the human RPE-1 cell model which shows one of the highest reported levels of cells with extra centrosomes and multipolar spindles, the frequency of missegregation of chromosomes was found to be very low. More specifically, these authors find that in RPE-1 cells in which 70% of the cells in the population have extra centrosomes and multipolar spindles, approximately only 12% of these (the equivalent of ~8.4 cells out of 70) missegregate a chromosome.

In our MEF study, we find 30% of centrosome amplification in p110 α ^{H1047R} MEFs. Assuming that all the cells with centrosome amplification go through mitosis, and assuming a similar situation as in Ganem's study, we would expect that only 3.6 cells out of 100 (12% of the 30 cells with centrosome amplification in 100) would missegregate a chromosome. However, this number might be even lower as we have found that p110 α ^{H1047R} MEFs are very efficient at clustering extra centrosomes. It is therefore possible that the number of cells with centrosome amplification/multipolar spindles (potentially missegregating chromosomes) in our MEF cell model is simply too low to enable us to observe differences in chromosome missegregation between WT and p110 α ^{H1047R} cells.

Taking this into account, we therefore agree that **we cannot completely exclude the possibility of a higher chromosome mis-segregation incidence in p110 α ^{H1047R} cells and a possible contribution to aneuploidy** in the context of this mutation. In this scenario, both chromosome segregation errors and tolerance to genome doubling/tetraploidization would cooperate to lead to higher incidence of aneuploidy in the presence of p110 α ^{H1047R} mutation, as shown in the Summary Figure 7.

We have now updated the discussion to reflect this updated interpretation of our data.

C. Inhibition of ROCK signalling prevents centrosome overduplication and also impedes tetraploidization: link between centrosome amplification and tetraploidization.

We agree with this Reviewer that the observation that if ROCK inhibition blocks centrosome amplification and tetraploidisation/aneuploidy, there could be a potential functional link between them. However, we do not think that such a link is direct since mutant cells with centrosome amplification do not go through cytokinesis failure/tetraploidization but cluster their centrosomes efficiently and go through cell division.

We have now included new data showing that ROCK inhibition (using the Y-27632 inhibitor) prevents centrosome amplification within this first round of division before cytokinesis takes place (New Supplementary Fig. 9a), further supporting a role of ROCK in the control of centriole duplication downstream of p110 α . ROCK inhibition also prevents tetraploidization, as shown in Fig.6a. We favour the interpretation of these data that **ROCK activation downstream of oncogenic PIK3CA leads to centrosome amplification (through increased signalling to the centriole duplication machinery) and in parallel is involved in the tolerance to genome doubling phenotype.**

New Supplementary Fig. 9a.

At present, we have no data that would allow us to conclude whether these two ROCK-controlled phenomena downstream of $PIK3CA^{H1047R}$ are related or not. We have now made this clear at the beginning of the revised discussion and revised Fig. 7 accordingly, as shown below:

New Figure 7.

Although acknowledged by the authors in their response, it is very confusing that no differences in DNA content in the mutant cells are observed by FACS given that as much as 50% of the cells are tetraploid (Fig. 5).

We agree with this Reviewer. The level of tetraploidy/aneuploidy detected in our experiments of metaphase chromosome spreads should also be detected by FACS analysis. This is something we do not understand at the moment.

In addition, despite the fact that Fig. 4 shows that chromosome missegregation is not increased, chromosome counts shown in Figs 5, 6 and Supplemental Fig. 10 show significant numbers of aneuploids (not fully tetraploids). If these cells come from unstable tetraploids, chromosome missegregation events must take place.

We agree with this Reviewer, and have updated the discussion accordingly. As clarified in question 2 above, it is possible that the numbers of cells with centrosome amplification/multipolar spindles (potentially missegregating chromosomes) are simply too low to enable us to observe differences in chromosome missegregation between WT and p110 α ^{H1047R} cells. Taking this into account, we agree that we cannot exclude the possibility of a higher chromosome missegregation incidence in p110 α ^{H1047R} cells and a possible contribution to aneuploidy in the context of this mutation. In this scenario, both chromosome segregation errors and tolerance to genome doubling/tetraploidization would cooperate to lead to higher incidence of aneuploidy in the presence of p110 α ^{H1047R} mutations.

3. Another point that could be better explained is the response of tetraploid cells to drugs that inhibit PI3K signaling. The data suggest that PI3K signaling plays a key role on tolerating tetraploidy yet this pathway is dispensable once tetraploids form. Are these cells more sensitive to the inhibition of PI3K signaling? Shouldn't they be?

This is a very interesting point. Indeed, our data suggest that overactivation of the PI3K pathway helps cells to tolerate genome doubling. However, we have found that the PI3K pathway is dispensable once the PIK3CA^{H1047R}-induced tetraploids have evolved and acquired further alterations, such as p53 loss, amongst others. We had commented on this in our discussion as follows:

'Our data show that pharmacological inhibition of the PI3K/ROCK pathway can reduce the fraction of cells that has centrosome amplification within a population, but cannot revert the tetraploidy-derived aneuploidy facilitated by the presence of Pik3ca^{H1047R}, once it has been established. This could help to explain the currently limited therapeutic impact of PI3K-targeted therapies in cancer and the observation that PIK3CA mutation is not, on its own, a clear predictor of sensitivity to PI3K pathway inhibitors⁵⁰.

We agree that we could have made this point more clearly, and have updated the discussion as follows:

'Our data show that pharmacological inhibition of the PI3K/ROCK pathway can reduce the fraction of cells that has centrosome amplification within a population of transformed cells, long after the appearance of the phenotype. However, in spite of preventing tetraploidization early after p110 α ^{H1047R} expression, inhibition of the PI3K pathway cannot revert the tetraploidy-derived aneuploidy facilitated by the presence

of *Pik3ca*^{H1047R}, once the tetraploids have been stably established. This could help to explain the currently limited therapeutic impact of PI3K-targeted therapies in cancer and the observation that *PIK3CA* mutation is not, on its own, a clear predictor of sensitivity to PI3K pathway inhibitors⁵⁰.

Below we show additional data (New Supplementary Figure 17) from an experiment in which 2 diploid and 4 tetraploid mutant *PIK3CA*-transformed clones were treated with a p110 α inhibitor (A66) for 3 days. We did not observe a relative increase in sensitivity in the tetraploid clones compared to the diploids, further supporting the idea that overactivation of the PI3K pathway is needed to help cells tolerate genome doubling early after induction but not in a transformed context when more genetic alterations have been acquired.

New Supplementary Fig. 17

Reviewer #3 (Remarks to the Author):

The authors have done a good job addressing the reviewers comments and updating the manuscript where required. In this paper, the authors used Flp recombinase-based knock in mouse models to activate the *Pik3ca*^{H1047R} hotspot mutation from its endogenous locus and in the heterozygous state. They were able to show that heterozygously expressed mutant *pik3ca* alone did not develop cancer. However, tamaxifen-induced expression of *pik3ca* mutant with intestine-specific heterozygous deletion of the *Apc* tumor suppressor gene accelerated the progression of cancer and further proved that the mutant *pik3ca* is a weak oncogene on its own. To better understand the biological effects of mutant *pik3ca*, the authors used both primary MEFs and tumor-derived cells. Their data show an induction of centrosome amplification through the AKT,

ROCKII and CDK2/Cyclin E-Nucleophosmin pathways and an increase in tolerance to spontaneous genome doubling. However, centrosome amplification did not lead to more segregation errors. In spite of the unaltered frequency of segregation errors in the mutant *pik3ca*, analysis of metaphase chromosome spreads showed an increase in aneuploidy cells pointing towards the possibility of genome doubling, suggesting an alternative tolerance routes to genome doubling than the p53 pathway. This was further proved by genetic analysis showing that mutations in *pik3ca* and p53 have a strong tendency to be mutually exclusive suggesting that the role of *pik3ca* mutation as a potential tolerance mechanism for genome doubling in breast cancer is independent of the p53 pathway.

This revised manuscript addressed several key deficiencies. The authors performed signaling studies which showed that mutant *Pik3ca* induces centrosome amplification through a pathway involving AKT, ROCKII and CDK2/Cyclin E-Nucleophosmin. They also included an improved analysis of the difference in the clustering efficiency of extra-centrosomes between WT and p110 α ^{H1047R} MEFs. This was improved by removing the anaphases from the analysis which revealed a higher difference between the WT and p110 α cells.

A few concerns remain to be addressed:

1. Although the authors rationalize why they focus their attention on the characterization of the H1047 mutation and explain the difficulty to model other mutations namely helical domain mutations in the rebuttal, I still believe that the paper would have a much greater impact if a larger number of variants were characterized.

We have now tested the impact of other cancer-associated modes of PI3K pathway activation, in addition to the PIK3CA^{H1047R} mutation, namely expression of the helical E545K mutant of PIK3CA, the D560Y mutant of PIK3R1 (the p85 regulatory subunit of PIK3CA) as well as overexpression of PIK3CA^{WT} (reflecting PIK3CA amplification) in human MCF-10A cells, all of which were found to lead to centrosome amplification (Supplementary Fig. 6d).

In addition, please explicitly state the high frequency of these mutations in cancers and please provide p-values for the likelihood occurrence of the mutations based on calculations that considers: 1) gene size 2) coverage of the particular gene 3) background mutation rate.

It is not clear to us what this Referee means with this comment, and trust that the additional experiments shown in Fig. 6d address this comment.

2. The authors indicated that there are no tools available to selectively interfere with centrosome amplification to test its interaction with other cancer-promoting genetic lesions. However, there was no attempt made to at least explain what the authors think the functional consequence of the observed centrosome amplification may be. Please elaborate on this point.

We had mentioned the potential functional consequences of centrosome amplification in the discussion, as shown below. We believe that these are sensible speculations and would prefer not to speculate further in order not to over-interpret our results.

*'It is important to stress that our study focused on assessing the cellular impact of *Pik3ca*^{H1047R} activation in a non-transformed context. It is possible that deregulation of centrosome biology by oncogenic *Pik3ca* in a complex genetic background, as in cancer, might increase the frequency of chromosome segregation errors in cells. It is also possible that other cellular processes that are known to be regulated by*

centrosomes, such as cell polarity invasion and metastasis [reviewed in Refs.^{10, 42}], will be affected by $Pik3ca^{H1047R}$ -induced alterations in centrosome numbers, as these processes are known to be controlled by $PI3K^{43, 44}$.

Minor Points:

3. There are still missing references in the introduction: ‘Several Cre recombinase-based mouse models have been created to explore the role of mutated p110 α in cancer.

We have now included two more references in this paragraph, as follows: ‘Several Cre recombinase-based mouse models have been created to explore the role of mutated p110 α in cancer⁷. Interestingly, whereas transgenic expression of mutant $Pik3ca$ has been found to be an effective inducer of cancer⁸.’

4. Fig. 3e needs fixing - panels are not aligned.

We have now aligned the panels.

5. In the text the authors referred to supplementary Fig 5f as Fig 5f: ‘Enhanced Akt phosphorylation was also observed in primary fibroblasts from human fibro-adipose overgrowth syndrome patients with mosaic, heterozygous expression of the $PIK3CA^{H1047L}$ mutation²¹ (Fig. 5f)’.

We have now corrected this, and refer to Supplementary Fig. 5f in this sentence.

Also Fig 6 d should be supplementary Fig 6d in the following: ‘and in MCF-10A human mammary epithelial cells transfected with p110 α^{H1047R} (Fig. 6d).

We have now corrected this, and refer to Supplementary Fig. 6d in this sentence.

6- Figure 6 is missing (D).

This comment is unclear to us, as we had not included a Fig. 6d. Fig. 6 is composed by ‘a, b (two lower graphs: left and right – two transformed clones) and c’.

We can see that the accompanying text and figure layout might be confusing, and we have updated the text as follows: ‘We next tested the impact of these inhibitors on spontaneously in vitro transformed p110 α^{H1047R} MEFs that differ in the percentage of diploid/tetraploid cells in the cell population (Fig. 6b; showing 2 clones – left and right graphs).

Reviewers' Comments:

Reviewer #1 (Remarks to the Author)

In the revised manuscript, Berenjeno et al. have responded to all of the Reviewers' comments. However, while the first half of this manuscript shows a novel role (well supported by the data) of oncogenic PIK3CA in centrosome amplification, the second part demonstrating a role of PIK3CA in inducing tolerance to genome doubling remains very confusing.

The authors show that PIK3CA activation does not cause: 1) proliferation defects, 2) senescence, 3) apoptosis, 4) increases in binucleated cells, 5) chromosome segregation errors, or, 6) inactivation of TP53. Yet, more than 50% of the MEFs-H1047R are aneuploid. How PIK3CA activation causes aneuploidy remains unknown.

Because the authors cannot explain the mechanisms by which the number of aneuploid cells increase by PIK3CA activation, they hypothesize that this mutation increases the tolerance to spontaneous tetraploidization of MEFs. While this is a plausible explanation, the data in the manuscript do not provide enough evidence to state this in the title of the manuscript, or make the general conclusion that PI3K activation does this in vivo.

Importantly, comparison of the DNA content by FACS in Figure S5e (which shows normal DNA content and cell cycle profiles) is in direct contradiction with the percent of aneuploid cells shown in Figure 5a-b (which shows that more than 50% of cells are aneuploid).

Reviewer #3 (Remarks to the Author)

The paper by Inma M. Berenjeno and colleagues investigated the impact of mutant PIK3CA during the early stages of malignancy. The authors generated a mouse model in which the *Pik3ca*H1047R hotspot mutation is activated from its endogenous locus and in the heterozygous state. The heterozygously expressed mutant *Pik3ca* had a major impact during embryonic development and also on adult life, however it was not sufficient to initiate tumor formation on its own, but when expressed together with intestine-specific heterozygous deletion of the *Apc* tumor suppressor gene the progression of the colon cancer was accelerated.

The authors used primary MEFs as the main model to study the early cellular impact of the expression of the mutant *Pik3ca*H1047R. Characterization of mutant MEFs revealed enhanced centrosome amplification by centriole over duplication, which is controlled by the activation of Akt/ROCK signaling. The centrosome amplification did not lead to more segregation errors. Analysis of metaphase chromosome spreads showed an increase in aneuploidy cells in *Pik3ca*H1047R MEFs populations, which points towards the possibility that *Pik3ca*H1047R expression induces tolerance to genome doubling rather than by the route of p53 inactivation. A further genetic analysis showed that in breast cancer, mutations in *Pik3ca* and p53 showed a tendency to be mutual exclusive, suggesting a role of *Pik3ca* mutation as a potential tolerance mechanism for genome doubling independently of the p53 inactivation pathway. Finally, the authors showed that pharmacological inhibition of the PI3K/ROCK pathway can reduce the fraction of the cells that have centrosome amplification, but cannot revert the aneuploidy once it has been established, which can explain the currently limited therapeutic impact of PI3K-targeted therapies. However, these inhibitors can be used to reduce tumor heterogeneity and evolution by limiting the perpetuation of genomically unstable tetraploid cells.

The findings in this manuscript are significant however a few additional changes to the manuscript would be required:

1. Have the authors considered testing siRNA against p110 α H1047R in order to prevent induction of tetraploidy in stably-transformed tetraploid MEFs and Nutu cells? It might be possible that

silencing in the mRNA level could rescue the normal ploidy, where treatment with p110 α , Akt or ROCK inhibitors failed to do so.

Nam et al, 2010 showed that knockdown of Akt with small interfering RNAs and overexpression of phosphatase and tensin homolog or dominant-negative Akt abrogated supernumerary centrosome formation, therefore it would be interesting to test the siRNA effect on the tetraploidisation inhibition.

2. Fig. 1a- if the pFOXO3a-T32 serves as an indicator for protein amount, then there is relatively higher detection in Pik3caH1047R+neo after 4-OHT treatment, while in all other samples the detection of the band is not so clear. Therefore the induction of pAkt-S473 in this sample relative to the others is not so convincing. The authors should consider using another antibody as a control (like β -actin, Vicullin or α -tubulin that were also used in this manuscript).

3. Can the authors explain why in Fig. 3e the band of pNPM-T199 is not detected in the middle WT MEFs cell extracts where no CDK2 inhibitor was added?

4. Fig. 3d- why one of the H1047R MEF extracts showed no expression of pNPM-T199 after the addition of p110 α inhibitor (A66), while the second extract showed a band?

5. Supplementary Fig. 3 - Can the authors explain why there is an especially stronger activation of the PI3K pathway in skin, colon and bladder compared to the other tissues?

6. Supplementary Fig. 8a - the pMLC-S19 band after 6h in the WT is decreased and increased again after 16h, shouldn't the level remain constant?

Minor points:

1. In the results section: "A mouse model of inducible expression of p110 α H1047R from its endogenous locus" Fig. 1b should be: Supplementary Fig. 1b.

2. Supplementary Tables 1 and 2 are missing from the manuscript.

3. In the results section: "Efficient clustering of extra centrosomes and lack of chromosome segregation errors upon p110 α H1047R expression" Supplementary Fig. 4a should be: Fig 4a.

4. Supplementary Fig. 13c is missing from the manuscript.

5. In the results section: "Pharmacological intervention before and after establishment of tetraploidy" Supplementary Fig. 15 should be: Supplementary Fig. 16.

6. Fig. 3f statistics is missing.

7. Supplementary Fig. 1c - the pAkt-T308 band in Breast-H1047R sample is not so clear, and also p110 α and Akt are missing from the blot.

8. Supplementary Fig. 13a- consider presenting the lines in the graph in color since they are overlapping and it is a bit difficult to distinguish between the treatments.

9. Supplementary Fig. 17b - the legend of the Y axis is missing.

Reply to Reviewers' Comments

Reviewer #1 (Remarks to the Author):

'Because the authors cannot explain the mechanisms by which the number of aneuploid cells increase by PIK3CA activation, they hypothesize that this mutation increases the tolerance to spontaneous tetraploidization of MEFs. While this is a plausible explanation, the data in the manuscript do not provide enough evidence to state this in the title of the manuscript, or make the general conclusion that PI3K activation does this *in vivo*'.

In our view, the observation in primary MEFs that pharmacological blockade of p110 α PI3K or its downstream targets Akt or ROCK during the induction phase of p110 α ^{H1047R} expression prevents tetraploidisation (Fig. 6a) proves that PIK3CA mutation provides tolerance to tetraploidisation.

This is also demonstrated under more 'artificial' conditions whereby p110 α ^{H1047R} MEFs were treated with DCB, an agent that blocks cytokinesis, to experimentally increase the fraction of tetraploid cells in the cultures. These data revealed that the number of p110 α ^{H1047R} cells dividing after DCB washout is much higher than in wild-type cells (60% versus 20%, respectively; Fig. 5d) indicating that p110 α ^{H1047R} cells display a higher tolerance to tetraploidy.

While we agree with the Referee that the underlying mechanism remains unclear at this point, we believe that this is an important observation that warrants communication. As mentioned in the cover letter of our resubmission in December 2016, bioinformatic studies from independent research groups in London and Harvard have now identified PIK3CA mutation as one of the key drivers of genome copy number changes/genome duplication in adenocarcinoma of the breast and colon. These results are currently being submitted for publication.

On the notion of a possible *in vivo* role for PI3K activation in tolerance to tetraploidisation, we believe that rather than firmly concluding that this phenomenon occurs *in vivo*, the text is sufficiently speculative.

Reviewer #3 (Remarks to the Author):

The findings in this manuscript are significant. However a few additional changes to the manuscript would be required:

1. Have the authors considered testing siRNA against p110 α ^{H1047R} in order to prevent induction of tetraploidy in stably-transformed tetraploid MEFs and Nutu cells? It might be possible that silencing in the mRNA level could rescue the normal ploidy, where treatment with p110 α , Akt or ROCK inhibitors failed to do so. Nam et al, 2010 showed that knockdown of Akt with small interfering RNAs and overexpression of phosphatase and tensin homolog or dominant-negative Akt abrogated supernumerary centrosome formation, therefore it would be interesting to test the siRNA effect on the tetraploidisation inhibition.

The use of the word 'prevent' in this context is not entirely clear to us, given that we had shown that kinase inhibitors can prevent the tolerance to tetraploidisation (see comments to Referee 1 above). We assume that the Referee instead means 'revert', and also believes that a protein scaffolding role of mutant p110 α could be responsible for the continued maintenance of tetraploidy in stably-transformed tetraploid MEFs and Nutu cells. We believe that this experiment has little relevance for the potential therapeutic intervention of PI3K activity in cancer, which would be achieved pharmacologically and not via RNAi in the clinic. It would also be technically very difficult to selectively silence the point-mutated gene over the wild-type *Pik3ca* gene using RNAi.

2. Fig. 1a - if the pFOXO3a-T32 serves as an indicator for protein amount, then there is relatively higher detection in Pik3ca^{H1047R+neo} after 4-OHT treatment, while in all other samples the detection of the band is not so clear. Therefore the induction of pAkt-S473 in this sample relative to the others is not so convincing.

The authors should consider using another antibody as a control (like β -actin, Vicullin or α -tubulin that were also used in this manuscript).

The pFOXO3a-T32 does not serve as an indicator of protein amount, but as a readout for PI3K pathway activation following recombination of the *Pik3ca*^{H1047R+neo} gene upon 4-OHT treatment. The data are perfectly in line with the expectation that pFOXO3a-T32 is only observed in the *Pik3ca*^{H1047R+neo} lane. Instead, the total levels of the Akt protein serve as a protein loading control for the gel.

3. Can the authors explain why in Fig. 3e the band of pNPM-T199 is not detected in the middle WT MEFs cell extracts where no CDK2 inhibitor was added?

This is the lane that shows short-term (2 h) treatment with Akt inhibitor, which effectively blocks the levels of pNPM-T199 in WT cells but not in *Pik3ca* mutant cells. The latter need longer drug treatment to observe this inhibition, as displayed in Suppl. Fig 8a (showing the impact of 6 h and 16 h treatment with Akt inhibitor). The explanation for this is not clear at the moment.

4. Fig. 3d - why one of the H1047R MEF extracts showed no expression of pNPM-T199 after the addition of p110 α inhibitor (A66), while the second extract showed a band?

This has most likely to do with the observed lower level of pNPM-T199 in the second H1047R MEF line tested (top left part of the gel), relative to the first MEF line. Upon A66 treatment, this weaker signal in this second H1047R MEF is therefore no longer detectable, in line with the strong reduction in the pNPM-T199 signal of the first H1047R MEF.

5. Supplementary Fig. 3 - Can the authors explain why there is an especially stronger activation of the PI3K pathway in skin, colon and bladder compared to the other tissues?

This is indeed an intriguing finding, given the apparently similar mutant *Pik3ca* recombination efficiency in all tissues tested, as shown in Suppl. Fig. 3a. We do not have an explanation for this at the moment.

6. Supplementary Fig. 8a - the pMLC-S19 band after 6h in the WT is decreased and increased again after 16h, shouldn't the level remain constant?

We currently have no explanation for this observation. We suspect this might have something to do with so-called feedback loops of 'rebound' PI3K pathway activity frequently observed upon long-term treatment with pathway inhibitors. This hypothesis is supported by slightly enhanced pAkt after 16h treatment with Akt inhibitor.

Minor points:

We thank this Referee for going through our dataset so carefully.

1. In the results section: "A mouse model of inducible expression of p110 α ^{H1047R} from its endogenous locus" Fig. 1b should be: Supplementary Fig. 1b.

In this part of the text, we should have referred to Fig. 1a instead of Fig. 1b; this has now been corrected. The decreased levels of p110 α ^{H1047R} in the hypomorph model can be seen in both Fig. 1a and Supplementary Fig. 1b. We have updated the text accordingly.

2. Supplementary Tables 1 and 2 are missing from the manuscript.

These tables are now included in the uploaded Supplementary Figures Word File.

3. In the results section: "Efficient clustering of extra centrosomes and lack of chromosome segregation errors upon p110 α ^{H1047R} expression" Supplementary Fig. 4a should be: Fig 4a.

This has now been addressed.

4. Supplementary Fig. 13c is missing from the manuscript.

The text should have referred to Supplementary Fig. 14c rather than Supplementary Figure 13c. This has now been corrected.

5. In the results section: “Pharmacological intervention before and after establishment of tetraploidy” Supplementary Fig. 15 should be: Supplementary Fig. 16.

This has now been addressed.

6. Fig. 3f: statistics is missing.

As was stated in the legend to Fig. 3f (see below), this figure shows data from 2 independent experiments, precluding the use of statistical tests.

.... (f) Impact of inhibition of p110 α (by 3 μ M A66), Akt (by 1 μ M Akti X) or ROCK (by 10 μ M Y27632 or 0.5 μ M H1152) on p110 α ^{H1047R}-induced centrosome amplification in primary MEFs. 100 cells were scored per condition using 2 independent p110 α ^{H1047R} MEF lines; values = mean \pm SD.

7. Supplementary Fig. 1c – the pAkt-T308 band in Breast-^{H1047R} sample is not so clear, and also p110 α and Akt are missing from the blot.

We showed this dataset to illustrate the lack of PI3K pathway activation *before* recombination of the mutant *Pik3ca* allele, and we believe the dataset as it stands makes this point sufficiently clear.

8. Supplementary Fig. 13a - consider presenting the lines in the graph in color since they are overlapping and it is a bit difficult to distinguish between the treatments.

This has now been addressed.

9. Supplementary Fig. 17b – the legend of the Y-axis is missing.

This has now been addressed.

Berenjeno et al. - Reply to Reviewers' Comments

Reviewer #1 (Remarks to the Author):

'In the revised manuscript, Berenjeno *et al.* have responded to all of the Reviewers' comments. (.....) Because the authors cannot explain the mechanisms by which the number of aneuploid cells increase by *PIK3CA* activation, they hypothesize that this mutation increases the tolerance to spontaneous tetraploidization of MEFs. While this is a plausible explanation, (1) the data in the manuscript do not provide enough evidence to state this in the title of the manuscript, or (2) make the general conclusion that PI3K activation does this *in vivo*'.

With regards to the second point, it is important to point out that we did not conclude that PI3K activation provides *in vivo* tolerance to tetraploidisation. We had tried to make the relevant text in the manuscript sufficiently speculative, but this may not have been clear enough. Therefore, in response to this Referee comment, we have now updated the text as follows (additional text inserted is underlined):

- (abstract): 'We report here that mutant *Pik3ca* induces centrosome amplification in cultured cells (through a pathway involving AKT, ROCKII and CDK2/Cyclin E-Nucleophosmin) as well as in mouse tissues, and also increases *in vitro* cellular tolerance to spontaneous genome doubling.'
- (results section p7): 'However, it remains to be demonstrated that *PIK3CA* mutation can provide tolerance to tetraploidisation *in vivo*.'

With regards to the first point, while we agree the underlying mechanism is unclear, we believe that our data are sufficiently strong to warrant mentioning in the manuscript as we currently do, especially if we make the changes to the abstract as mentioned above. Indeed, our data in primary MEFs showing that pharmacological blockade of p110 α PI3K (or its downstream target ROCK) during the induction phase of p110 α ^{H1047R} expression prevents tetraploidisation (Fig. 6a) indicates that *PIK3CA* mutation provides tolerance to tetraploidisation by helping to establish tetraploids. This is clearly supported by data under more 'artificial' conditions whereby p110 α ^{H1047R} MEFs were treated with DCB, an agent that blocks cytokinesis, to experimentally increase the fraction of tetraploid cells in the cultures. These data revealed that the number of p110 α ^{H1047R} cells dividing after DCB washout is much higher than in wild-type cells (60% *versus* 20%, respectively; Fig. 5d), indicating that p110 α ^{H1047R} cells display a higher tolerance to tetraploidy.

We believe that to mention that *PIK3CA* mutation can provide tolerance to spontaneous and chemically-induced tetraploidisation is important, especially now that our observations are supported by bioinformatic studies from research groups in London and Harvard who, independently, whilst being unaware of our data, have recently identified *PIK3CA* mutation as one of the earliest mutation events, typically occurring before whole-genome doubling or other copy number changes, in several cancer types, including adenocarcinoma of the breast and colon (*Peter Van Loo; personal communication - data currently being submitted for publication*). These data are in line with our own bio-informatic analysis of human breast cancer which shows that *PIK3CA* mutations in these cancers are found to generally precede the genome doubling event (Fig. 5f).

Reviewer #3 (Remarks to the Author):

The findings in this manuscript are significant. However a few additional changes to the manuscript would be required:

1. Have the authors considered testing siRNA against p110 α ^{H1047R} in order to prevent induction of tetraploidy in stably-transformed tetraploid MEFs and Nutu cells? It might be possible that silencing in the mRNA level could rescue the normal ploidy, where treatment with p110 α , Akt or ROCK inhibitors failed to do so. Nam *et al.* 2010 showed that knockdown of Akt with small interfering RNAs and overexpression of phosphatase and tensin homolog or dominant-negative Akt abrogated supernumerary centrosome formation, therefore it would be interesting to test the siRNA effect on the tetraploidisation inhibition.

Unfortunately, the proposed experiment to rescue the normal ploidy by RNAi is not similar to the experiment performed in the Nam *et al.* 2010 study and we believe also technically not feasible, as explained below.

1. We are not aware of precedents for reversal of stable tetraploidy to diploidy - Indeed we would contend this is simply not possible for the following reasons. Following the onset of tetraploidy, chromosomal instability (CIN) propagates much more readily. As we and others have shown, this is reflected by both numerical and structural chromosomal aberrations and a loss of chromosome complement resulting in cells drifting towards a triploid state (see Fig. 1A of our Dewhurst *et al.* paper in *Cancer Discovery* 2014 Feb;4(2):175-85). The resulting karyotype bears little resemblance to the genome immediately pre- or post-genome doubling and it would therefore be inconceivable that such a cell could be reverted to a diploid state. This would entail loss of specific chromosomes and rearrangement of residual chromosomes to a structurally normal karyotype - an impossibility. Therefore, even if this experiment were technically possible (see points 2-4 below), it is inconceivable that genetic manipulation of mutant *PIK3CA* would be able to induce a cell with a complex mix of chromosomes to revert to a normal diploid state.

Importantly, this is unlike centrosome amplification (CA), to which Referee 3 refers in his/her reference to the Nam *et al.* 2010 paper, which is known to be a dynamic, non-genetic phenomenon. This is in line with the notion that in our experiments, we could revert CA but not tetraploidy by small molecule PI3K pathway inhibitors.

2. Transient versus stable RNAi expression - The Referee refers to the Nam *et al.* paper in which the authors tested the impact of Akt RNAi on CA, induced in human cells that stably express a mutant MET tyrosine kinase receptor. Given that CA is a non-genetic, dynamic phenomenon that can be reverted within 24 h, these authors had the opportunity to use transient transfection of RNAi to Akt and did not test the impact of stable expression of RNAi to Akt. In our experiments, reversion of genomic alterations would be expected to take weeks or months to materialise as we would have to stably express shRNA to *PIK3CA*.

3. Selective RNAi of the mutant *PIK3CA* in a heterozygous WT/MUT *PIK3CA* context - Because we are seeking to demonstrate the role of a pathogenic heterozygous activating *Pik3ca* allele specifically, allele-specific knockdown of the mutant would be required. In other words, in cells which express both the wild-type and mutant *PI3KCA* allele, we would have to selectively inactivate the mutant *PIK3CA* allele, which in our model is the consequence of a single point mutation i.e. RNAi would have to be based on a single basepair mismatch. The experiment requested is thus not a 'simple' total *PIK3CA* downregulation approach. As a matter of fact, one of the co-authors of our manuscript (Robert Semple) has tried this approach extensively in human overgrowth cells for other heterozygous point mutations, with modest if any success, after months of effort.

There are also no published data on selective inactivation of the mutant *PIK3CA* allele by RNAi. At the most recent Keystone Conference (Santa Fe, Jan 19-23, 2017), we (BV) have consulted many PI3K experts on this issue, including several scientists from pharmaceutical companies, and unfortunately, no one has been successful at this approach as of yet.

4. Measurement of mutant p110 α protein expression - Following selective downregulation of the RNA of mutant *PIK3CA*, we would have to test the impact of expression of the mutant *versus* the wild-type p110 α protein, which differ in one single amino acid. Unfortunately, there are no antibodies available that discriminate between the mutant and wild-type p110 α protein. One of the authors of our manuscript (Robert Semple) has tried to generate selective antibodies against either the *PIK3CA* 1047 Arg allele (3 rabbits; 32 B-cell lines and derived HEK293 cell lines screened by Immunoprecise Antibodies Ltd) or the *PIK3CA* 1047 Leu allele (5 mice; many hybridomas screened at first round; 10 most promising evaluated further; GenScript), at a cumulative cost of several thousand pounds, however in neither case did the antibodies generated show sufficient specificity by either Western blotting or immunostaining in fixed cells.

2. Fig. 1a - The authors should consider using another antibody as a control (like β -actin, Vicullin or α -tubulin that were also used in this manuscript).

We appreciate that using a loading control that is connected to the signalling pathway in question is not ideal. We have therefore replaced Fig. 1a with a new Fig. 1a in which we use vinculin as a loading control.

We have in the meantime also published a paper in which signalling in *Pik3ca*^{H1047R} MEFs from these mice has been studied extensively (left panel of Fig. 1a in *Scientific Reports* 2017 Jan 6;7:39985 - Reference 39 in manuscript), shown below for the perusal of the Referee. These data confirm clear inducibility of PI3K pathway activation in *Pik3ca* mutant cells, similar as observed in PTEN KO cells. Note that each lane represents an independent MEF line, all cells were treated with 4-OHT.

3. Can the authors explain why in Fig. 3e the band of pNPM-T199 is not detected in the middle wild-type MEFs cell extracts where no CDK2 inhibitor was added?

This is the lane that shows short-term (2 h) treatment with Akt inhibitor, which effectively blocks the levels of pNPM-T199 in wild-type cells but not in *Pik3ca* mutant cells. The latter need longer drug treatment to observe this inhibition, as also displayed in Suppl. Fig 8a (showing the impact of 6 h and 16 h treatment with Akt inhibitor). The explanation for this is not clear at the moment.

4. Fig. 3d - why one of the H1047R MEF extracts showed no expression of pNPM-T199 after the addition of p110 α inhibitor (A66), while the second extract showed a band?

This has most likely to do with the observed lower 'starting' level of pNPM-T199 in the second H1047R MEF line tested (top left part of the gel), relative to the first MEF line. Upon A66 treatment, this weaker signal in this second H1047R MEF is therefore no longer detectable, in line with the strong reduction in the pNPM-T199 signal of the first H1047R MEF.

5. Supplementary Fig. 3 - Can the authors explain why there is an especially stronger activation of the PI3K pathway in skin, colon and bladder compared to the other tissues?

This is indeed an intriguing finding, given the apparently similar mutant *Pik3ca* recombination efficiency in all tissues tested, as shown in Suppl. Fig. 3a. We do not have an explanation for this at the moment, this may have to do with relative activities and/or expression levels of lipid phosphatases in different tissues, although this is pure speculation at this point.

6. Supplementary Fig. 8a - the pMLC-S19 band after 6h in the wild-type is decreased and increased again after 16h, shouldn't the level remain constant?

We currently have no explanation for this observation but suspect that this might have something to do with so-called feedback loops of 'rebound' in PI3K pathway activity that are frequently observed upon long-term treatment with PI3K pathway inhibitors. This is supported by the also slightly enhanced pAkt after 16h treatment with Akt inhibitor.

Minor points:

We thank this Referee for going through our dataset so carefully.

1. In the results section: "A mouse model of inducible expression of p110 α ^{H1047R} from its endogenous locus" Fig. 1b should be: Supplementary Fig. 1b.

In this part of the text, we should have referred to Fig. 1a instead of Fig. 1b; this has now been corrected. The decreased levels of p110 α ^{H1047R} in the hypomorph model can be seen in both Fig. 1a and Supplementary Fig. 1b. We have corrected the text accordingly.

2. Supplementary Tables 1 and 2 are missing from the manuscript.

These tables are now included in the uploaded Supplementary Figures Word File.

3. In the results section: "Efficient clustering of extra centrosomes and lack of chromosome segregation errors upon p110 α ^{H1047R} expression" Supplementary Fig. 4a should be: Fig 4a.

This has now been addressed.

4. Supplementary Fig. 13c is missing from the manuscript.

The text should have referred to Supplementary Fig. 14c rather than Supplementary Fig. 13c. This has now been corrected.

5. In the results section: "Pharmacological intervention before and after establishment of tetraploidy" Supplementary Fig. 15 should be: Supplementary Fig. 16.

This has now been addressed.

6. Fig. 3f: statistics is missing.

As was stated in the legend to Fig. 3f (see below), this figure shows data from 2 independent experiments, precluding the use of statistical tests.

.... (f) Impact of inhibition of p110 α (by 3 μ M A66), Akt (by 1 μ M Akti X) or ROCK (by 10 μ M Y27632 or 0.5 μ M H1152) on p110 α ^{H1047R}-induced centrosome amplification in primary MEFs. 100 cells were scored per condition using 2 independent p110 α ^{H1047R} MEF lines; values = mean \pm SD.

7. Supplementary Fig. 1c – the pAkt-T308 band in Breast-^{H1047R} sample is not so clear, and also p110 α and Akt are missing from the blot.

We showed this dataset to illustrate the lack of PI3K pathway activation *before* recombination of the mutant *Pik3ca* allele, and we believe the dataset as it stands makes this point sufficiently clear.

8. Supplementary Fig. 13a - consider presenting the lines in the graph in color since they are overlapping and it is a bit difficult to distinguish between the treatments.

This has now been addressed.

9. Supplementary Fig. 17b – the legend of the Y-axis is missing.

This has now been addressed.

Berenjeno et al. - Reply to Reviewers' Comments

Reviewer #1 (Remarks to the Author):

(Note from the authors: the numbering between brackets in the Referee comment is added by us, to facilitate answering to each point)

In the revised manuscript, Berenjeno *et al.* have responded to all of the Reviewers' comments. However, while the first half of this manuscript shows a novel role (well supported by the data) of oncogenic *PIK3CA* in centrosome amplification, the second part demonstrating a role of *PIK3CA* in inducing tolerance to genome doubling remains very confusing.

The authors show that *PIK3CA* activation does not cause: 1) proliferation defects, 2) senescence, 3) apoptosis, 4) increases in binucleated cells, 5) chromosome segregation errors, or, 6) inactivation of TP53. Yet, more than 50% of the MEFs-^{H1047R} are aneuploid. How *PIK3CA* activation causes aneuploidy remains unknown.

Because the authors cannot explain the mechanisms by which the number of aneuploid cells increase by *PIK3CA* activation, they hypothesize that this mutation increases the tolerance to spontaneous tetraploidization of MEFs. While this is a plausible explanation, (1) the data in the manuscript do not provide enough evidence to state this in the title of the manuscript, or (2) make the general conclusion that *PI3K* activation does this *in vivo*.

(3) Importantly, comparison of the DNA content by FACS in Figure S5e (which shows normal DNA content and cell cycle profiles) is in direct contradiction with the percent of aneuploid cells shown in Figure 5a-b (which shows that more than 50% of cells are aneuploid).

We have now resolved the issue mentioned by Referee 1 under point (3) above.

As we had mentioned in an earlier response letter to the Referee, the DNA profiles shown in the original Suppl. Fig. 5e and Suppl. Fig. 7d were generated with the aim of analysing the different phases of the cell cycle. We wanted to investigate whether the observed supernumerary centrosomes could be derived from an altered cell cycle profile (i.e. prolonged time in G1/S or G2/M) in *PIK3CA* mutant MEFs. For this reason, the cells with a higher DNA content (>4N) were filtered out on purpose by FACS.

It is also important to mention that in all experiments shown in earlier versions of the manuscript, we had performed metaphase spread (Fig. 5b, Suppl. Fig. 11) and FACS analysis (Suppl. Fig. 5e) on independent MEF lines and in separate experiments.

In a new set of experiments, we have now performed parallel FACS and metaphase spread analysis on the same MEF populations, whereby the >4N cell population was not gated out in the FACS experiments. Both analytic methods reveal an increase in the tetraploid cell population in *PIK3CA* mutant cells compared to WT cells (new Suppl. Figure 13) and also confirm the data on metaphase spread analysis (compare the original Fig. 5b to the new Suppl. Fig. 13c).

We believe that these data support the interpretation for an increase in spontaneous tetraploidisation in *PIK3CA* mutant cells, most likely due to a better propagation of tetraploid cells. In the revised text, we now make it clear that this could be due to increased tolerance to tetraploidy (for which we provide experimental evidence - see below), and also because of efficient centrosome clustering (for which we also provide evidence).

Additional support of a role for *PIK3CA* mutation in tolerance to tetraploidization comes from our experiments of chemically-induced tetraploidization, whereby WT and p110 α ^{H1047R} MEFs were treated with DCB, an agent that blocks cytokinesis, to experimentally increase the fraction of tetraploid cells in the cultures. These data (Fig. 5d), revealed that the number of p110 α ^{H1047R} cells dividing after DCB washout is higher than in wild-type cells (60% versus 20%, respectively).

In order to avoid confusion, we have now removed Suppl. Fig. 5e.

In response to the Referee, we accept it would be prudent to be less explicit in our conclusions with regards to possible tolerance to tetraploidization. We have therefore made textual changes, as highlighted in the relevant text, including in the abstract, as shown below.

*We report here that mutant *Pik3ca* induces centrosome amplification in cultured cells (through a pathway involving AKT, ROCKII and CDK2/Cyclin E-Nucleophosmin) and in mouse tissues, and increased in vitro cellular tolerance to spontaneous genome doubling.*

We have now also included reference in the abstract to our bio-informatic studies performed in human cancers in the MS, as follows: ‘We also present evidence that the majority of *PIK3CA*^{H1047R} mutations in the TCGA breast cancer cohort precede genome doubling.’

Reviewer #3 (Remarks to the Author):

The paper by Inma M. Berenjeno and colleagues investigated the impact of mutant *PIK3CA* during the early stages of malignancy. The authors generated a mouse model in which the *Pik3ca*^{H1047R} hotspot mutation is activated from its endogenous locus and in the heterozygous state. The heterozygously expressed mutant *Pik3ca* had a major impact during embryonic development and also on adult life, however it was not sufficient to initiate tumor formation on its own, but when expressed together with intestine-specific heterozygous deletion of the *Apc* tumor suppressor gene the progression of the colon cancer was accelerated.

The authors used primary MEFs as the main model to study the early cellular impact of the expression of the mutant *Pik3ca*^{H1047R}. Characterization of mutant MEFs revealed enhanced centrosome amplification by centriole over duplication, which is controlled by the activation of Akt/ROCK signaling. The centrosome amplification did not lead to more segregation errors. Analysis of metaphase chromosome spreads showed an increase in aneuploidy cells in *Pik3ca*^{H1047R} MEFs populations, which points towards the possibility that *Pik3ca*^{H1047R} expression induces tolerance to genome doubling rather than by the route of p53 inactivation. A further genetic analysis showed that in breast cancer, mutations in *Pik3ca* and p53 showed a tendency to be mutual exclusive, suggesting a role of *Pik3ca* mutation as a potential tolerance mechanism for genome doubling independently of the p53 inactivation pathway. Finally, the authors showed that pharmacological inhibition of the PI3K/ROCK pathway can reduce the fraction of the cells that have centrosome amplification, but cannot revert the aneuploidy once it has been established, which can explain the currently limited therapeutic impact of PI3K-targeted therapies. However, these inhibitors can be used to reduce tumor heterogeneity and evolution by limiting the perpetuation of genomically unstable tetraploid cells.

The findings in this manuscript are significant however a few additional changes to the manuscript would be required:

1. Have the authors considered testing siRNA against p110 α ^{H1047R} in order to prevent induction of tetraploidy in stably-transformed tetraploid MEFs and Nutu cells? It might be possible that silencing in the mRNA level could rescue the normal ploidy, where treatment with p110 α , Akt or ROCK inhibitors failed to do so. Nam *et al*, 2010 showed that knockdown of Akt with small interfering RNAs and overexpression of phosphatase and tensin homolog or dominant-negative Akt abrogated supernumerary centrosome formation, therefore it would be interesting to test the siRNA effect on the tetraploidisation inhibition.

Unfortunately, the proposed experiment to rescue the normal ploidy by RNAi is not similar to the experiment performed in the Nam *et al*. 2010 study and we believe also technically not feasible, as explained below.

1. We are not aware of precedents for reversal of tetraploidy to diploidy - Indeed we would contend this is simply not possible for the following reasons. Following the onset of tetraploidy, chromosomal instability (CIN) propagates much more readily. As we and others have shown, this is reflected by both numerical and structural chromosomal aberrations and a loss of chromosome complement resulting in cells drifting towards

a triploid state (see Fig. 1A of our Dewhurst *et al.* paper in *Cancer Discovery* 2014 Feb;4(2):175-85). The resulting karyotype bears little resemblance to the genome immediately pre- or post-genome doubling and it would therefore be inconceivable that such a cell could be reverted to a diploid state. This would entail loss of specific chromosomes and rearrangement of residual chromosomes to a structurally normal karyotype - an impossibility. Therefore, even if this experiment were technically possible (see points 2-4 below), it is inconceivable that genetic manipulation of mutant *PIK3CA* would be able to induce a cell with a complex mix of chromosomes to revert to a normal diploid state.

Importantly, this is unlike centrosome amplification (CA), to which Referee 3 refers in his/her reference to the Nam *et al.* 2010 paper, which is known to be a dynamic, non-genetic phenomenon. This is in line with the notion that in our experiments, we could revert CA but not tetraploidy by small molecule PI3K pathway inhibitors.

2. Transient versus stable RNAi expression - The Referee refers to the Nam *et al.* paper in which the authors tested the impact of Akt RNAi on CA, induced in human cells that stably express a mutant MET tyrosine kinase receptor. Given that CA is a non-genetic, dynamic phenomenon that can be reverted within 24 h, these authors had the opportunity to use transient transfection of RNAi to Akt and did not test the impact of stable expression of RNAi to Akt. In our experiments, reversion of genomic alterations would be expected to take weeks or months to materialise as we would have to stably express shRNA to *PIK3CA*.

3. Selective RNAi of the mutant *PIK3CA* in a heterozygous WT/MUT *PIK3CA* context - Because we are seeking to demonstrate the role of a pathogenic heterozygous activating *Pik3ca* allele specifically, allele-specific knockdown of the mutant would be required. In other words, in cells which express both the wild-type and mutant *PIK3CA* allele, we would have to selectively inactivate the mutant *PIK3CA* allele, which in our model is the consequence of a single point mutation i.e. RNAi would have to be based on a single basepair mismatch. The experiment requested is thus not a 'simple' total *PIK3CA* downregulation approach. As a matter of fact, one of the co-authors of our manuscript (Robert Semple) has tried this approach extensively in human overgrowth cells for other heterozygous point mutations, with modest if any success, after months of effort.

There are also no published data on selective inactivation of the mutant *PIK3CA* allele by RNAi. At the most recent Keystone Conference (Santa Fe, Jan 19-23, 2017), we (BV) have consulted many PI3K experts on this issue, including several scientists from pharmaceutical companies, and unfortunately, no one has been successful at this approach as of yet.

4. Measurement of mutant p110 α protein expression - Following selective downregulation of the RNA of mutant *PIK3CA*, we would have to test the impact of expression of the mutant *versus* the wild-type p110 α protein, which differ in one single amino acid. Unfortunately, there are no antibodies available that discriminate between the mutant and wild-type p110 α protein. One of the authors of our manuscript (Robert Semple) has tried to generate selective antibodies against either the *PIK3CA* 1047 Arg allele (3 rabbits; 32 B-cell lines and derived HEK293 cell lines screened by Immunoprecise Antibodies Ltd) or the *PIK3CA* 1047 Leu allele (5 mice; many hybridomas screened at first round; 10 most promising evaluated further; GenScript), at a cumulative cost of several thousand pounds, however in neither case did the antibodies generated show sufficient specificity by either Western blotting or immunostaining in fixed cells.

2. Fig. 1a- if the pFOXO3a-T32 serves as an indicator for protein amount, then there is relatively higher detection in *Pik3ca*^{H1047R+neo} after 4-OHT treatment, while in all other samples the detection of the band is not so clear. Therefore the induction of pAkt-S473 in this sample relative to the others is not so convincing. The authors should consider using another antibody as a control (like β -actin, Vicullin or α -tubulin that were also used in this manuscript).

We appreciate that using a loading control that is connected to the signalling pathway in question is not ideal. We have therefore replaced Fig. 1a with a new Fig. 1a in which we use vinculin as a loading control.

We have in the meantime also published a paper in which signalling in *Pik3ca*^{H1047R} MEFs from these mice has been studied extensively (left panel of Fig. 1a in *Scientific Reports* 2017 Jan 6;7:39985 - Reference 36 in manuscript, shown below for the perusal of the Referee). These data confirm clear inducibility of PI3K pathway activation in *Pik3ca* mutant cells, similar to that observed in PTEN KO cells. Note that each lane

represents an independent MEF line, all cells were treated with 4-OHT. This confirms a very low-level but significant increase in PI3K pathway activation.

3. Can the authors explain why in Fig. 3e the band of pNPM-T199 is not detected in the middle WT MEFs cell extracts where no CDK2 inhibitor was added?

This is the lane that shows short-term (2 h) treatment with Akt inhibitor, which effectively blocks the levels of pNPM-T199 in wild-type cells but not in *Pik3ca* mutant cells. The latter need longer drug treatment to observe this inhibition, as also displayed in Suppl. Fig 8a (showing the impact of 6 h and 16 h treatment with Akt inhibitor). The explanation for this is not clear at the moment.

4. Fig. 3d- why one of the H1047R MEF extracts showed no expression of pNPM-T199 after the addition of p110α inhibitor (A66), while the second extract showed a band?

This has most likely to do with the observed lower 'starting' level of pNPM-T199 in the second H1047R MEF line tested (top left part of the gel), relative to the first MEF line. Upon A66 treatment, this weaker signal in this second H1047R MEF is therefore no longer detectable, in line with the strong reduction in the pNPM-T199 signal of the first H1047R MEF.

5. Supplementary Fig. 3 - Can the authors explain why there is an especially stronger activation of the PI3K pathway in skin, colon and bladder compared to the other tissues?

This is indeed an intriguing finding, given the apparently similar mutant *Pik3ca* recombination efficiency in all tissues tested, as shown in Suppl. Fig. 3a. We do not have an explanation for this at the moment, this may have to do with relative activities and/or expression levels of lipid phosphatases in different tissues, although this is pure speculation at this point.

6. Supplementary Fig. 8a - the pMLC-S19 band after 6h in the WT is decreased and increased again after 16h, shouldn't the level remain constant?

We currently have no explanation for this observation but suspect that this might have something to do with so-called feedback loops of 'rebound' in PI3K pathway activity that are frequently observed upon long-term treatment with PI3K pathway inhibitors. This is supported by the also slightly enhanced pAkt after 16h treatment with Akt inhibitor.

Minor points:

We thank this Referee for going through our dataset so carefully.

We also point out that, due to the inclusion of an additional Suppl. Fig 13, text referring to any figure after Suppl. Fig 13 has a +1 added to the figure numbering.

1. In the results section: “A mouse model of inducible expression of p110 α H1047R from its endogenous locus” Fig. 1b should be: Supplementary Fig. 1b.

In this part of the text, we should have referred to Fig. 1a instead of Fig. 1b; this has now been corrected. The decreased levels of p110 α ^{H1047R} in the hypomorph model can be seen in both Fig. 1a and Supplementary Fig. 1b. We have corrected the text accordingly.

2. Supplementary Tables 1 and 2 are missing from the manuscript.

These tables are now included in the uploaded Supplementary Figures Word File.

3. In the results section: “Efficient clustering of extra centrosomes and lack of chromosome segregation errors upon p110 α ^{H1047R} expression” Supplementary Fig. 4a should be: Fig 4a.

This has now been addressed.

4. Supplementary Fig. 13c is missing from the manuscript.

The text should have referred to Supplementary Fig. 14c (+1: see above) rather than Supplementary Fig. 13c. This has now been corrected.

5. In the results section: “Pharmacological intervention before and after establishment of tetraploidy” Supplementary Fig. 15 (+1: see above) should be: Supplementary Fig. 16 (+1: see above).

This has now been addressed.

6. Fig. 3f: statistics is missing.

As was stated in the legend to Fig. 3f (see below), this figure shows data from 2 independent experiments, precluding the use of statistical tests.

.... (f) Impact of inhibition of p110 α (by 3 μ M A66), Akt (by 1 μ M Akti X) or ROCK (by 10 μ M Y27632 or 0.5 μ M H1152) on p110 α ^{H1047R}-induced centrosome amplification in primary MEFs. 100 cells were scored per condition using 2 independent p110 α ^{H1047R} MEF lines; values = mean \pm SD.

7. Supplementary Fig. 1c – the pAkt-T308 band in Breast-^{H1047R} sample is not so clear, and also p110 α and Akt are missing from the blot.

We showed this dataset to illustrate the lack of PI3K pathway activation *before* recombination of the mutant *Pik3ca* allele, and we believe the dataset as it stands makes this point sufficiently clear.

8. Supplementary Fig. 13a (+1: see above) - consider presenting the lines in the graph in color since they are overlapping and it is a bit difficult to distinguish between the treatments.

This has now been addressed.

9. Supplementary Fig. 17b (+1: see above) – the legend of the Y-axis is missing.

This has now been addressed.

Additional Note: We have removed the scheme in Suppl. Figure 10a (and accompanying text), based on additional feedback received from collaborators whom we have consulted during revision of the manuscript. Indeed, the scheme shown did not fully cover the many different possible scenarios of chromosomal changes observed upon changes in centrosome biology, and could therefore have been misleading.

Reviewers' Comments:

Reviewer #1:

Remarks to the Author:

Berenjeno et al. successfully addressed most of my and other reviewers' comments. As a result, the revised version of their manuscript is significantly improved. Mainly, 1) new FACS analyses confirms an increase in the tetraploid cell population in PIK3CA mutant cells. 2) Additional data presented in Fig 5d shows that chemically-induced tetraploidization is better tolerated in PIK3CA mutant cells.

The discovery that PIK3CA signaling plays a role in tolerance to tetraploidization is novel and important. The authors accurately acknowledge that the mechanisms by how this works are not clear and need further investigation. However, the scope of this manuscript is vast and the findings provide a potential mechanism by which tetraploid cells with wild-type p53 can proliferate.

Minor points:

Line 146. Is Fig. 1b correct?

Lines 244-248. The statement "When considering the absolute numbers ..." is unnecessary and confusing.

In figure 5b, there is no clear explanation why the x-axis (number of chromosomes) is different than 5c.

REVIEWERS' COMMENTS

Reviewer #1 (Remarks to the Author):

Berenjeno et al. successfully addressed most of my and other reviewers' comments. As a result, the revised version of their manuscript is significantly improved. Mainly, 1) new FACS analyses confirms an increase in the tetraploid cell population in PIK3CA mutant cells. 2) Additional data presented in Fig. 5d shows that chemically-induced tetraploidization is better tolerated in PIK3CA mutant cells.

The discovery that PIK3CA signaling plays a role in tolerance to tetraploidization is novel and important. The authors accurately acknowledge that the mechanisms by how this works are not clear and need further investigation. However, the scope of this manuscript is vast and the findings provide a potential mechanism by which tetraploid cells with wild-type p53 can proliferate.

We are pleased with this feedback from the Referee.

Minor points:

Line 146. Is Fig. 1b correct?

We have now correctly referred to this figure. Please note that, in response to the Nature Communications Editorial Office, we have now included additional display items in the Figures, and transferred some supplementary items to the main manuscript, this has resulted in a renumbering of the figures. The original Fig. 1b is now Supplementary Fig. 1c.

Lines 244-248. The statement "When considering the absolute numbers ..." is unnecessary and confusing.

We agree, and have removed this first part of the sentence.

In figure 5b, there is no clear explanation why the x-axis (number of chromosomes) is different than 5c.

The x-axis of Fig. 5c (now Fig. 6c) has now been made similar to that of Fig. 5b (now Fig. 6b).